# Dendritic nonlinearities are tuned for efficient spike-based computations in cortical circuits

**Balázs B Ujfalussy[1,2,3,4*], Judit K Makara[4,5], Tiago Branco[3,6], Máté Lengyel[1,7]**

[1]Computational and Biological Learning Lab, Department of Engineering, University of Cambridge, Cambridge, United Kingdom; [2]Wigner Research Centre for Physics, Hungarian Academy of Sciences, Budapest, Hungary; [3]MRC Laboratory of Molecular Biology, Cambridge, United Kingdom; [4]Lendület Laboratory of Neuronal Signaling, Institute of Experimental Medicine, Hungarian Academy of Sciences, Budapest, Hungary; [5]Janelia Farm Research Campus, Howard Hughes Medical Institute, Ashburn, United States; [6]Wolfson Institute for Biomedical Research, University College London, London, United Kingdom; [7]Department of Cognitive Science, Central European University, Budapest, Hungary

**Abstract** Cortical neurons integrate thousands of synaptic inputs in their dendrites in highly nonlinear ways. It is unknown how these dendritic nonlinearities in individual cells contribute to computations at the level of neural circuits. Here, we show that dendritic nonlinearities are critical for the efficient integration of synaptic inputs in circuits performing analog computations with spiking neurons. We developed a theory that formalizes how a neuron's dendritic nonlinearity that is optimal for integrating synaptic inputs depends on the statistics of its presynaptic activity patterns. Based on their *in vivo* preynaptic population statistics (firing rates, membrane potential fluctuations, and correlations due to ensemble dynamics), our theory accurately predicted the responses of two different types of cortical pyramidal cells to patterned stimulation by two-photon glutamate uncaging. These results reveal a new computational principle underlying dendritic integration in cortical neurons by suggesting a functional link between cellular and systems–level properties of cortical circuits.

**\*For correspondence:** balazs.ujfalussy@gmail.com

**Competing interests:** The authors declare that no competing interests exist.

## Introduction

The dendritic tree of a cortical neuron performs a highly nonlinear transformation on the thousands of inputs it receives from other neurons, sometimes resulting in a markedly sublinear (*Longordo et al., 2013*) and often in strongly superlinear integration of synaptic inputs (*Losonczy and Magee, 2006*; *Nevian et al., 2007*; *Branco and Häusser, 2011*; *Makara and Magee, 2013*). These nonlinearities have been traditionally studied from the perspective of single-neuron computations, using a few well-controlled synaptic stimuli, revealing a remarkable repertoire of arithmetic operations that the dendrites of cortical neurons carry out (*Poirazi and Mel, 2001*; *London and Häusser, 2005*; *Branco et al., 2010*) including additive, multiplicative and divisive ways of combining individual synaptic inputs in the cell's response (*Silver, 2010*). More recently, the role of nonlinear dendritic integration in actively shaping responses of single neurons under *in vivo* conditions has been demonstrated in several cortical areas including the hippocampus (*Grienberger et al., 2014*), as well as visual (*Smith et al., 2013*) and somatosensory cortices (*Xu et al., 2012*; *Lavzin et al., 2012*; *Palmer et al., 2014*).

**eLife digest** Imagine that you are in the habit of checking three different weather forecasts each day, and then one day in early September the first forecast suddenly predicts snow. If you live in an area where it doesn't normally snow in September, your initial reaction is likely to be surprise. However, you will not be quite so surprised to see a prediction of snow in the second forecast, and by the third forecast you will hardly be surprised at all.

In these three cases, you have responded to the same piece of information in a different way. In mathematics, this type of response is referred to as "nonlinear" because the output (varying degrees of surprise) is not directly proportional to the input (identical predictions of snow). In the case of the weather forecasts, the source of the nonlinearity was the fact that the three predictions were not truly independent. Instead, they corresponded with one another, or "correlated", because all three depended on the weather itself.

In the brain, a single neuron can receive thousands of inputs from other cells. These are received via junctions called synapses that form between the cells. In many cases, the synapses form on the receiving neuron's dendrites – the short branches that protrude from its cell body. Each dendrite can receive signals from hundreds of other neurons, and must combine these inputs to produce a single neuronal response. How dendrites do this is not clear.

Ujfalussy et al. have now developed a computational model that predicts the optimal response of dendrites to complex and realistic inputs from other neurons. The model shows that when dendrites receive inputs from neurons that independently respond to different stimuli, the optimal response is for the dendrites to average the inputs. This is a form of linear processing. By contrast, when the inputs are correlated – for example, because they come from neurons responding to the same stimulus – the optimal response is nonlinear processing. In this and other cases, the optimal response predicted by the model is similar to the response observed in real dendrites.

The model also makes a number of testable predictions; for example, that neurons with correlated activities will tend to form clusters of synapses close together on the dendrites of a target neuron, whereas neurons with unrelated activities will tend to form synapses that are further apart. Somewhat unexpectedly, Ujfalussy et al. show that compensating for input correlations accounts for almost all the nonlinearities that can be found in real neurons' dendrites – at least in response to relatively simple input patterns. Thus, it remains to be shown whether nonlinear dendritic responses to more complex input patterns can also be explained by this single principle. Further studies are also required to understand how different plasticity mechanisms enable neurons to achieve this close match between input correlations and dendritic processing.

However, while many of the basic biophysical mechanisms underlying these nonlinearities are well understood (*Stuart et al., 2007*), it has proven a daunting task to include all these mechanisms in larger scale network models to understand their interplay at the level of the circuit (*Herz et al., 2006*). Conversely, studies of cortical computation and dynamics have largely ignored the complex and highly nonlinear information processing capabilities of the dendritic tree and concentrated on circuit-level computations emerging from interactions between point-like neurons with single, somatic nonlinearities (*Hopfield, 1984*; *Seung and Sompolinsky, 1993*; *Gerstner and Kistler, 2002*; *Vogels et al., 2011*). Therefore, it is unknown how dendritic nonlinearities in individual cells contribute to computations at the level of a neural circuit.

A limitation of most theories of nonlinear dendritic integration is that they focus on highly simplified input regimes (*Mel et al., 1998*; *Poirazi et al., 2003*; *Archie and Mel, 2000*; *Poirazi and Mel, 2001*; *Ujfalussy et al., 2009*), essentially requiring both the inputs and the output of a cell to have stationary firing rates. This approach thus ignores the effects and consequences of temporal variations in neural activities at the time scale of inter-spike intervals characteristic of *in vivo* states in cortical populations (*Crochet et al., 2011*; *Haider et al., 2013*). In contrast, we propose an approach which is specifically centered on these naturally occurring statistical patterns – in analogy to the principle of 'adaptation to natural input statistics' which has been highly successful in accounting for the input-output relationships of cells in a number of sensory areas at the systems level (*Simoncelli and*

*Olshausen, 2001*). We pursued this principle in understanding the integrative properties of individual cortical neurons, for which the relevant statistical input patterns are those characterising the spatio-temporal dynamics of their presynaptic spike trains. Thus, rather than modelling specific biophysical properties of a neuron directly, our goal was to predict the phenomenological input integration properties that result from those biophysical properties and are matched to the statistics of the presynaptic activities.

Our theory is based on the observation that cortical neurons mainly communicate by action potentials, which are temporally punctate all-or-none events. In contrast, the computations cortical circuits perform are commonly assumed to involve the transformations of analog activities varying continuously in time, such as firing rates or membrane potentials (*Rumelhart et al., 1986*; *Hopfield, 1984*; *Dayan and Abbott, 2001*; *Archie and Mel, 2000*; *London et al., 2010*). This implies a fundamental bottleneck in cortical computations: the discrete and stochastic firing of spikes by neurons conveys only a limited amount of information about their rapidly fluctuating activities (*Pfister et al., 2010*; *Sengupta et al., 2014*). Formalising the implications of this bottleneck mathematically reveals that the robust operation of a circuit requires its neurons to integrate their inputs in highly nonlinear ways that specifically depend on two complementary factors: the computation performed by the neuron and the long-term statistics of the inputs it receives from its presynaptic partners.

To critically evaluate our theory, we first illustrate qualitatively the nonlinearities that most efficiently overcome the spiking bottleneck for different classes of presynaptic correlation structures. Next, to provide biophysical insight, we demonstrate that the form of optimal input integration for these presynaptic correlations can be efficiently approximated by a canonical, biophysically-motivated model of dendritic integration. Finally, we test the prediction that cortical dendrites are optimally tuned to their input statistics in *in vitro* experiments. For this, we use available *in vivo* data to characterize the presynaptic population activity of two different types of cortical pyramidal cells. Based on these input statistics, our theory accurately predicts the integrative properties of the postsynaptic dendrites measured in two-photon glutamate uncaging experiments. We also show that NMDA receptor activation is necessary for dendritic integration to approximate the optimal response. These results suggest a novel functional role for dendritic nonlinearities in allowing postsynaptic neurons to integrate their richly structured synaptic inputs near-optimally, thus making a key contribution to dynamically unfolding cortical computations.

## Results

Suppose that every day you check your three favorite websites for the weather forecast. On a September day, the first website forecasts snow which you find hard to believe as it is highly unusual in your area – so you dismiss it as the forecaster's mistake. However, when you read a similar forecast on the second site, you become convinced that snow is coming, and by the time the third site brings you the same news you are hardly surprised at all. Thus, even though all three sources conveyed the same information (snow), they had different impact on you – in other words, their cumulative effect was *nonlinear*. This nonlinearity was due to the fact that the information you get from these sites tends to be correlated as they are all related to a common cause, the actual weather. Below we argue that the same fundamental statistical principle, that correlated information sources require nonlinear integration, accounts for the dendritic nonlinearities of cortical pyramidal neurons.

### Overcoming the spiking bottleneck in circuit computations

To introduce our theory, we consider a postsynaptic neuron computing some function, $f$, of the activity of its presynaptic partners, $\mathbf{u}$ (*Figure 1A*, top):

$$\dot{v} = f(\mathbf{u}) \tag{1}$$

where $\dot{v}$ is the resultant temporal change of the activity of the postsynaptic neuron. We chose $\mathbf{u}$ and $v$ to be analog variables, rather than for example digital spike trains, in line with the vast bulk of theories of network computations (*Hopfield, 1984*; *Dayan and Abbott, 2001*; *Pouget et al., 2003*) and experimental results suggesting analog coding in the cortex (*London et al., 2010*; *Shadlen and Newsome, 1998*). In particular, we considered these variables to correspond to the coarse-grained (low-pass filtered) somatic membrane potentials of neurons (in particular, excluding the action

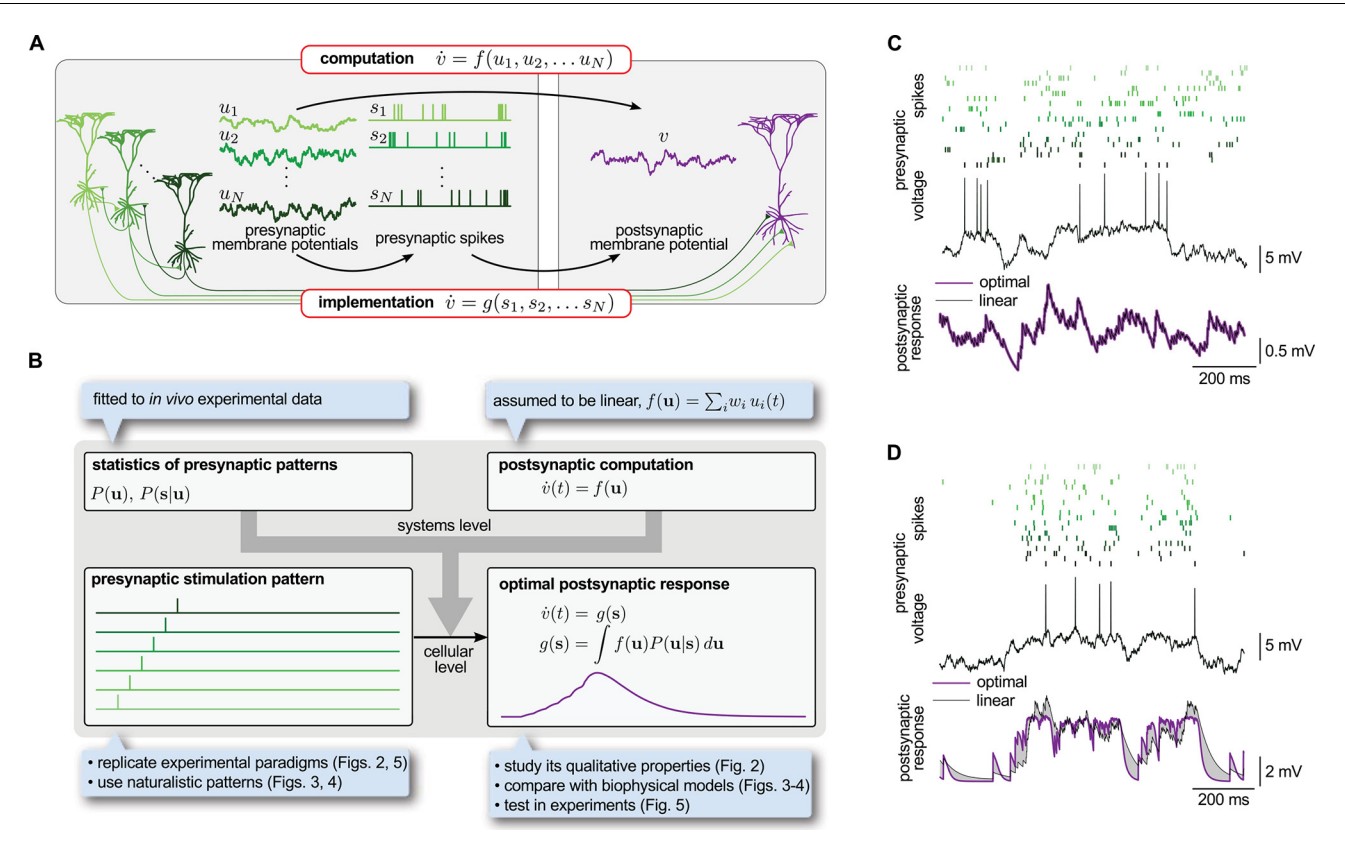

**Figure 1.** Spike-based implementation of analogue computations in neural circuits. (A) Computation (top) is formalized as a mapping, $f$, from presynaptic activities, $u_1, \ldots u_N$ (left), to the postsynaptic activity, $v$ (right). As neurons communicate with spikes, the implementation (bottom) of any computation must be based on the spikes the presynaptic neurons emit, $s_1, \ldots s_N$ (middle). Optimal input integration in the postsynaptic cells requires that the output of $g$ is close to that of $f$. (B) The logic and plan of the paper. Grey box in the center shows theoretical framework, blue boxes around it show steps necessary to apply the framework to neural data. To compute the transformation from stimulation patterns (bottom left) to the optimal response (bottom right) we assumed linear computation (top right) and specified the presynaptic statistics based on cortical population activity patterns observed *in vivo* (top left). To demonstrate the validity of the approach, we studied the fundamental qualitative properties of the optimal response (*Figure 2*), compared it to biophysical models (*Figures 3–4*) and tested it in *in vitro* experiments (*Figure 5*). (C) The optimal postsynaptic response (purple line, bottom) linearly integrates spikes from different presynaptic neurons (top: rasters in shades of green; middle: membrane potential of one presynaptic cell) if their activities are statistically independent. (D) Optimal input integration becomes nonlinear (purple line, bottom) if the activities of the presynaptic neurons are correlated (rasters in shades of green, top), even though the long-term statistics and spiking nonlinearity of individual neurons remains the same as in (C). In this case, the best linear response (black line, bottom) is unable to follow the fluctuations in the signal.

The following figure supplements are available for figure 1:

**Figure supplement 1.** An example of supralinear input integration with firing rate-based rather than membrane potential-based computations.

**Figure supplement 2.** The range of total dendritic inputs *in vitro* and *in vivo*.

**Figure supplement 3.** Nonlinear computation.

potentials themselves, as often reported in experimental data; *Carandini and Ferster, 2000*), although the theory can equally be formalized in terms of instantaneous firing rates (Materials and methods, *Figure 1-figure supplement 1*).

The standard description of neural circuit dynamics in *Equation 1* hides an important informational bottleneck intrinsic to the operation of cortical circuits. While according to *Equation 1*, the postsynaptic neuron's analog activity, $v$, is required to depend directly on the analog activities of its presynaptic partners, $\mathbf{u}$, in reality it only accesses these presynaptic activities through the spikes the

presynaptic population transmits, s, incurring a substantial loss of information (*Alenda et al., 2010*; *Sengupta et al., 2014*). Therefore, the function $g$ a neuron actually implements on its inputs can only depend directly on the presynaptic spikes, not the underlying activities (*Figure 1A*, bottom):

$$\dot{v} = g(\mathbf{s}) \tag{2}$$

Importantly, while $f(\mathbf{u})$ is dictated by the computational function of the circuit, the actual transformation of the synaptic input to the postsynaptic response, expressed by $g(\mathbf{s})$, is determined by the morphological and biophysical properties of the cell. (For these purposes, we regard the presynaptic side of synapses, transforming presynaptic spike trains to synaptic transmission events, as conceptually being part of the postsynaptic cell's $g(\mathbf{s})$ function.) How can then the neuron integrate the incoming presynaptic spikes, as formalized by $g(\mathbf{s})$, such that the resulting postsynaptic response best matches the required computational function, $f(\mathbf{u})$, thereby alleviating the fundamental informational bottleneck of spiking-based communication?

Determining the best $g(\mathbf{s})$ is nontrivial because the same presynaptic spike train may be the result of many different underlying presynaptic activities (*Paninski, 2006*), each potentially implying a different output of the computational function. This ambiguity is formalized mathematically as a posterior probability distribution, $\mathrm{P}(\mathbf{u}|\mathbf{s})$, expressing the probability that the analog activities of the presynaptic cells might currently be $\mathbf{u}$ given their spike trains, s (*Pfister et al., 2010*; *Ujfalussy et al., 2011*). The optimal response, i.e. the $g(\mathbf{s})$ that minimizes the average squared error relative to $f(\mathbf{u})$, is the expectation of $f(\mathbf{u})$ under the posterior:

$$g(\mathbf{s}) = \int f(\mathbf{u}) \, \mathrm{P}(\mathbf{u}|\mathbf{s}) \, \mathrm{d}\mathbf{u} \tag{3}$$

Crucially, the expression for the posterior, given by Bayes' rule, is:

$$\mathrm{P}(\mathbf{u}|\mathbf{s}) \propto \mathrm{P}(\mathbf{s}|\mathbf{u}) \, \mathrm{P}(\mathbf{u}) \tag{4}$$

Note that while *Equations 3–4* do not reveal directly the specific *biophysical properties* the postsynaptic cell should have, they tell us phenomenologically what *signal integration properties* should result from its biophysical properties. In particular, they make it explicit that the optimal $g(\mathbf{s})$ depends fundamentally on two factors (*Figure 1B*, top):

1. the function that needs to be computed, $f(\mathbf{u})$, and
2. the statistics of presynaptic activities: $\mathrm{P}(\mathbf{u})$, the prior probability distribution characterizing the long-run statistics of multi-neural activity patterns in the presynaptic ensemble, and the likelihood $\mathrm{P}(\mathbf{s}|\mathbf{u})$, expressing the potentially probabilistic relationship between analog activities (e.g. somatic membrane potential trajectories) and emitted spike trains.

In the following, we show that the outcome of the integration of presynaptic spike trains in cortical neurons approximates very closely the optimal response, and that dendritic nonlinearities are crucial for achieving this near-optimality. For this, 1) we make an assumption about the computational function of the postsynaptic cell, $f(\mathbf{u})$ (*Figure 1B*, top right); 2) we constrain presynaptic statistics, $\mathrm{P}(\mathbf{u})$ and $\mathrm{P}(\mathbf{s}|\mathbf{u})$, by *in vivo* data about cortical population activity patterns (*Figure 1B*, top left); and with these 3) we compute the optimal response they jointly determine for various stimulation patterns (*Figure 1B*, bottom left and right).

## Optimal input integration is nonlinear

To specify our model, we considered the case when $f(\mathbf{u})$ itself is linear. Although networks with purely linear dynamics can perform non-trivial computations already (*Dayan and Abbott, 2001*; *Hennequin et al., 2014*), in the general case, we do expect $f(\mathbf{u})$ to be nonlinear, e.g. sigmoidal (*Hopfield, 1984*). Nevertheless, in typical electrophysiological experiments only a small fraction of the full dynamic range of a neuron's total input is stimulated (*Figure 1—figure supplement 2*), and so we approximate the computational function, $f(\mathbf{u})$, as being linear on this limited input range without loss of generality. (See *Figure 1—figure supplement 3* for the application of the theory to the case of nonlinear $f$.) Yet, as we show below, for physiologically realistic statistics of presynaptic activity patterns, the optimal response combines input spike trains in highly nonlinear ways even in the case of linear computation, predicting experimentally characterized nonlinearities in dendritic input integration. In particular, second- and higher-order prior presynaptic correlations, represented by

$P(\mathbf{u})$, will have a major role in determining the form of the corresponding optimal response. The likelihood, $P(\mathbf{s}|\mathbf{u})$, also influences the optimal response, but only in its quantitative details, as it does not involve correlations across neurons: each neuron's firing is independent from the others', given its own somatic membrane potential (Materials and methods).

Previous suggestions for how postsynaptic neurons achieve reliable computation despite the substantial ambiguity about the individual presynaptic activities relied on the linear averaging of inputs arriving from a sufficiently large pool of presynaptic neurons (*Dayan and Abbott, 2001*; *Pfister et al., 2010*). However, linear averaging is only guaranteed to produce the correct output, as dictated by *Equations 3-4*, if the activities of presynaptic neurons are statistically independent under the prior distribution, i.e. $P(\mathbf{u}) = \prod_i P(u_i)$ (Materials and methods). In contrast, the membrane potential (*Crochet et al., 2011*) and spiking (*Cohen and Kohn, 2011*) of cortical neural populations often show complex patterns of correlations, which include both 'spatial' (cross-correlations between different neurons) and temporal components (auto-correlations, i.e. the correlation of the activity of the same cell with itself at different moments in time). Thus, in this more general case, we expect the optimal response to involve a nonlinear integration of spike trains. While temporal correlations alone do not require nonlinear dendritic integration across synapses, only local nonlinearities within each synapse, as brought about e.g. by short term synaptic plasticity (*Pfister et al., 2010*), spatial correlations require the non-linear integration of spikes emitted by different presynaptic neurons.

To illustrate that presynaptic spatial correlations require nonlinear integration across synapses, we compared the best linear response to a given presynaptic spike pattern with the optimal response (*Equation 3*, as approximated by *Equation 23*) for two different input statistics that differed only in the correlations between the presynaptic cells but not in the activity dynamics or spiking of individual neurons (temporal correlations). To compute the postsynaptic response, we assumed that dendritic integration in the postsynaptic cell was linear but, in order to dissect the role of dendritic integration across synapses from the effects of nonlinearities in individual synapses, we allowed spikes from the same presynaptic neuron still to be integrated nonlinearly (*Pfister et al., 2010*). In the first case (*Figure 1C*), when the presynaptic neurons were independent, the best linear response was identical to the optimal response. However, if presynaptic neurons became correlated, the optimal response became nonlinear and the best linear response was unable to accurately follow the fluctuations in the input (*Figure 1D*).

Thus, inputs from presynaptic neurons whose activity tends to be correlated need to be nonlinearly integrated, while inputs from uncorrelated sources need to be integrated linearly. This could be naturally achieved in the same dendritic tree by clustering synapses of correlated inputs to efficiently engage dendritic nonlinearities, while distributing the synapses of uncorrelated inputs on different dendritic branches (*Larkum and Nevian, 2008*). Crucially, for correlated inputs it is also necessary that the dendritic nonlinearities have just the appropriate characteristics for the particular pattern of correlations in presynaptic activities.

## The form of the optimal nonlinearity depends on the statistics of presynaptic inputs

In order to systematically study the nonlinearities in the optimal response in the face of naturalistic input patterns, we derived and analyzed its behavior for a flexible class of richly structured, correlated inputs. Our statistical model for presynaptic activities, specifying the parametric forms of $P(\mathbf{u})$ and $P(\mathbf{s}|\mathbf{u})$ (Materials and methods and *Figure 2—figure supplement 1*), was able to generate a variety of multi-neural activity patterns resembling the statistical properties described in *in vitro* and *in vivo* multielectrode recordings of neuronal population activities (*Figure 2A and D* show two representative examples). Once we have specified the statistical model of presynaptic activities, it uniquely determined the optimal response to any given input pattern (*Equations 3–4*). Thus, we used the same statistical model in two fundamentally different ways: first, to generate "naturalistic" *in vivo*-like patterns of presynaptic membrane potential traces and spike trains; and second, to compute the optimal response pattern to *any* stimulation pattern, be it "naturalistic" or parametrically varying "artificial" as used in typical *in vitro* experiments.

The optimal response determined by this statistical model, for essentially any setting of parameters, was inherently nonlinear because the additional effect of a presynaptic spike depended on the pattern of spikes that had been previously received from the presynaptic population. Temporal correlations in the presynaptic population caused the optimal response to depend on the spiking

history of the same cell (*Pfister et al., 2010*), while crucially, the additional presence of spatial correlations introduced a dependency on the past spikes of other cells. Thus, the integrated effect of multiple spikes could not be computed as a simple linear sum of their individual effects in isolation. Specifically, a spike that was consistent with the information already gained from recent presynaptic spikes had only a small effect on the response (*Figure 2B*). Conversely, a spike that was unexpected based on the recent spiking history caused a larger change (*Figure 2E*).

As could be anticipated based on *Equations 3-4*, whether a spike counted as expected or unexpected relative to recently received spikes, and hence whether it had a small or large postsynaptic effect, depended on the long-run prior distribution of presynaptic activities, $P(\mathbf{u})$. As a result, the same pattern of presynaptic spikes led to qualitatively different responses under different prior distributions. In particular, sublinear integration was optimal when presynaptic activities exhibited Gaussian random walks and thus they did not contain statistical dependencies beyond second order correlations (*Figure 2A-C*), as seen in the retina and cortical cultures (*Schneidman et al., 2006*). This was because with the activities of presynaptic neurons being positively correlated, successive spikes conveyed progressively less information about the presynaptic signal resulting in sublinear integration (*Figure 2C*) and the strength of the sublinearity depended on the magnitude of correlations (*Ujfalussy and Lengyel, 2011*). In contrast, supralinear integration was optimal when the presynaptic population exhibited coordinated switches between distinct states associated with large differences in the activity levels compared to activity-fluctuations within each state (*Figure 2D–F*). These switches led to higher order statistical dependencies as seen in the cortex *in vivo*, either due to population-wide modulation by cortical state (*Gentet et al., 2010*; *Crochet et al., 2011*), or due to stimulus-driven activation of particular cell ensembles (*Harris et al., 2003*; *Ohiorhenuan et al., 2010*;

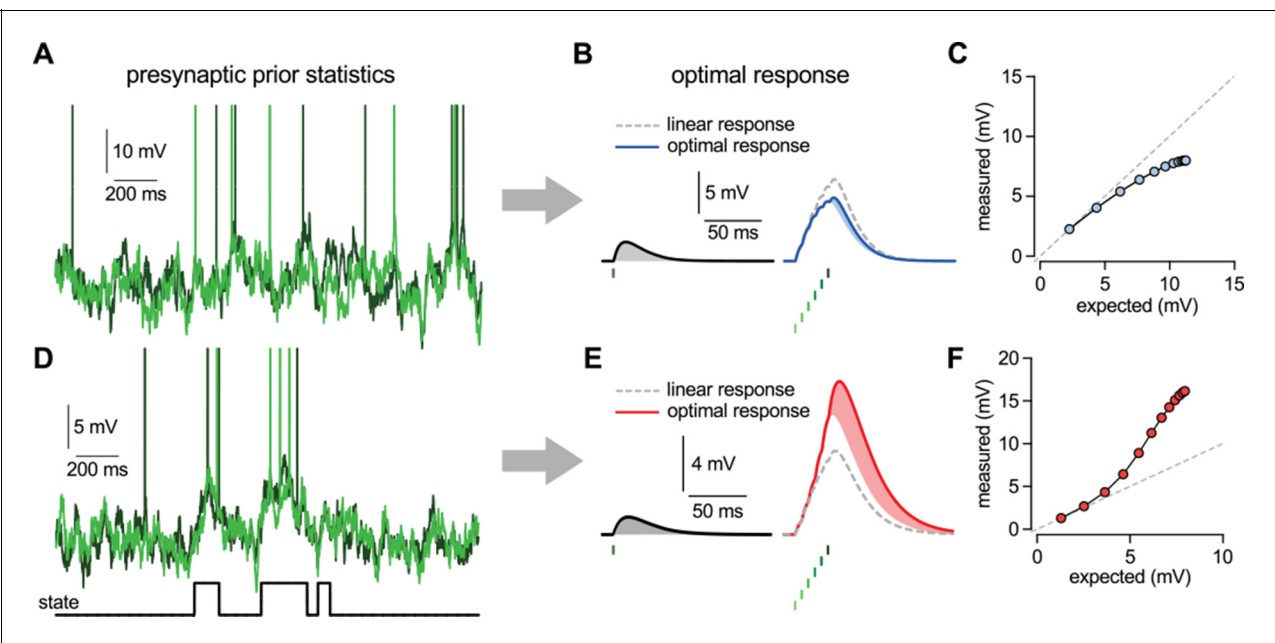

**Figure 2.** Nonlinearities in the optimal response. (A–C) Second order correlations between presynaptic neurons (A) imply sublinear integration (B–C). (A) Membrane potentials and spikes of two presynaptic neurons with correlated membrane potential fluctuations. (B) The optimal response (solid line) to a single spike (left) and to a train of six presynaptic spikes (right, green colors correspond to different presynaptic cells, two of which are shown in A) when the long-run statistics of presynaptic neurons are like those shown in (A). Shaded areas highlight how response magnitudes to a single spike from the same presynaptic neuron differ in the two cases: the response to the sixth spike in the train (right, light blue shading) is smaller than the response to a solitary spike (left, gray shading) implying sublinear integration. Dashed line shows linear response. (C) Response amplitudes for 1–12 input spikes versus linear expectations. (D–F) Same as (A–C) but for presynaptic neurons exhibiting synchronized switches between a quiescent and an active state, introducing higher order correlations between the neurons (D, bottom). In this case, the optimal response shows supralinear integration (E–F).

The following figure supplement is available for figure 2:

**Figure supplement 1.** Definition of the statistical model describing presynaptic activities and illustration of the inference process in the model.

*Miller et al., 2014*). In this case, while observing a few spikes was consistent with random membrane potential fluctuations within the quiescent state, thus only warranting a small response, further spikes suggested that the presynaptic population was in the active state now and thus the response should be larger, leading to supralinear integration (*Figure 2F*).

Note, that nonlinearities in the optimal postsynaptic response needed not simply compensate for the nonlinearities in the presynaptic spike generation process, as captured by $\mathrm{P(s|u)}$, but they critically depended on the presynaptic correlations, as captured by $\mathrm{P(u)}$. Indeed, in *Figures 1C, D* and *2A–F*, the same spiking nonlinearity was used and yet very different input integration was required depending on the form of the presynaptic statistics: linear integration for uncorrelated inputs (*Figure 1B*) and nonlinear integration for correlated inputs (*Figure 1C*), with sub- or supralinear integration being optimal depending on whether only second order (*Figure 2A–C*) or also higher order correlations were present in the presynaptic population (*Figure 2D–F*). Moreover, optimal input integration remained nonlinear even if the postsynaptic neuron computed a function of the presynaptic firing rates (rather than membrane potentials) which were linearly related to spikes (*Figure 1—figure supplement 1*).

## Nonlinear dendrites can approximate the optimal response

The nonlinear input integration seen in the optimal response strongly resembled dendritic nonlinearities. Indeed, the basic biophysical mechanisms present in dendrites naturally yield nonlinearities that are qualitatively similar to those of the optimal response: purely passive properties lead to sublinear integration (*Koch, 1999*), whereas locally generated dendritic spikes endow dendrites with strong supralinearities (*Nevian et al., 2007*; *Branco and Häusser, 2011*). However, the full mathematical implementation of the optimal response is excessively complex (Materials and methods) and thus, there is unlikely to be a one-to-one mapping between the variables necessary for implementing it and the biophysical quantities available in dendrites. Therefore, we sought to establish a formal proof that dendritic-like dynamics can implement, even if approximately, the optimal response. For this, we considered two limiting cases of our statistical model of presynaptic activities, $\mathrm{P(u)}$ and $\mathrm{P(s|u)}$, and compared the properties of the corresponding optimal response to a well-established simplified model of nonlinear dendritic integration, using a combination of analytical and numerical techniques.

First, we considered a limiting case in which the statistics of a large presynaptic population were strongly dominated by the simultaneous switching of presynaptic neurons between a quiescent and an active state (as shown in *Figure 2D*). In this limiting case we were able to show mathematically (see Materials and methods) that a simple, biophysically-motivated, canonical model of nonlinear dendritic integration (*Poirazi and Mel, 2001*) is able to produce responses that are near-identical to the optimal response for any sequence of presynaptic spikes (*Figure 3A*, see also *Figures 4C*). In this simple dendritic model (*Figure 3A*, inset; *Equations 24–25*), inputs within a branch are integrated linearly and the local dendritic response is then obtained by transforming this linear combination through a sigmoidal nonlinearity, which is a hallmark of supralinear behavior in dendrites (*Poirazi et al., 2003b*).

Second, we considered another limiting case in which the statistics of the presynaptic population were fully characterized by second-order correlations (as shown in *Figure 2A*). In this case, the same type of dendritic model, but with a sublinear input-output mapping, was able to approximate the optimal response very closely. Although a closed-form solution for the optimal nonlinear mapping could not be obtained in this case, it could be shown to be sublinear (Appendix), and was well approximated by a sigmoidal nonlinearity parametrized to be dominantly saturating (*Figures 3B* and *Figure 3—figure supplement 1*).

We also noted that it was the same type of sigmoidal nonlinearity which could implement supralinear and sublinear integration depending on the input regime (low background, synchronous spikes: supralinear; high background, asynchronous spikes: sublinear integration, compare *Figure 3A and B*, inset). This suggests that dendritic integration may adapt to systematic changes in presynaptic statistics, such as those brought about by transitioning between the desynchronized and synchronized states of the neocortex, or sharp waves and theta activity in the hippocampus, without having to change the parameters of its nonlinearity (*Borst et al., 2005*) (*Figure 3—figure supplement 2*). Indeed, *Gasparini and Magee (2006)* demonstrated that dendritic integration in hippocampal pyramidal cells was supralinear when inputs were highly synchronized (as they are during

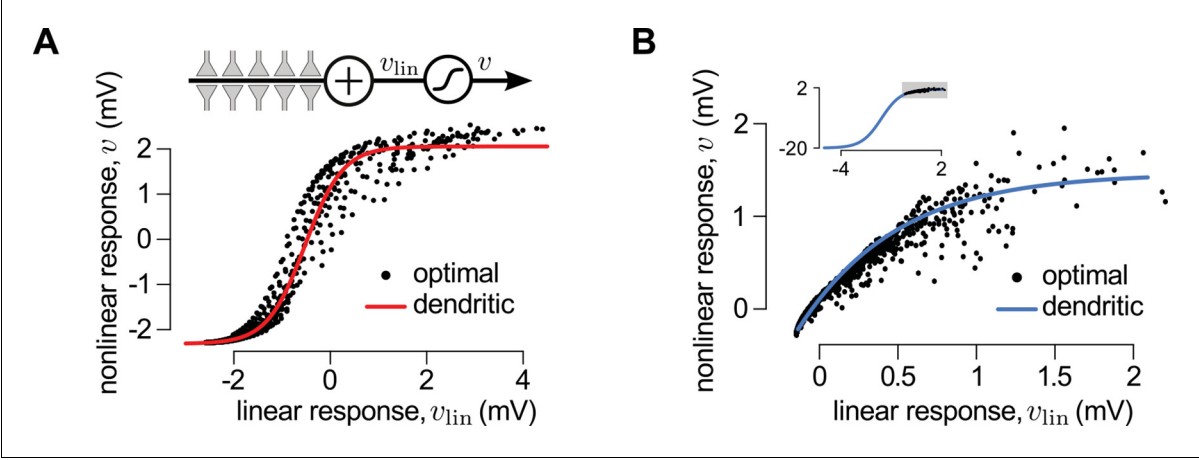

**Figure 3.** A canonical model of dendritic integration approximates the optimal response. (A) The optimal response (black) and the response of a canonical model of a dendritic branch, $v$ (inset), with a sigmoidal nonlinearity (red, *Equation 25*) as functions of the linearly integrated input, $v_{lin}$ (inset, *Equation 24*), when the presynaptic population exhibits synchronized switches between a quiescent and an active state, as in *Figure 2D*. Black dots show optimal vs. linear postsynaptic response sampled at regular 2.5 ms intervals during a 3 s-long simulation of the presynaptic spike trains. (B) Optimal response (black) approximated by the saturating part of the sigmoidal nonlinearity (blue) when the presynaptic population is fully characterized by second-order correlations, as in *Figure 2A*. Inset shows the same data on a larger scale to reveal the sigmoidal nature of the underlying nonlinearity (gray box indicates area shown in the main plot).

The following figure supplements are available for figure 3:

**Figure supplement 1.** Reducing the optimal response with second order correlations to a canonical model of dendritic integration.

**Figure supplement 2.** Adaptation without parameter change.

sharp waves), while integration was linear if the input was asynchronous (such as during theta activity).

## Nonlinear dendrites are computationally advantageous

While the foregoing analyzes proved that dendritic-like nonlinearities can closely approximate the optimal response in certain limiting cases, they do not address directly whether having such nonlinearities in input integration is crucial for attaining near-optimal computational performance for more realistic input statistics, or simpler forms of input integration could achieve similar computational power. To study this, we considered a scenario in which the presynaptic population consisted of four ensembles, such that neurons belonging to each ensemble underwent synchronized switches in their activity levels which were independent across the four ensembles, while there were also independent fluctuations in the activity of individual presynaptic neurons which were comparable in magnitude to those caused by these synchronized activity switches (*Figure 4A*). We then assessed the performance of four different variants of a simple dendritic model relative to that of the optimal response (*Figure 4B*): a model with linear dendrites and soma; a model in which only the soma was nonlinear, and two models in which nonlinearities resided in the dendrites with either random or clustered connectivity between the presynaptic assemblies and the dendritic branches.

We quantified the performance of each of the models based on how closely their output approximated the linear average of the analog presynaptic activities giving rise to the spike trains they were integrating (*Figure 4—figure supplement 1*, Materials and methods). For a fair comparison, we tuned the parameters of each variant of the dendritic model to obtain the best possible performance with it (*Figure 4C*). The model with nonlinear dendrites and clustered connectivity had near-optimal cross-validated performance (*Figure 4D*) while all other models performed significantly worse (n = 20 runs, t = 51, t = 35, t = 20, and $P<10^{-15}$, $P<10^{-15}$, $P<10^{-13}$; respectively from left to right as shown in *Figure 4D*). This remained true when we varied the number and firing rate of presynaptic neurons over a wide range, and under a diverse set of qualitatively different population-level

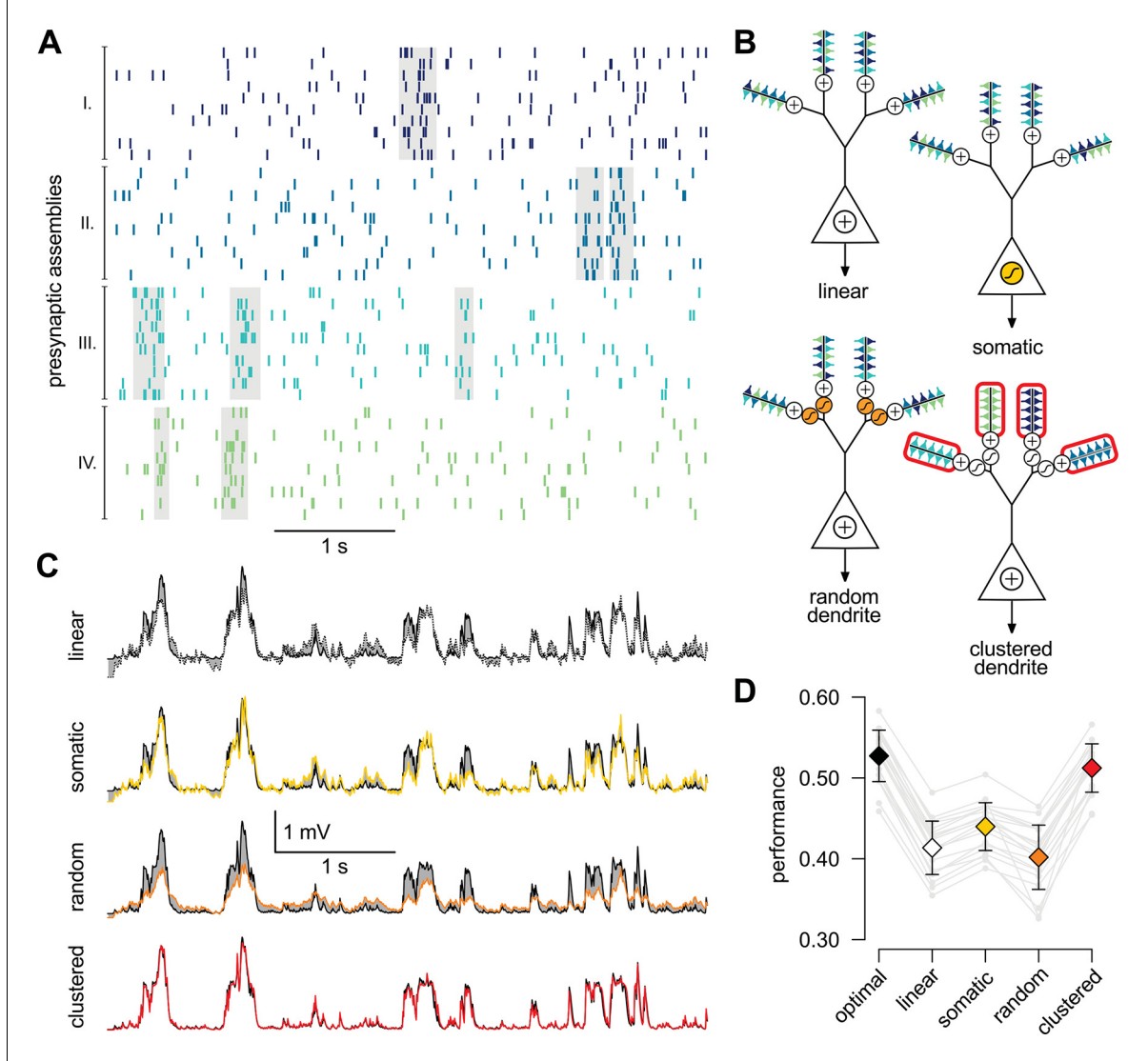

**Figure 4.** A simple nonlinear dendritic model closely approximates the optimal response for realistic input patterns. (A) Presynaptic spiking activity matching the statistics observed during hippocampal sharp waves. Spike trains (rows) belonging to four different assemblies are shown (colors), gray shading indicates assembly activations. (B) Different variants of the dendritic model, parts colored in yellow, orange, and red highlight the differences between successive variants (see text for details). (C) Estimating the mean of the presynaptic membrane potentials based on the observed spiking pattern (shown in A) by the optimal response (black) compared to the linear (dotted), somatic (yellow), random (orange) and clustered (red) models. (D) Performance of the four model variants compared to that of the optimal response. Gray lines show individual runs, squares show mean±s.d. Performance is normalized such that 0 is obtained by predicting only the time-average of the signal, and 1 means perfect prediction attainable only with infinitely high presynaptic rates (Materials and methods).

The following figure supplements are available for figure 4:

**Figure supplement 1.** Responses of different variants of the dendritic model compared to the true signal.

**Figure supplement 2.** Performance of different neuron models over a wide range of input statistics.

statistics, determining the dynamics of assembly switchings and within-assembly membrane potential correlations (*Figure 4—figure supplement 2*).

Taken together, these results demonstrate that the clustering of correlated inputs together with nonlinearities akin to those found in dendrites is necessary to achieve optimal estimation performance in the face of presynaptic correlations. However, in order to be tractable, our dendritic model

was mathematically simplified and, as a result, only qualitatively reproduced the nonlinearities of real dendrites. Thus, we directly compared experimentally recorded responses in dendrites to the optimal response.

## Nonlinear integration in cortical neurons is matched to their input statistics

A crucial prediction of our theory is that dendritic nonlinearities act to achieve near-optimal responses in a way that the form of the nonlinearity is specifically matched to the long-run statistics of the presynaptic population. We tested this prediction in experiments in which two different types of cortical pyramidal neurons, from layer 2/3 of the neocortex (*Figure 5A–E*) and from area CA3 of the hippocampus (*Figure 5F–J*), received patterned dendritic stimulation using two-photon glutamate uncaging, and compared their subthreshold somatic responses with the optimal responses predicted by the theory.

For generating our predictions of the optimal response in these two cell types, we fitted the parameters describing presynaptic statistics in our model, $P(\mathbf{u})$ and $P(\mathbf{s}|\mathbf{u})$, to the statistical patterns in the activity of their respective presynaptic populations. For neocortical pyramidal cells, we fitted *in vivo* data available on the membrane potential fluctuations of layer 2/3 pyramidal cell-pairs in the barrel cortex during quiet wakefulness (*Gentet et al., 2010*; *Crochet et al., 2011*) (NC, *Figure 5A*, *Table 2*). For hippocampal pyramidal cells, we fitted presynaptic statistics to membrane potential fluctuations (*Ylinen et al., 1995*; *English et al., 2014*) and to multineuron spiking patterns of hippocampal pyramidal cells during sharp wave activity (*Csicsvari et al., 1999*; *2000*) (HP, *Figure 5F*, *Table 3*). Due to the limitations of available hippocampal data sets, extracellular rather than intracellular data was used for fitting correlations. The motivation for our choice of the particular neocortical and hippocampal states used for fitting presynaptic statistics was two-fold. First, the general network state of the slice preparations in which we tested dendritic integration was likely most analogous to these states (A Gulyás, personal communication; see also *Karlocai et al., 2014*; *Schlingloff et al., 2014*), characterized by relatively suppressed neural excitability due to low levels of cholinergic modulation (*Harris and Thiele, 2011*; *Eggermann et al., 2014*). Second, the stimulation protocol used in our study (short bursts of synaptic stimuli following longer silent periods) was also most consistent with population activity during hippocampal sharp waves and quiet wakefulness in the cortex. In order to capture variability across the cells we recorded from, the parameters related to postsynaptic dendritic filtering (amplitude and decay of the response to a single stimulation, and the size of the dendritic subunit, *Figure 5—figure supplement 1B-C*) were tuned for the individual dendrites. Importantly, the parameters describing presynaptic statistics were fitted without regard to our dendritic experimental data, thus allowing a strong test of our predictions about dendritic integration (see Materials and methods).

We found that the non-linear integration of individual spike patterns in cortical neurons was remarkably well fit by the optimal response when it was tuned to the correct presynaptic statistics (*Figure 5C,H*). The systematic dependence of response amplitudes on the inter-stimulus interval (ISI) in individual cells (*Figure 5D,I*) was also well predicted by the optimal response. We quantified the quality of match between the predicted and experimentally recorded time course of responses across a population of n = 6 (neocortex) and n = 6 (hippocampus) dendrites under a range of conditions varying ISI or the number of stimuli, and found that the precision of our predictions was not statistically different from that expected from the inherent variability of responses in individual dendrites (*Figure 5E,J*; neocortex: t = 0.2, P = 0.85; hippocampus: t = 1.85, P = 0.12). In contrast, when the optimal response was tuned to unrealistic presynaptic statistics characterized purely by second-order correlations (cor2), or by a lack of any correlations implying statistically independent presynaptic firing (ind), the quality of fits became significantly worse (*Figure 5E,J*; neocortex: t = −4.6, P = 0.006 for cor2, and t = −4.9, P = 0.004 for ind; hippocampus: t = −4, P = 0.01 for cor2, and t = −4.9, P = 0.004 for ind).

Moreover, using realistic presynaptic statistics, but matching hippocampal rather than neocortical activities, also resulted in significantly worse fits for neocortical responses (*Figures 5E*; t= −3.6, P = 0.02). The converse was not observed in the case of hippocampal neurons (*Figures 5J*; t= 0.43, P = 0.68). This might be because hippocampal neurons also receive neocortical inputs (albeit on their apical not basal dendrites) that show similar population activity patterns to the ones we matched here for the neocortical cells (*Isomura et al., 2006*), while the primary sensory cortical

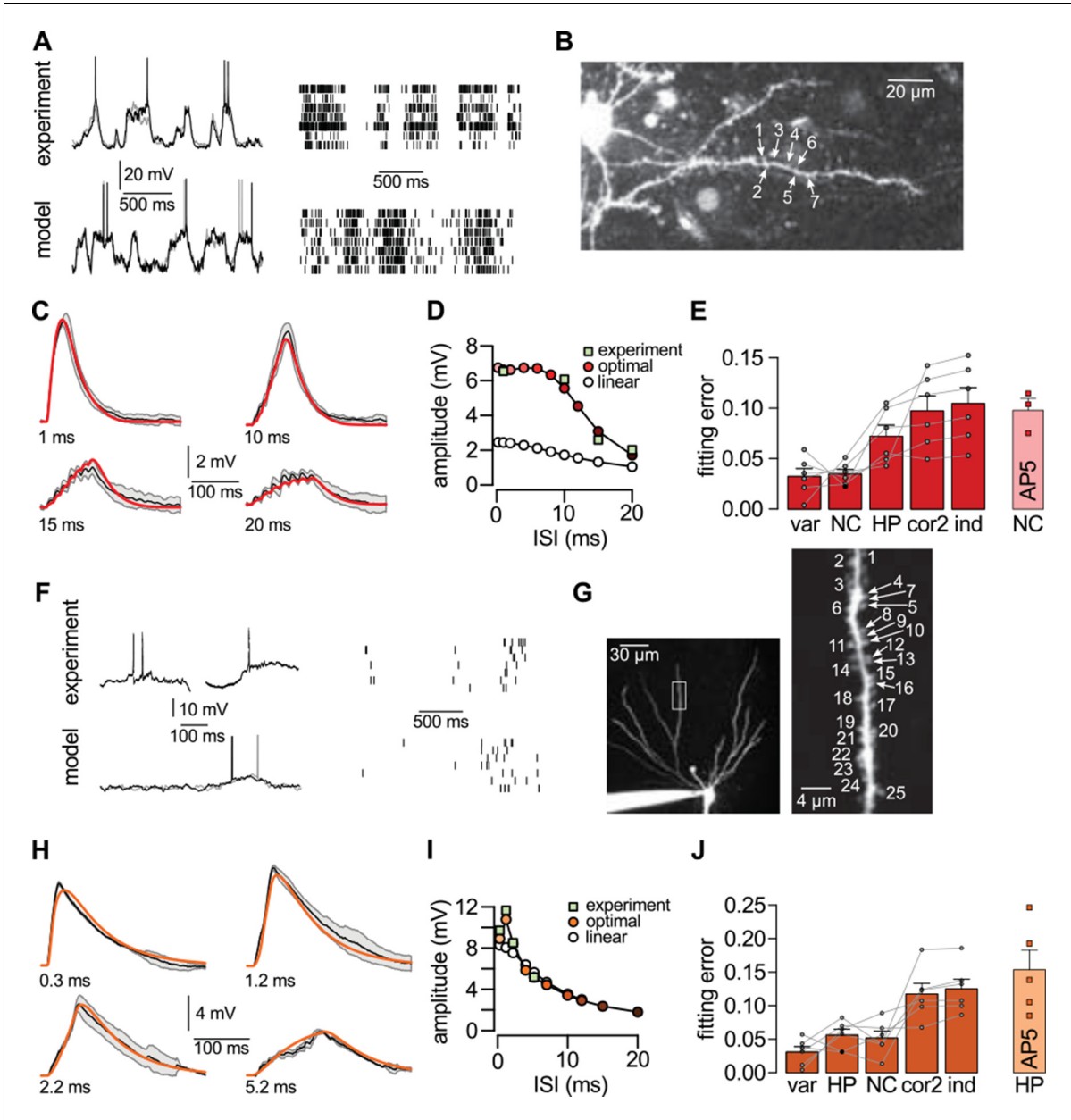

**Figure 5.** Nonlinear dendritic integration is matched to presynaptic input statistics. (A) Sample membrane potential fluctuations (left, adapted from *Gentet et al., 2010*) and multineuron spiking patterns (right, adapted from *Ji and Wilson, 2007*) recorded from the neocortex (top), and matched in the model (bottom, see also *Tables 1–3*). (B) Two-photon image of a neocortical layer 2/3 pyramidal cell, numbers indicate individual dendritic spines stimulated in the experiment. (C) Responses to trains of seven stimuli using different inter-stimulus intervals (ISI, shown below traces) recorded in the cell shown in (B) (black; mean±s.d.) and predicted by the optimal response tuned to the presynaptic statistics shown in A (red). Parameters related to postsynaptic dendritic filtering were tuned for the specific dendrite ( *Figure 5—figure supplement 1B–C*). (D) Dependence of response amplitudes on ISI in the same dendrite shown in B-C (squares), and as predicted by the optimal response (filled circles) or linear integration (empty circles). (E) Average error of fitting dendritic recordings across all dendrites and conditions using the optimal response tuned to different presynaptic statistics (NC, HP, cor2, ind; see text for details) compared to within-data variability (var). Gray lines show individual dendrites. Rightmost bar (NC-AP5) shows fit using NC presynaptic statistics to responses obtained after pharmacological blockade of NMDA receptor activation. (F–J) Same as (A–E) for presynaptic patterns characterized by hippocampal sharp waves (F) and recordings from hippocampal CA3 pyramidal cells (H–J) when stimulating synapses on its basal dendrites (G). *In vivo* data in (F) was adapted from (*Ylinen et al., 1995*) (left, membrane potential traces, not simultaneously recorded) and (*O'Neill et al., 2006*) (right, multineuron spike trains). Error bars show s.e.m.

The following figure supplements are available for figure 5:

**Figure supplement 1.** Best fit parameters for fitting dendritic responses.

*Figure 5 continued on next page*

*Figure 5 continued*

**Figure supplement 2.** Dendritic integration in cerebellar stellate cells is not predicted by cortical presynaptic statistics.

pyramidal cells we recorded from do not receive direct input from the hippocampus. Nevertheless, when we analyzed the quality of fit between our predictions and recorded responses in hippocampal and neocortical data together, we found a small, but significant interaction between the source of the input statistics (neocortex or hippocampus) and the location of the postsynaptic neurons (ANOVA F = 5, P < 0.05). This suggests that dendritic nonlinearities in cortical pyramidal neurons are specifically tuned to the dynamics of their presynaptic cortical ensembles. Furthermore, the blockade of NMDA receptor activation by AP5 resulted in dendritic responses that afforded substantially poorer fits by the model, even after refitting the postsynaptic parameters (*Figure 5E,J*, AP5). This indicated that the fine tuning of dendritic nonlinearities to input statistics relied on the action of NMDA receptors.

As dendrites in both of our cortical cell types integrated inputs supralinearly, as a further control, we analyzed similar data available from cerebellar stellate cell dendrites, which are known to integrate their inputs sublinearly (*Abrahamsson et al., 2012*) (*Figure 5—figure supplement 2*). In this case, we fitted the statistics of individual presynaptic cells to those of cerebellar granule cells. The correlations between these cells are less known, but we found that assuming simple second-order correlations made the optimal response a close match to dendritic responses. In contrast, the hippocampal- or neocortical-like statistics that were crucial for matching responses in cortical dendrites (*Figure 5D,H*) resulted in a substantially poorer fit in this cerebellar cell type. This demonstrates a double dissociation in the matching of cortical and subcortical neuron types to cortical and non-cortical input statistics.

**Table 1.** Parameters used in *Figures 1–5* of the main paper (see also *Tables 2–3*). $\Omega_-$ ($\Omega_+$) is the rate of switching from the active to the quiescent (from the quiescent to the active) state. The resting potential corresponding to the active and quiescent states is $\overline{u}$ and $-\overline{u}$, respectively. $\overline{\Sigma}_{ii}$ ($\overline{\Sigma}_{ij}$) is the posterior variance (covariance) of the presynaptic membrane potential fluctuations in a given state where $\overline{\Sigma} = \frac{Q\tau}{2}$. $\tau_{\text{refr}}$ is the length of the refractory period and $p_{\text{rel}}$ is the baseline transmission probability in these synapses (13, 49).

| parameter | unit | Figure 1 | Figure 2 | | Figure 3 | | Figure 4 | Figure 5 | | | |
|---|---|---|---|---|---|---|---|---|---|---|---|
| | | B,C | A,B | C,D | A | B | A-D | ind | cor2 | NC | HP |
| $\Omega_-$ | **Hz** | 10 | – | 10 | 10 | – | 10 | – | – | 10 | 10 |
| $\Omega_+$ | **Hz** | 10 | – | 0.27 | 0.67 | – | 0.67 | – | – | 4 | 0.027 |
| $\overline{u}$ | **mV** | 2.4 | 0 | 2.3 | 2.3 | 0 | 2.3 | 0 | 0 | 10 | 2.3 |
| $\tau$ | **ms** | 20 | 20 | 20 | 20 | 20 | 20 | 20 | 20 | 20 | 20 |
| $\overline{\Sigma}_{ii}$ | **mV$^2$** | 1 | 16 | 4 | 1 | 1 | 1 | 1 | 1 | 10 | 1 |
| $\overline{\Sigma}_{ij}\ \forall i \neq j$ | **mV$^2$** | 0 | 0.5 | 0.5 | 0 | 0.5 | 0 | 0 | * | 5 | 0.5 |
| $g$ | **Hz** | 1 | 1 | 1 | 5.3 | 0.5 | 5 | 0.5 | 0.5 | 1 | 2 |
| $\beta$ | **mV$^{-1}$** | 1 | 0.4 | 0.4 | 0.5 | 0.4 | 0.4 | 1 | 2 | 0.1 | 0.6 |
| $\tau_{\text{refr}}$ | **ms** | 3 | 3 | 3 | 1 | 3 | 1 | 3 | 3 | 3 | 3 |
| $p_{\text{rel}}$ | **–** | 1 | 1 | 1 | 1 | 1 | 1 | 1 | 1 | 0.5 | 0.2 |
| $N$ | **–** | 20 | 70 | 20 | 10 | 10 | 10 | +0[†] | +20[†] | * | * |
| $\tau_{\text{post}}$ | **ms** | 0 | 10 | 10 | 0 | 0 | 0 | * | * | * | * |
| $w_i$ | **–** | 1/N | 1/N | 1/N | 1/N | 1/N | 1/N | * | * | * | * |

*These parameters were fitted to experimentally recorded dendritic responses, see *Figure 5—figure supplement 1*. [†]The numbers 0 and 20 indicated here are in addition to the number of stimulated synaptic sites in the experiment. For the *ind* model, this number does not affect the predictions, for the *cor2* model its effects could phenomenologically be incorporated into which we chose to fit instead.

**Table 2.** Features of neocortical population activity during quiet wakefulness. Parameters of the model are given in column NC of **Table 1**.

| | Data | Model (NC) | Reference |
|---|---|---|---|
| duration of active states | 130 ms | 100 ms | *Gentet et al. (2010)* |
| duration of quiescent states | 200 ms | 250 ms | *Gentet et al. (2010)* |
| $r^+$, firing rate during active states | 2.5 Hz | 2.86 Hz | *Gentet et al. (2010)* |
| $r^-$, firing rate during quiescent states | $\leq 1/3$ Hz | 0.39 Hz | *Gentet et al. (2010)* |
| $2\bar{u}$, depolarisation during active states | 20 mV | 20 mV | *Gentet et al. (2010)* |
| time constant | 20 ms | 20 ms | *Poulet and Petersen (2008)* |

## Discussion

We established a functional link between the statistics of the synaptic inputs impinging on the dendritic tree of a neuron and the way those inputs are integrated within the dendritic tree. We first demonstrated that efficient computation in spiking circuits requires nonlinear input integration if the activities of the neurons are correlated and that the structure of the presynaptic correlations determines the form of the optimal input integration. Second, we showed that the optimal response can be efficiently approximated by a canonical biophysically-motivated model of dendritic signal processing both for linearly correlated inputs and for cell-assembly dynamics. Third, we found that nonlinear dendrites with synaptic clustering carry significant benefits for decoding richly structured presynaptic spike trains. Finally, in vitro measurements of dendritic integration in two different types of cortical pyramidal neurons yielded postsynaptic responses that closely matched those predicted to be optimal given the *in vivo* input statistics of those particular cell types. These results suggest that nonlinear dendrites are essential to decode complex spatio-temporal spike patterns and thus play an important role in network-level computations in neural circuits.

### Biophysical substrate

The central insight of our theory is the relationship between presynaptic statistics and postsynaptic response, formalized as the optimal response. The optimal response can be expressed as a set of nonlinear differential equations that requires storing and continuously updating $\sim N^2$ variables within the dendritic tree, where $N$ is the number of synapses (Materials and methods). Thus, it is unlikely to be implemented by the postsynaptic neuron as such. Consequently, to demonstrate the biophysical feasibility of our theory, we derived a simple approximation to the optimal response that performs about equally well using just a few postsynaptic variables and that corresponds to a canonical descriptive model of dendritic integration (*Poirazi et al., 2003*; *Poirazi et al., 2003b*).

**Table 3.** Features of hippocampal population activity during sharp wave-ripple states. Parameters of the model are given in column HP of **Table 1** A recent intracellular study (*English et al., 2014*) recording from CA1 neurons in awake mice found parameters similar to our previous estimates. Using the parameters found in that study – $r^+ = 12.8$ Hz, $r^- = 2.85$ Hz (**Table 1** of *English et al., 2014*), $2\bar{u} = 5$ mV and $\bar{\Sigma}_{ii} = 4$ mV$^2$ (**Figure 3A** of *English et al., 2014*) yielding $g = 5$ Hz and $\beta = 0.3$ mV$^{-1}$ – did not influence our results (not shown).

| | Data | Model (HP) | Reference |
|---|---|---|---|
| activation rate of an ensemble | $\ll 0.25$ Hz | 0.027 Hz | *Grosmark et al. (2012)*; *Pfeiffer and Foster (2013)* |
| duration of SPWs | 105 ms | 100 ms | *Csicsvari et al. (2000)* |
| $r^+$, firing rate during SPW | 10 Hz | 9.5 Hz | *Csicsvari et al. (2000)* |
| $r^-$, firing rate between SPWs | 0.5 Hz | 0.6 Hz | *Grosmark et al. (2012)*; *Csicsvari et al. (2000)* |
| $2\bar{u}$, depolarisation during SPWs | 0–10 mV | 4.6 mV | *Ylinen et al. (1995)* |
| time constant | 8–22 ms | 20 ms | *Epsztein et al. (2011)* |

We found that simple second order correlations between presynaptic neurons imply sublinear integration which can be implemented by the saturating nonlinearity characteristic of passive dendrites. Conversely, the biophysical substrate for the type of supralinear integration that was optimal for state-switching dynamics likely involves NMDA receptors because the particular dendritic nonlinearites observed in the cortical cells in which we tested our theory are known to be mediated primarily through NMDA receptor activation (*Branco et al., 2010*; *Makara and Magee, 2013*; *Major et al., 2013*). Indeed, we found that pharmacological inactivation of NMDA receptors abolished the precise match between dendritic integration and presynaptic statistics in these neurons (*Figure 5*). Moreover, the local plateau potentials generated by NMDA currents have been shown to have graded response durations (*Major et al., 2008*), and the resulting nonlinearities could be continuously tuned between weaker and stronger forms (boosting and bistability, *Major et al., 2013*). These properties make NMDA receptor mediated dendritic nonlinearities ideally suited for being matched to presynaptic statistics, as the optimal response involves sustained dendritic depolarisations of varying duration (*Figure 4*) that depend parametrically on those statistics.

## Input statistics and clustering

A central prediction of our theory that awaits confirmation is the existence of a tight relationship between the structure of correlations in the activity of presynaptic cells and the morphological clustering of their synapses on the postsynaptic dendrite. This is because our theory requires nonlinear integration of inputs from neurons with statistically dependent activity, while spikes from independent neurons need to be integrated linearly. Biophysical considerations suggest (*Koch, 1999*) and experimental data supports (*Polsky et al., 2004*; *Losonczy and Magee, 2006*) that, when synchronous, nearby inputs on the same dendritic branch are summed nonlinearly, whereas widely separated inputs are combined linearly. Consequently, our theory predicts that the correlation structure of the inputs will be mapped to the dendritic tree in a way that presynaptic neurons with strongly correlated activities target nearby locations while independent neurons innervate distinct dendritic subunits.

According to our theory, the kind of correlation relevant for determining synaptic clustering is the 'marginal' correlations between the membrane potentials of presynaptic neurons. Marginal correlations include both signal and noise correlations (*Averbeck et al., 2006*) and thus can reach substantial magnitudes even when noise correlations alone are small, as e.g. during desynchronized cortical states (*Renart et al., 2010*), especially for neurons with overlapping receptive fields (*Froudarakis et al., 2014*), and when measured between the membrane potentials of neurons rather than their spike counts (*Dorn and Ringach, 2003*; *de la Rocha et al., 2007*; *Poulet and Petersen, 2008*).

At the level of different dendritic regions, the segregation of different input pathways along the dendritic tree of hippocampal neurons supports this prediction (*Witter et al., 1989*; *Druckmann et al., 2014*). At the level of individual synapses, the degree and the existence of clustering among inputs showing correlated activity is currently debated. High resolution imaging revealed subcellular topography of sensory inputs in the tadpole visual system (*Bollmann and Engert, 2009*), clustered patterns of axonal activity in the parallel fibres that provide input to cerebellar Purkinje cells (*Wilms and Häusser, 2015*), and experience-driven synaptic clustering in the barn owl auditory localization pathway (*McBride et al., 2008*). Furthermore, it has been demonstrated that neighboring synapses are more likely to be coactive than synapses that are further away from each other in developing hippocampal pyramidal cells (*Kleindienst et al., 2011*) as well as in hippocampal cultures and *in vivo* in the barrel cortex during spontaneous activity (*Takahashi et al., 2012*). These results thus suggest clustering of correlated inputs.

In contrast, an interspersion of differently tuned orientation-, frequency- or whisker-specific synaptic inputs on the same dendritic segments was found in the mouse visual, auditory or somatosensory cortex, respectively, thus challenging the prevalence of synaptic clustering (*Jia et al., 2010*; *Chen et al., 2011*; *Varga et al., 2011*). However, in all these studies the stimuli used were non-naturalistic and varied along a single stimulus dimension only (direction of drifting gratings, pitch of pure tones, or the identity of the single whisker being stimulated), which may account for the apparent lack of clustering. In particular, our theory predicts clustering based on the long-term statistical dependencies between the responses of the presynaptic neurons for naturalistic inputs, which can be quite poorly predicted from their tuning properties for single stimulus dimensions (*Harris et al.,*

*2003*; *Fiser et al., 2004*). In contrast, the statistical dependencies relevant for our theory are well represented by those found during spontaneous activity (*Berkes et al., 2011*). Indeed, studies finding evidence in favor of synaptic clustering analyzed the structure of synaptic input to dendritic branches during spontaneous network activity (*McBride et al., 2008*; *Kleindienst et al., 2011*; *Makino and Malinow, 2011*; *Takahashi et al., 2012*). Thus, presynaptic correlations for naturalistic stimulus sets may be predictive of synaptic clustering and providing more direct evidence for or against such clustering will offer a crucial test of our theory.

## Linear vs. nonlinear postsynaptic computations

Although, in general, we expect single-neuron computations to be nonlinear (*Zador, 2000*), and our theory indeed applies to nonlinear computations (*Figure 1—figure supplement 3*), we assumed the postsynaptic computation to be linear for matching experimental data. This choice was justified by two reasons. First, it is difficult to determine, without making strong prior assumptions, what kind of nonlinear function the neuron actually computes; and so the choice of any particular such function would have been arbitrary. Note that even in relatively well-characterized cortical areas (such as the visual cortex) it is unknown how much of the computationally relevant output of individual neurons (such as orientation or direction selectivity) is brought about by specific nonlinearities in the input-output transformations of these neurons, or by multiple steps of feed-forward and recurrent processing carried out at various stages of the visual pathway between the retina and those neurons. Moreover, in some cases, even networks with linear dynamics can provide a remarkably close fit to experimentally observed cortical population dynamics (*Hennequin et al., 2014*). This issue may be best addressed in systems that are more specialized than the cortex so that there are well-supported hypotheses about the particular nonlinear computations individual neurons need to perform, such as the fly visual system (*Single and Borst, 1998*) or the mammalian and avian auditory brain stem (*Agmon-Snir et al., 1998*). In order to test our theory in these systems, *in vivo* multineural data will need to be collected from the afferent brain areas, preferably in the unanesthetized animal, for characterising the relevant statistical properties of the presynaptic population to which dendritic nonlinearities are adapted according to our prediction.

Second, any nonlinear function can be approximated to high precision by a linear function over a sufficiently limited input range. Currently available experimental techniques for systematically probing dendritic nonlinearities, including those used in our study, only provide data over such a very limited range (~0.1% of the number of excitatory inputs impinging a neuron, *Megías et al., 2001*). Inputs in this small range do not sufficiently engage global nonlinearities brought about by active somatic conductances or global events such as $Ca^{2+}$ spikes. Thus, we could assume linear computation over this range without loss of generality (*Figure 1—figure supplement 2*). In fact, from this perspective, it is a non-trivial phenomenon to account for on its own right that stimulating such a small fraction of inputs already leads to observable nonlinearities in the postsynaptic dendrite. By defining the computation to be linear, we could demonstrate that such strong dendritic nonlinearities arise naturally in our theory, entirely due to the correlations in the prior input statistics, thus providing a functional account for this remarkable phenomenon.

Once patterned dendritic stimulation over a broader and more realistic range of inputs becomes feasible, our theory will provide a principled method for dissecting the roles of presynaptic correlations vs. genuine nonlinear computations in shaping dendritic nonlinearities. A sufficiently rich set of such data will allow the fitting of presynaptic parameters, as we did here, followed by fitting postsynaptic transfer functions to dendritic responses without having to make strong prior assumptions about their (linear) nature.

## Analog communication, stochastic synaptic transmission and short-term synaptic plasticity

Our formalism was based on the assumption that cortical neurons only influence each other's membrane potentials via the action potentials they emit. While there exist other, more analog forms of communication, such as the modulation of the effects of action potentials by subthreshold potentials (*Clark and Häusser, 2006*), the propagation of voltage signals through gap junctions (*Vervaeke et al., 2012*), and ephaptic interactions between nearby cells (*Anastassiou et al., 2011*), these either require slow membrane potential dynamics, small distances between interacting cells,

or large degrees of population synchrony, and are thus generally believed to have a supplementary role beside spike-based communication (*Sengupta et al., 2014*). Note that our theory is self-consistent even though it considers spiking only in the presynaptic population and not in the postsynaptic neuron. This is because we assumed that the computationally relevant mapping is that between the membrane potentials of the presynaptic neurons and the postsynaptic cell (*Figure 1A*, *Equation 1*), and so, by induction, the spikes of the postsynaptic neuron will effect the mapping from its membrane potential to those of its postsynaptic partners.

We also assumed that presynaptic spikes deterministically and uniformly impact the postsynaptic response, and thus apparently neglected the stochasticity in synaptic transmission, and in particular systematic variations in synaptic efficacy brought about by short-term synaptic plasticity. Nevertheless, these presynaptic features are compatible with our theory. The stochasticity of synaptic transmission, due to a baseline level of synaptic failures, is straight-forward to incorporate in the model by reducing the effective presynaptic firing rate, which can thus be interpreted as a 'transmission rate' instead. In fact, we have already done this while matching hippocampal and neocortical presynaptic statistics (*Table 1*).

Short-term synaptic plasticity, resulting in dynamical changes in synaptic efficacy as a function of the recent spiking history of the presynaptic neuron, is not only a constraint in our framework, but as we have shown in related work, it can act itself as an optimal estimator of the membrane potentials of individual presynaptic neurons (*Pfister et al., 2010*). Thus, the effects of short-term plasticity can be regarded as a special case of what can be expected from our optimal response: when presynaptic neurons are statistically independent, spikes arriving at different synapses are integrated linearly, and local nonlinearities acting at individual synapses suffice (*Figure 1C*, see also Materials and methods). However, the importance of nonlinear interactions between inputs from different presynaptic neurons, brought about by dendritic nonlinearities, rapidly increases with the magnitude of presynaptic correlations, especially in large populations (*Figure 1D*, see also *Ujfalussy et al., 2011*).

These considerations suggest that short-term synaptic plasticity and dendritic nonlinearities have complementary roles in tuning the postsynaptic response to the statistics of the presynaptic population along the orthogonal dimensions of time and space. The former is useful in the face of temporal correlations private to individual presynaptic neurons (auto-correlations, e.g., brought about by spike frequency adaptation, *Pfister and Surace, 2014*), while the latter is matched to spatio-temporal correlation patterns present across the presynaptic population.

## Inhibitory neurons

We focused on the nonlinear integration of excitatory inputs in the dendritic tree of cortical neurons that have been extensively studied and described over the past decades, giving rise to a strong body of converging evidence as to their characteristics and mechanisms (*Spruston, 2008*). Recent work studying the nonlinear interaction between inhibitory and excitatory inputs in active dendrites (*Gidon and Segev, 2012*; *Jadi et al., 2012*; *Müller et al., 2012*; *Wilson et al., 2012*; *Lovett-Barron et al., 2012*) demonstrated that local inhibition has a powerful control over the excitability of the dendritic tree.

However, it is not yet clear whether these inhibitory inputs are directly involved in the computation performed by the circuit, just as excitatory neurons but with negative signs (*Koch et al., 1982*), or, alternatively, they may have a more ancillary role in supporting computations carried out primarily by excitatory neurons (*Vogels et al., 2011*).

Our theory can be extended to include both possibilities, by allowing inhibitory inputs to contribute to the computational function, $f(\mathbf{u})$, with negative weights, or by considering them as providing auxiliary information about the common state of the excitatory presynaptic ensemble, especially when this state is in the more suppressed regime. Indeed, our preliminary results suggest that such an extension of our theory (*Ujfalussy and Lengyel, 2013*) successfully accounts for the interaction of (excitatory) Schaffer collateral inputs with the feedforward inhibitory effects of the temporo-ammonic pathway (*Remondes and Schuman, 2002*), likely mediated by interneurons delivering dendritic inhibition (*Dvorak-Carbone and Schuman, 1999*).

In the present paper we focused on dendritic integration in pyramidal neurons because dendritic nonlinearities have traditionally been more extensively characterized in this cell type, but our theory equally applies to synaptic integration in other types of neurons, including inhibitory interneurons. Therefore, our theory predicts a qualitative similarity of dendritic integration in different neuron

types (i.e. interneurons versus principal cells) when they receive inputs from overlapping presynaptic populations. Indeed, it has been recently found that inhibitory interneurons can exhibit dendritic NMDA spikes under certain experimental circumstances (*Katona et al., 2011*; *Chiovini et al., 2014*) in addition to standard sublinear integration. The differences between dendritic integration in excitatory and inhibitory neurons could be attributed to their different computational function, $f(\mathbf{u})$, or differences in the specific presynaptic populations innervating them.

### Adaptation of dendritic nonlinearities to presynaptic statistics

According to our theory, the optimal response depends on prior information about the input statistics. Consequently, for dendritic processing to approximate the optimal response, this prior information needs to be implicitly encoded in the form of the particular nonlinearity a dendrite expresses. Therefore, our theory predicts an ongoing adaptation of dendritic nonlinearities to presynaptic firing statistics across several time-scales.

First, there is a simple yet potent mechanism implicit in our theory that can ensure that a match of dendritic integration to presynaptic statistics is maintained as those statistics are changing over time. This is based on the observation that essentially instantaneous, albeit probably incomplete, adaptation can occur even without specific changes in the integrative properties of dendrites per se, simply due to the fact that a critical level of input synchrony is required to elicit dendritic spikes, and so the same sigmoid-looking dendritic transfer function can be used as superlinear, linear, or sublinear, depending on which part of its input range is being used (*Figure 3—figure supplement 2*).

Second, to match the more specific modulation of the statistics of presynaptic activities by global cortical states (*Crochet et al., 2011*; *Mizuseki and Buzsaki, 2014*), dendritic integration may also be modulated by these states. As different cortical states are typically accompanied by changes in the neuromodulatory milieu (*Hasselmo, 1995*; *Harris and Thiele, 2011*), neuromodulators may be the ideal substrates to ensure that dendritic integration also changes according to the current cortical activity mode. This may provide a functional account of changes in the excitability of the dendritic tree as dynamically regulated by acetylcholine and monoamines (*Sjöström et al., 2008*).

Third, experience-dependent synaptic plasticity can gradually change the statistics of the presynaptic population activity implying that the optimal form of input integration should also change as a function of experience. We propose that branch-specific forms of plasticity of dendritic excitability (*Losonczy et al., 2008*; *Makara et al., 2009*; *Müller et al., 2012*) may have a functional role in enabling dendrites to adjust the form of input integration to such slowly developing and long-lasting changes in the statistics of their inputs.

Finally, whether inputs from two presynaptic cells are integrated linearly or nonlinearly in a dendrite depends critically on the distance between their synapses within the dendritic tree (*Polsky et al., 2004*; *Losonczy and Magee, 2006*). Our theory requires nonlinear integration of inputs from neurons with statistically dependent activity, predicting a mapping of presynaptic correlations on the postsynaptic dendritic tree. Local electrical and biochemical signals can drive synaptic plasticity (*Larkum and Nevian, 2008*; *Govindarajan et al., 2011*; *Winnubst et al., 2015*) and rewiring (*DeBello, 2008*) leading to synaptic clustering of correlated inputs along the dendritic tree (*Kleindienst et al., 2011*; *Takahashi et al., 2012*).

A combination of all these mechanisms may be crucial for achieving and dynamically maintaining, at the level of individual neurons, a detailed matching of dendritic nonlinearities to presynaptic statistics. Thus, our theory provides a novel framework for studying a range of phenomena regarding the dynamical regulation of dendritic nonlinearities from the perspective of circuit-level computations.

## Materials and methods

Source code for reproducing the analyses and simulations presented in the paper as well as the experimental data we used for testing our predictions are available online (https://bitbucket.org/bbu20/optimdendr).

### Computing the optimal response

In order to study the optimal form of input integration with realistic input statistics, we need to make two important assumptions. First, we need to assume a particular algebraic form for the

computation that a neuron performs. Second, we need to define what the relevant presynaptic statistics are, that is, the membrane potential and spiking dynamics of the presynaptic population under naturalistic *in vivo* circumstances. Given these two assumptions, the theory uniquely defines the optimal response of a neuron to any input pattern. The optimal response has qualitatively similar properties whether computations are defined as mappings between pre- or postsynaptic membrane potentials or firing rates (*Figure 1—figure supplement 1*).

Throughout the paper the term *input* refers to the spatio-temporal spiking pattern impinging the neuron while the *response* of a neuron refers to its (subthreshold) somatic membrane potential (or firing rate, see below). All parameters used in the paper are given in *Table 1* or in the caption of the corresponding Figure.

## Postsynaptic computation

We assumed the postsynaptic computation to be linear, i.e. the dynamics of the postsynaptic membrane potential $v(t)$ evolves according to a weighted sum of the presynaptic membrane potential values, $\mathbf{u}(t)$ (cf. *Equation 1*):

$$\tau_{\text{post}}\dot{v}(t) = -v(t) + \sum_{i=1}^{N} w_i u_i(t) \tag{5}$$

where $\tau_{\text{post}}$ is the time constant of the postsynaptic neuron, $N$ is the number of presynaptic neurons, and $w_i$ is the computational weight assigned to presynaptic neuron $i$. As the postsynaptic neuron cannot access presynaptic membrane potentials, $\mathbf{u}$, directly only the spikes the presynaptic cells emit, $\mathbf{s}$, (*Figure 1*, *Equation 2*), the optimal response (that minimizes mean squared error) is the expectation of *Equation 5* under the posterior distribution of the presynaptic membrane potential at time $t$, $\mathbf{u}(t)$ given the history of presynaptic spiking up to that time, $\mathbf{s}(0:t)$ (cf. *Equation 3*):

$$\tau_{\text{post}}\dot{\tilde{v}}(t) = -\tilde{v}(t) + \int P(\mathbf{u}(t)|\mathbf{s}(0:t)) \left[ \sum_{i=1}^{N} w_i u_i(t) \right] d\mathbf{u}(t) \tag{6}$$

Throughout the paper we call the output of *Equation 6* the *optimal response* and compare its behavior to input integration in the dendrites of cortical pyramidal cells.

*Table 1* shows the values of the parameters in *Equation 5* ($N$, $\tau_{\text{post}}$, and $w_i$) used in the simulations. In short, to illustrate the contributions of inference to *Equation 6* (the term including the integral), we used $\tau_{\text{post}} = 0$ in *Figures 1*, *3* and *4* as well as in all Supplemental Figures, unless otherwise stated. We used $\tau_{\text{post}} = 10$ ms in *Figure 2* to aid comparison with experimental data and fitted $\tau_{\text{post}}$ to data for *Figure 5*. Throughout the paper we used $w_i = 1/N$, except in *Figure 5* where we fit $N$ and $w_i = w$ jointly to the data.

## Presynaptic statistics

Computing the posterior, $P\big(\mathbf{u}(t)|\mathbf{s}(0:t)\big)$ in *Equation 6* requires a model for the joint membrane potential and spiking statistics of the presynaptic population, $P(\mathbf{u},\mathbf{s})$ (see also *Equation 4*). For mathematical convenience, we present some of our results below in discrete time with time step size $\delta t$, which we will eventually take to zero to derive time-continuous equations. We distinguish discrete and continuous time results by using time as an index versus as an argument of the corresponding time-dependent quantities, e.g. $\mathbf{u}_t$ vs. $\mathbf{u}(t)$.

We describe the joint statistics of presynaptic membrane potentials and spikes by a hierarchical generative model that has three layers of variables (*Figure 2—figure supplement 1A,B*). The global state of the system is described by a single binary variable, $z$ that switches between a quiescent ($-$) and an active ($+$) state following first-order Markovian dynamics (see Appendix for the extension to an arbitrary number of states). The transition rates to the active and quiescent states are given by $\Omega_+$ and $\Omega_-$, respectively.

The dynamics of (subthreshold) membrane potentials $\mathbf{u}$ are modeled as a multivariate Ornstein-Uhlenbeck (mOU) process, which yields random walk-like behavior that (unlike simple Brownian motion) decays exponentially towards a baseline defined by the resting potential $\bar{\mathbf{u}}$, which in turn depends on the momentary global state of the system, $z_t$:

$$\mathrm{P}(\mathbf{u}_t | \mathbf{u}_{t-\delta t}, z_t) \stackrel{\triangle}{=} \mathcal{N}\left(\mathbf{u}_t; \left(1 - \frac{\delta t}{\tau}\right)\mathbf{u}_{t-\delta t} + \frac{\delta t}{\tau}\,\overline{\mathbf{u}}^{(z_t)}, \delta t\,\mathbf{Q}\right) \tag{7}$$

where $\tau$ is the presynaptic time constant of the exponential decay, and $\mathbf{Q}$ is the 'process noise' covariance matrix determining the variance of individual membrane potentials (together with $\tau$) and, importantly, also the correlations between presynaptic neurons. It is straightforward to extend the model by also making these parameters state- (or in fact, neuron-) dependent.

Note that both the state switching and mOU components of this model introduce both spatial and temporal statistical dependencies in the membrane potentials and spike trains of presynaptic cells. In the rest of this paper, we informally refer to any statistical dependency (second or higher order) as 'correlation', and we write 'auto-correlation' when we refer to the correlations between the membrane potential (firing rate) values of the same neuron at different times, and 'cross-correlation' when referring to the correlation between the activities of two different cells (at the same time, or at different times). Also note that temporal and spatial correlations can not be studied in complete isolation in the case of smoothly varying signals, such as membrane potentials, as the cross-correlation between the activity of two presynaptic neurons always has a characteristic temporal profile. While it is possible to consider a presynaptic neuronal population completely lacking spatial correlations (i.e. independent presynaptic neurons, as in *Figure 1C*), having a population with only spatial but not temporal correlations would require the membrane potentials of the individual neurons to be temporally white noise – which is so far removed from reality that we did not consider this case worth pursuing.

More specifically, the timescale of temporal correlations (auto-correlations) in the model depend on the transition rates of the switching component, $\Omega_+$ and $\Omega_-$, and the presynaptic time constant of the mOU component, $\tau$, such that cells are auto-correlated as long as $\tau$, $\Omega_+^{-1}$, and $\Omega_-^{-1} > 0$. Spatial correlation (cross-correlations between different presynaptic neurons) also emerge from both components. First, the pairs of presynaptic neurons corresponding to the non-zero off-diagonal elements of $\mathbf{Q}$ matrix of the mOU component become correlated. Second, synchronous state transitions during state switching in the presynaptic ensemble introduce positive correlations. Importantly, in both the temporal and spatial domains, while the mOU process can only introduce second-order correlations (i.e. it makes membrane potentials be distributed according to a multivariate normal), the switching process introduces higher order correlations (such that membrane potentials are not normally distributed any more). These higher order correlations are stronger when the effect of state-switching is large relative to the membrane potential fluctuations within a single state.

Finally, instead of modeling the detailed dynamics of action potential generation, we model spiking phenomenologically by introducing a single discrete variable, $s_{i,t}$, for each presynaptic neuron that represents the number of spikes neuron $i$ fires in time step $t$. (Note that in the limit $\delta t \longrightarrow 0$ this variable becomes binary, i.e. there can never be more than one spike fired in a $\delta t$ time window.) Spiking in each cell only depends on the membrane potential of that cell, and follows an inhomogeneous Poisson process with the firing rate, $r$, being an exponential function of the membrane potential (*Gerstner and Kistler, 2002*):

$$\mathrm{P}(\mathbf{s}_t | \mathbf{u}_t) \stackrel{\triangle}{=} \prod_i \mathrm{Poisson}(s_{i,t}; \delta t\, r_{i,t}), \quad \text{with } r_{i,t} = g e^{\beta u_{i,t}} \tag{8}$$

where $\beta$ describes how deterministically the cell switches to firing at threshold ($u = 0$) and $g$ is the firing rate at that threshold. We modeled the absolute refractory period by not allowing the generation of spikes (i.e. setting $r_{i,t} = 0$) within a time window of length $\tau_{\mathrm{refr}}$ following each spike in a cell, regardless of its membrane potential.

The parameters of the presynaptic statistics used in the paper are given in *Table 1*. Examples of neural dynamics generated by the model are shown in *Figures 1*, *2*, *4*, and *5*.

## Inference and the optimal response

Our goal was to infer the posterior distribution of the membrane potential based on the spiking pattern observed up to time $t$, $\mathrm{P}(\mathbf{u}_t | \mathbf{s}_{0:t})$.

We first show that linear dendritic integration is sufficient when presynaptic neurons are statistically independent. We start by noting that by marginalising out the past membrane potential history of the presynaptic cells and using Bayes' rule, the posterior can *always* be written as

$$\mathrm{P}(\mathbf{u}_t|\mathbf{s}_{0:t}) = \int \mathrm{P}(\mathbf{u}_{0:t}|\mathbf{s}_{0:t})\mathrm{d}\mathbf{u}_{0:t-\delta t} \propto \int \mathrm{P}(\mathbf{s}_{0:t}|\mathbf{u}_{0:t})\ \mathrm{P}(\mathbf{u}_{0:t})\mathrm{d}\mathbf{u}_{0:t-\delta t} \tag{9}$$

and as the spikes of each neuron are independent from all other neurons conditioned on its own membrane potential history (*Equation 8*), this can be rewritten as

$$P(u_t|s_{0:t}) \propto \int \prod_i P(s_{i,0:t}|u_{i,0:t})\ P(u_{0:t})\mathrm{d}u_{i,0:t-\delta t} \tag{10}$$

In the special case when we assume that presynaptic neurons are *statistically independent*, i.e. their prior factorizes $\mathrm{P}(\mathbf{u}_{0:t}) = \prod_i \mathrm{P}(u_{i,0:t})$, the posterior also becomes factorised

$$\mathrm{P}(\mathbf{u}_t|\mathbf{s}_{0:t}) \propto \prod_i \int \mathrm{P}(s_{i,0:t}|u_{i,0:t})\ \mathrm{P}(u_{i,0:t})\mathrm{d}u_{i,0:t-\delta t} \tag{11}$$

$$= \prod_i \mathrm{P}(u_{i,t}|s_{i,0:t}) \tag{12}$$

which in continuous time reads simply as

$$\mathrm{P}\Big(\mathbf{u}(t)|\mathbf{s}(0:t)\Big) = \prod_i \mathrm{P}\Big(u_i(t)|s_i(0:t)\Big) \tag{13}$$

Thus, taking our usual assumption that the postsynaptic computation is linear (*Equation 5*), the optimal response in *Equation 6* can be written as

$$\tau_{\mathrm{post}}\ \dot{\tilde{\mathrm{v}}}(t) = -\tilde{\mathrm{v}}(t) + \sum_{i=1}^{N} w_i \int u_i(t)\ \mathrm{P}\Big(u_i(t)|s_i(0:t)\Big)\mathrm{d}u_i(t) \tag{14}$$

indicating that integration of inputs from different neurons is linear in this case (it is a weighted sum of terms each depending on just a single presynaptic neuron). However, even in this case, note that integration of input spikes from the same presynaptic neuron, i.e. the result of the integral over each $u_i(t)$ as a function of $s_i(0:t)$, is still nonlinear in general (*Pfister et al., 2010*). Indeed, *Equation 14* including these local nonlinearties was used to compute the linear response in *Figure 1C–D*.

In the general case inference can be performed using filtering such that in each step we update the inferred state of the hidden variables, $z_t$ and $\mathbf{u}_t$, using information from two different sources: the likelihood of emitting a particular spiking pattern (*observation*) and the dynamics of the hidden variables combined with the previous estimate (*innovation*):

$$\mathrm{P}(\mathbf{u}_t = \mathbf{u}, z_t = z|\mathrm{s}_{0:t}) \propto \mathrm{P}(\mathbf{s_t} = \mathbf{s}|\mathbf{u_t} = \mathbf{u}) \sum_{z'} \mathrm{P}(z_t = z|z_{t-\delta t} = z')\ \cdot$$
$$\cdot \int \mathrm{d}\mathbf{u}'\ \mathrm{P}(\mathbf{u}_t = \mathbf{u}|\mathbf{u}_{t-\delta t} = \mathbf{u}', z_t = z)\ \mathrm{P}(\mathbf{u}_{t-\delta t} = \mathbf{u}', z_{t-\delta t} = z'|\mathrm{s}_{0:t-\delta t}) \tag{15}$$

where the likelihood $\mathrm{P}(\mathbf{s}_t|\mathbf{u}_t)$ is defined by *Equation 8*, the dynamics of the global state variable $\mathrm{P}(z_t|z_{t-\delta t})$ is first order, Markovian (see above) and the state-dependent membrane potential dynamics $\mathrm{P}(\mathbf{u}_t|\mathbf{u}_{t-\delta t}, z_t)$ is given by *Equation 7*. *Equation 15* thus defines a mapping between the posterior distribution of the hidden variables in the previous time step (last term on RHS) and their current distribution (LHS). The posterior over membrane potentials can then be obtained by simply marginalising out the state variable:

$$\mathrm{P}(\mathbf{u}_t|\mathbf{s}_{0:t}) = \sum_z \mathrm{P}(\mathbf{u}_t, z_t = z|\mathbf{s}_{0:t}) \tag{16}$$

For the following, it is useful to represent the posterior as a product of two terms:

$$\mathrm{P}(\mathbf{u}_t = \mathbf{u}, z_t = z|\mathbf{s}_{0:t}) = \mathrm{P}(z_t = z|\mathbf{s}_{0:t})\ \mathrm{P}(\mathbf{u}_t = \mathbf{u}|z_t = z, \mathbf{s}_{0:t}) \tag{17}$$

As the state variable is binary, its posterior is a Bernoulli distribution which we parametrize by $\zeta$, without loss of generality:

$$P(z_t = +|\mathbf{s}_{0:t}) \stackrel{\triangle}{=} \zeta_t \tag{18}$$

However, in general, the posterior of the membrane potentials conditioned on the current state, $z_t$, can be arbitrarily complex. To allow an analytical reduction of the inference process, we adopted an assumed density filtering approach in which this distribution is moment-matched in each time step by a multivariate normal distribution which is thus described by two (sets of) parameters, its mean, $\mu_t^{(z)}$, and covariance, $\Sigma_t^{(\mathbf{z})}$:

$$P(\mathbf{u}_t = \mathbf{u}|z_t = +, \mathbf{s}_{0:t}) \stackrel{\triangle}{\simeq} \mathcal{N}(\mathbf{u}; \mu_t^+, \Sigma_t^+) \tag{19}$$

with the analogous equation for the posterior of $\mathbf{u}_t$ conditioned on $z_t$ being in the $-$ state.

One advantage of this parametric approach is that inference (filtering) can be implemented by updating only the parameters describing the (approximate) posterior distribution (*Equations 18–19*): $\zeta_t$, $\mu_t^{(z)}$, and $\Sigma_t^{(z)}$. In the Appendix we derive an analytical form for these parameter updates resulting in the following set of differential equations:

$$\dot{\zeta} = -\zeta(1-\zeta)(\overline{\gamma}^+ - \overline{\gamma}^-) + \zeta(1-\zeta)\ \mathbf{s}(t)^{\mathrm{T}}\langle\mathbf{\Gamma}\rangle^{-1}(\gamma^+ - \gamma^-) + (1-\zeta)\ \Omega_+ - \zeta\Omega_- \tag{20}$$

$$\dot{\mu}^+ = \frac{\overline{\mu}^+ - \mu^+}{\tau} + \beta\,\Sigma^+\Big(\mathbf{s}(t) - \gamma^+\Big) + \frac{1-\zeta}{\zeta}\Omega_+(\mu^- - \mu^+) \tag{21}$$

$$\dot{\Sigma}^+ = \frac{2}{\tau}\Big(\frac{\tau}{2}\mathbf{Q} - \Sigma^+\Big) - \beta^2\Sigma^+\Gamma^+\Sigma^+ + \frac{1-\zeta}{\zeta}\Omega_+[(\Sigma^- - \Sigma^+) + (\mu^- - \mu^+)(\mu^- - \mu^+)^{\mathrm{T}}] \tag{22}$$

where $\gamma^{(z)}$ ($\Gamma^{(z)}$) is a state-dependent vector (diagonal matrix) of which the elements $\gamma_i^{(z)} = \Gamma_{ii}^{(z)} = ge^{\beta\mu_i^{(z)} + \frac{1}{2}\beta^2\Sigma_{ii}^{(z)}}$ are the expected firing rates of the neurons in a given state, $\overline{\gamma}^{(z)} = \sum_i \gamma_i^{(z)}$ is the expected total population firing rate in state $z$, and $\langle\Gamma\rangle = \zeta\ \Gamma^+ + (1-\zeta)\ \Gamma^-$ is the expected firing rate of the cells averaged across states. In these equations, the spike trains of the presynaptic neurons are represented by the sum of Dirac-delta functions in continuous time and are denoted by $\mathbf{s}(t)$, to be distinguished from its discrete time analog, $\mathbf{s}_t$, such that $\mathbf{s}(t) = \lim_{\delta t \to 0} \mathbf{s}_t/\delta t$ (see *Equation A62* in the Appendix). The differential equations for the conditional mean and variance in the $-$ state, $\dot{\mu}^-$ and $\dot{\Sigma}^-$, are analogous to *Equations 21–22*. The absolute refractory period is taken into account by setting $\gamma_i = \Gamma_{ii} = 0$ after each observed spike for the duration of the refractory period, $\tau_{\mathrm{refr}}$, thus omitting the effect of the likelihood (terms containing $\gamma^{(z)}$ or $\Gamma^{(z)}$) from *Equations 20–22*.

The first term in *Equation 20* captures the decay in $\zeta_t$ that is proportional to the difference in the state conditional firing rates in the absence of presynaptic spikes; the second term expresses the instantaneous change in $\zeta_t$ after observing a spike, proportional to both the state estimation uncertainty, $\zeta(1-\zeta)$, and the differences in the conditional firing rates ($\gamma^+ - \gamma^-$); and the last term captures the decay of $\zeta_t$ to its steady state in the absence of observations. The filtering equations for the conditional mean and covariance (*Equations 21–22*) are each composed of three terms: the first term expresses the decay of the variable towards its baseline in the absence of observations; the second term captures the effect of the current observation (i.e. the presence or absence of a spike) on the variable; and the third term describes the changes in the variable caused by potential state transitions. This can be viewed an extension and generalization of earlier work deriving the equivalents of *Equations 21–22* for the special case of a single neuron without state-switching dynamics (*Pfister et al., 2010*).

Another advantage of the parametrization of the posterior we chose is that computing the optimal response, i.e. the posterior expectation of the simple linear functions that we consider in this paper, becomes straightforward (cf. *Equation 6*):

$$\tau_{\mathrm{post}}\,\dot{\tilde{v}}(t) = -\tilde{v}(t) + \sum_i w_i\Big(\zeta(t)\ \mu_i^+(t) + \Big(1 - \zeta(t)\Big)\ \mu_i^-(t)\Big) \tag{23}$$

In order to verify the assumed density filtering approximations used above we numerically integrated the system of differential equations (*Equations 20–22*) using the software package *R* (*R Core Team, 2012*; *Soetaert et al., 2010*) and compared the results with those obtained using standard

particle filters (**Doucet et al., 2001**). In these simulations we used 500,000 particles to evaluate **Equation 15** with 1 neuron and 2 states. **Figure 2—figure supplement 1C** shows that the results of assumed density filtering are essentially identical to those of particle filtering, confirming that the approximations we used were valid.

## Dendritic approximation of the optimal response

Here we first describe a simple canonical model of dendritic integration following Poirazi and Mel (**Poirazi and Mel, 2001**), and then show that it provides an approximation to the optimal response (**Equations 20–23**) in the limiting case in which presynaptic dynamics are dominated by simultaneous switching between a quiescent and an active state. In this simple dendritic model, inputs within a branch are integrated linearly:

$$\dot{v}_{\mathrm{lin}} = -\mathscr{A} v_{\mathrm{lin}} + \mathscr{B} \mathrm{s}(t) - \mathscr{C} \tag{24}$$

where $v_{\mathrm{lin}}$ is the variable linearly integrating inputs with weight $\mathscr{B}$, dendritic time constant $1/\mathscr{A}$ and steady state value $-\mathscr{C}/\mathscr{A}$ (in the absence of spikes), and $s(t) = \sum_i s_i(t)$ denotes the spike train of presynaptic neurons, collecting all spikes from the presynaptic population. The actual dendritic response, $v_{\mathrm{den}}$, is then given by mapping this linear response through a sigmoidal nonlinearity, scaled to be between $v_{\mathrm{min}}$ and $v_{\mathrm{max}}$ (**Figure 3A**, inset):

$$v_{\mathrm{den}}(t) = v_{\mathrm{min}} + (v_{\mathrm{max}} - v_{\mathrm{min}}) \, \frac{1}{1 + e^{-v_{\mathrm{lin}}(t)}} \tag{25}$$

To demonstrate that this reduced model of dendritic integration closely approximates the optimal response, we first note that under appropriate conditions ($\tau_{\mathrm{post}}$ is small, $N$ is large, $\mathbf{Q}$ is diagonal, and $\beta$ is small relative to the diagonal elements of $\mathbf{Q}$) the dynamics of the optimal response are dominated by the state switching process (**Equation 20**; see Appendix). Thus, the optimal response essentially follows the inference about the global state variable, $\zeta$, up to linear rescaling and filtering:

$$\tilde{\mathrm{v}}(t) \approx \overline{u}^- + (\overline{u}^+ - \overline{u}^-) \, \zeta(t) \tag{26}$$

with $\overline{u}^+$ and $\overline{u}^-$ respectively denoting the resting membrane potential in the active and quiescent states. As **Equation 26** is linear, all nonlinear interactions, corresponding to dendritic nonlinearities, must be contained in the temporal dynamics of the posterior probability of this global state variable being in the active state, $\zeta(t)$, which can be expressed as

$$\dot{\zeta} \approx \zeta(1 - \zeta)[B \, s(t) - C] \tag{27}$$

where constants $B$ and $C$ depend on the parameters of the presynaptic statistics (see Appendix). Note that the fact that $\zeta(1 - \zeta)$ multiplies **Equation 27** expresses the simple intuition that the size of the update to $\zeta$ (the posterior probability of $z = +$) in response to incoming information (presence, $B$ term, or absence of a spike, $C$ term) should be proportional to our current (posterior) uncertainty about $z$; and since the posterior is a Bernoulli distribution, the uncertainty associated with it is simply $\zeta(1 - \zeta)$.

The solution of **Equation 27** can be expressed in a form that is similar, albeit not identical (see below), to the canonical model for dendritic integration (**Equations 24–25**). This form requires the linear integration of incoming spikes

$$\dot{\nu} = B \, s(t) - C \tag{28}$$

and the temporal evolution of $\zeta$ is expressed as a sigmoidal function of the linearly integrated inputs $\nu$:

$$\zeta(t) = \frac{1}{1 + e^{-\nu(t)}} \tag{29}$$

Thus, dendrites with sigmoidal nonlinearity are near-optimal when their synaptic inputs switch between a quiescent and an active state.

The main difference between the dendritic integrator and the optimal response is that the dynamics of spike integration imply exponential decay towards a finite baseline in the former (there is a negative term in **Equation 24** which is scaled by $v_{\mathrm{lin}}$) and steady decrease towards negative

infinity in the latter (the only negative term in *Equation 28* is a constant, independent of $\nu$). This is because the approximations we used for deriving *Equation 27* were accurate only in the quasi-static case, when the state switching dynamics are infinitely slow. In this case, remote and more recent observations should have identical effects on the current value of $\zeta$ as they all correspond to the same underlying state. In the more general case, when state switching occurs with non-zero probability, more remote observations likely correspond to a state which has changed in the meantime, and should thus count for less, such that their effect on the current value of $\zeta$ should decay with time – leading to leaky integration of incoming spikes, similar to that in *Equation 24*.

## Simplified neuron models

To compare quantitatively the response of the linear and nonlinear dendrites to the optimal response in a computational task using realistic input statistics, we divided the presynaptic population into four groups (cell assemblies), where neurons within each group were statistically dependent (either through simple second-order correlations, *Figure 4—figure supplement 2*, or through sharing a common state variable, *Figure 4* and *Figure 4—figure supplement 2*) while neurons from different groups were independent (*Figure 4A*). In this task we used 4 different versions of the simplified neuron model (*Figure 4B*):

- The linear model responded to all incoming spikes with an identical postsynaptic potential (PSPs) characterized by its amplitude ($w_\ell$), decay time constant ($\tau_\ell$) and the resting potential ($\overline{v}_\ell$):

$$\dot{\tilde{v}}_\ell(t) = \frac{\overline{v}_\ell - \tilde{v}_\ell}{\tau_\ell} + w_\ell \, s(t) \tag{30}$$

where $s(t)$ is the total incoming spike train (as before). This model had three parameters.

- The model with a somatic nonlinearity had linear dendrites but a nonlinear soma. Motivated by our analytical calculations we used a sigmoidal nonlinearity:

$$\tilde{v}_g(t) = \frac{a_g}{1 + e^{-\beta_g(\tilde{v}_\ell(t) - \theta_g)}} - v_g \tag{31}$$

where we computed $\tilde{v}_\ell(t)$ as defined in *Equation 30* above with $w_\ell = 1$ and $\overline{v}_\ell = 0$ (as these parameters were interchangeable with $\beta_g$ and $\theta_g$). This model had five free parameters.

- The random dendrites model had four nonlinear dendritic subunits, each receiving inputs from a unique set of 10 neurons randomly selected from the four presynaptic assemblies, and integrating these inputs using a sigmoidal nonlinearity (*Equations 30-31*). Each subunit had its own set of five parameters, and the outputs of the subunits were simply averaged in the soma (without loss of generality), resulting in 20 parameters in total.
- The clustered dendrites model was similar to the random dendrites model, with the important difference that neurons in each cell assembly selectively targeted a single nonlinear dendritic branch. Since the presynaptic statistics was the same for all four branches, we constrained the parameters of the four dendritic subunits to be identical and this model had only five parameters (the same performance was achieved when we relaxed this constraint and optimised all 20 parameters).

To fit the models we generated 240 s-long samples of presynaptic activity and optimized the parameters of the models to minimize the squared error between the signal ($v$, the true average of the stimulated presynaptic potentials, cf. *Equation 5* with $w_i = \frac{1}{N}$) and their estimates ($\hat{v}$, the outputs of the models), averaged over the duration of the sample:

$$\epsilon_{\text{estimation}} = \frac{1}{T} \sum_{t=1}^{T} (\hat{v}_t - v_t)^2 \tag{32}$$

After training, we tested the different models in cross-validation, on a novel 120 s-long input sequence, and quantified their performance by the fraction of variance unexplained, i.e. the temporally averaged squared error, $\epsilon_{\text{estimation}}$, normalized by the variance of the signal (as a sensible upper

limit on the error – achievable by an estimator that predicts the prior mean, ignoring incoming spikes altogether):

$$\overline{\epsilon}_{\text{estimation}} = \frac{\epsilon_{\text{estimation}}}{\text{Var}[v]} \tag{33}$$

where $\text{Var}[v] = \frac{1}{T-1}\sum_t (v_t - \text{E}[v])^2$ and $\text{E}[v] = \frac{1}{T}\sum_t v_t$. (Normalization was unnecessary during training because the parameters of the models that we were optimizing obviously did not influence the variance of the signal.) *Figure 4* shows $1 - \overline{\epsilon}_{\text{estimation}}$, i.e. the fraction of variance explained as 'performance', and *Figure 1—figure supplement 3*, *Figure 3—figure supplement 2*, and *Figure 4—figure supplement 2* show $\overline{\epsilon}_{\text{estimation}}$ as 'estimation error'.

## Fitting and predicting experimental data

To predict dendritic integration in hippocampal and neocortical neurons we fitted the parameters describing presynaptic statistics in our model, $\text{P}(\mathbf{u})$ and $\text{P}(\mathbf{s}|\mathbf{u})$, to the statistical patterns in the activity of their respective presynaptic populations.

The basal dendrites of neocortical layer 2/3 pyramidal cells are targeted by neighbouring pyramidal neurons as well as by neurons from layer 4 (*Douglas and Martin, 2004*). We used *in vivo* intracellular paired recordings from layer 2/3 pyramidal neurons in the barrel cortex (*Poulet and Petersen, 2008*; *Gentet et al., 2010*; *Crochet et al., 2011*) to set the parameters of our model to reproduce the salient features of the presynaptic population dynamics during quiet wakefulness (*Tables 1–2*).

In the hippocampal experiments we stimulated synapses on the proximal dendrites of CA3 neurons targeted by recurrent collaterals of neighbouring pyramidal cells (*Andersen et al., 2007*). We fitted the presynaptic statistics to *in vivo* population activity patterns recorded from the hippocampus during quiet wakefulness, characterised by sharp wave (SPW) activity (*Csicsvari et al., 2000*). As intracellular recordings from CA3 pyramidal neurons during SPW activity in the awake animal are not available, we fitted the presynaptic statistics to awake extracellular data from CA3 (*Csicsvari et al., 2000*; *Grosmark et al., 2012*) and intracellular (*Ylinen et al., 1995*; *English et al., 2014*) data from CA1 pyramidal neurons (*Tables 1* and *3*).

## Comparing the optimal response to dendritic integration

We used four different parameter sets (models) to describe the activity of the presynaptic population (*Figure 1*). The parameters of the HP and NC models were fitted to *in vivo* recordings from the corresponding presynaptic populations as described above. As a control, we used two simpler models with no state switching dynamics. The *cor2* model had correlated membrane potential fluctuations with all cross-correlations between presynaptic neurons being the same, $-\frac{1}{N-1} \le \rho \le 1$. Neurons in the last model, *ind*, had independent membrane potential fluctuations. The HP, NC, and *ind* models had no free parameters, while parameter $\rho$ of the *cor2* model was left free and later tuned to fit dendritic data. Note that to fit supralinear cortical responses, $\rho$ had to be tuned to unnaturally large negative values in this model (*Figure 5—figure supplement 1A*) – and it still produced significantly poorer fits than the HP and NC models (*Figure 5*).

After setting the parameters of the presynaptic population, we computed the optimal response by numerically integrating *Equations 20–22* in the software package *R* (*R Core Team, 2012*; *Soetaert et al., 2010*). When comparing the optimal response to experimental data, we assumed that each uncaging event corresponded to a single spike at the presynaptic axon terminal, and spines not showing measurable gluEPSP were considered to be non-stimulated. For all four presynaptic parameter sets, we varied two postsynaptic parameters to fit the responses of the optimal estimator to dendritic integration data, i.e. the somatic membrane potential traces recorded in our in vitro experiments. These two parameters were the weight $w$ of the presynaptic neurons and the time constant of the postsynaptic filtering, $\tau_{\text{post}}$ (*Equation 5*). To avoid overfitting, we assumed that all presynaptic neurons had equal weight, i.e. $\forall i\; w_i = w$. A final free parameter that we had to consider was the number of synapses that were in the same functional cluster as the synapses we stimulated in our experiments – where the term 'functional cluster' refers to a set of synapses for which the presynaptic cells are correlated. This parameter was irrelevant for the *ind* model (by definition), it was fixed at 20 for the *cor2* model (because its effects on the optimal response were largely indistinguishable from that of varying $\rho$, see above), and it was tuned to fit dendritic integration for the

state-switching models (NC and HP). In sum, the number of free parameters used to fit dendritic integration data was 2 for the *ind* model and 3 for the *cor2*, NC, and HP models. We confirmed that the higher number of free parameters in the latter models did not result in an unfair advantage in fitting performance by using Bayesian Information Criterion (BIC), rather than squared error (see below), as our measure of performance. BIC includes an explicit term penalizing the number of parameters, and our results were not qualitatively affected by it: fitting the model using the relevant *in vivo* statistics resulted in 3500±940 (NC, mean±s.d.) and 2000±1100 (HP) higher BIC scores than when using independent statistics (where each unit of BIC difference corresponds to a likelihood that is higher by a factor of $e \simeq 2.71$).

We fitted each recorded neuron independently using these parameters by minimizing the mean squared error between the predicted, $\tilde{v}$, and the recorded postsynaptic membrane potential (averaged across repetitions of the same stimulation in the same cell), $\overline{v}^*$:

$$\epsilon_{\text{fitting}} = \frac{1}{T}\sum_{t=1}^{T}(\tilde{v}_t - \overline{v}_t^*)^2 \tag{34}$$

To be able to compare results across different neurons and different stimulation protocols, we normalized the error by the total variance of the data:

$$\overline{\epsilon}_{\text{fitting}} = \frac{\epsilon_{\text{fitting}}}{\text{Var}[\overline{v}^*]} \tag{35}$$

where $\text{Var}[\overline{v}^*] = \frac{1}{T-1}\sum_t(\overline{v}_t^* - \text{E}[\overline{v}^*])^2$ and $\text{E}[\overline{v}^*] = \frac{1}{T}\sum_t \overline{v}_t^*$. A natural lower bound of our fitting error was the intrinsic variability of the data, so we computed the mean of the variance of the experimental data across repetitions, normalized by the total variance of the data:

$$\overline{\epsilon}_{\text{min}} = \frac{\frac{1}{L-1}\sum_{l=1}^{L}\frac{1}{T}\sum_t(v_{t,l}^* - \overline{v}_t^*)^2}{\text{Var}[\overline{v}^*]} \tag{36}$$

where $v_{t,l}^*$ is the raw data before averaging across repetitions, $L$ is the number of repetitions using the same stimulation protocol in the same cell and $\overline{v}_t^* = \frac{1}{L}\sum_L v_{t,l}^*$. *Figures 5D,H* and *Figure 5—figure supplement 1G* show $\overline{\epsilon}_{\text{fitting}}$ as the 'fitting error' and $\overline{\epsilon}_{\text{min}}$ as 'var'. The best fitting parameter values for the postsynaptic time constant, $\tau_{\text{post}}$, and total number of neurons in a functional cluster, $N$, are shown in *Figure 5—figure supplement 1B,C*.

## Experimental methods

### Neocortex

#### Slice preparation and electrophysiology

Acute sagittal brain slices (300 μm) incorporating both visual and somatosensory cortex were prepared from 3–6 week-old Sprague-Dawley rats as previously described (*Sjöström and Häusser, 2006*) and in accordance with institutional and national guidelines. Experiments were carried out at 32°C–35°C in artificial cerebrospinal fluid (ACSF) containing (in mM): NaCl 125, KCl 2.5, glucose 25, NaH$_2$PO$_4$ 1.25, NaHCO$_3$ 25, MgCl$_2$ 1, CaCl$_2$ 2 (pH 7.3 when bubbled with 95% O$_2$ and 5% CO$_2$). Somatic whole-cell recordings were obtained with a Multiclamp 700B amplifier (Molecular Devices, Sunnyvale, CA), and data was acquired at 50 kHz using custom-written software in Matlab 7.2 (Mathworks, Natick, MA) interfacing with an ITC-18 A/D board (Instrutech, Holliston, MA). Patch pipettes had a resistance of 3–6 MΩ when filled with a solution containing (in mM): KMeSO$_4$ 130, HEPES 10, KCl 7, MgATP 2, Na2ATP 2, Na$_2$GTP 0.3, EGTA 0.05 (pH 7.2) and R$_{\text{series}}$ was <30 MΩ. For visualization of cell morphology Alexa Fluor 594 (100 μM; Invitrogen, Carlsbad, CA) was added to the internal solution.

#### Two-Photon imaging and uncaging

Simultaneous 2-photon imaging and uncaging was performed using a dual galvanometer-based scanning system (Prairie Technologies, Middleton, WI) using two Ti:sapphire pulsed lasers (MaiTai, Spectra-Physics, Santa Clara, CA), one tuned to 840 nm for imaging cell morphology, and another tuned to 720 nm for photolysis of MNI-caged-L-glutamate. Neurons were visualized using an Olympus BX51WI objective (60x, 0.9 NA; Olympus, Melville, NY). Two-photon glutamate uncaging was

carried out based on previously published methods (*Gasparini and Magee, 2006*; *Losonczy and Magee, 2006*; *Branco and Häusser, 2011*). MNI-caged-L-glutamate (12 mM, Tocris Cookson, UK) was dissolved in (in mM): NaCl 125, KCl 2.5, HEPES 10, CaCl$_2$ 2, MgCl$_2$ 1, glucose 25, and puffed locally. To block NMDA receptors (*Figure 5E,J*, blue squares), 500 µM D-AP5 was included in the glutamate puffing pipette.

## Stimulation and data analysis

A short burst of presynaptic uncaging events equally spaced in time with inter stimulus interval (ISI) between 1 and 20 ms was applied on 1–7 visually identified dendritic spines. For each dendritic branch either the number of stimuli or the ISI was varied. Uncaging exposure time was 100–500 µs and the inter-trial interval was 10 s. All data was acquired using custom written software in Matlab 7.2 (Mathworks). The original data recorded at 50 kHz were averaged across identical trials, filtered with a Gaussian kernel with $\sigma = 0.2$ ms and subsampled at 2 kHz for analyses. All spines were responsive in the neocortical experiments.

## Hippocampus

### Slice preparation and electrophysiology

Adult male Sprague-Dawley rats (8–12 week-old) were used to prepare transverse slices (400 $\mu$m) from the hippocampus similarly to that described previously (*Losonczy and Magee, 2006*), according to methods approved by the Janelia Farm Institutional Animal Care and Use Committee and the Animal Care and Use Committee (ACUC) of the Institute of Experimental Medicine, Hungarian Academy of Sciences, and in accordance with 86/609/EEC/2 and DIRECTIVE 2010/63/EU Directives of the EU. Experiments were carried out at 33°C–35°C in artificial cerebrospinal fluid (ACSF) containing (in mM): NaCl 125, KCl 3, glucose 25, NaH$_2$PO$_4$ 1.25, NaHCO$_3$ 25, MgCl$_2$ 1, CaCl$_2$ 1.3, Na-pyruvate 3, and ascorbic acid 1 saturated with 95% O$_2$ and 5% CO$_2$. Somatic whole-cell recordings were obtained with BVC-700 amplifier (Dagan, Minneapolis, MN) in the active 'bridge' mode, filtered at 3 kHz and data was acquired at 50 kHz. Patch pipettes had a resistance of 2–6 M$\Omega$ when filled with a solution containing (in mM): K-gluconate 120, KCl 20, HEPES 10, NaCl 4, Mg$_2$ATP 4, Tris$_2$GTP 0.3, phosphocreatine 14, complemented with 100 $\mu$M Alexa Fluor 488 (Invitrogen-Molecular Probes, Eugene, OR) and ~0.1–0.3% biocytin (Sigma), pH=7.25 and R$_{series}$ was <30 M$\Omega$. Alexa Fluor 488 fluorescence or biocytin labeling with immunoperoxidase reaction was used for post hoc verification of the localization of neurons along the proximodistal axis of CA3. All CA3 neurons included in this study had resting membrane potentials between −62 and −72 mV. Cells were hyperpolarized when necessary to avoid action potential firing during synaptic stimulation.

### Two-Photon imaging and uncaging

A dual galvanometer based two photon scanning system (Prairie Technologies, Middleton, WI) equipped with an Olympus BX-61 microscope (60X, 0.9 NA objective) was used to image Alexa 488-loaded neurons and to uncage glutamate at individual dendritic spines as described (*Losonczy and Magee, 2006*; *Makara and Magee, 2013*). Two ultrafast pulsed laser beams (Chameleon Ultra II; Coherent, Auburn, CA, USA) were used, one at 920 nm for imaging Alexa 488 and the other at 720 nm to photolyze MNI-caged-L-glutamate (Tocris Cookson, Ballwin, MO, USA; 10 mM applied through a pipette above the slice). Laser beam intensity was independently controlled with electro-optical modulators (Model 350–50, Conoptics, Danbury, CT, USA). Unitary gluEPSP amplitude and rise time was close to that of mEPSPs as measured by sucrose application at dendritic segments 70–168 $\mu$m from the soma as described before (*Magee and Cook, 2000*; *Makara and Magee, 2013*). To standardize these experiments, results were included in the analysis only if 1) at least $65\%$ of the selected spines were responsive (see below), 2) the average amplitude of the successful unitary gluEPSPs was 0.2–0.6 mV and maximum unitary gluEPSP amplitude was $\leq 1.2$ mV, 3) at least 5 mV expected amplitude was achieved, and 4) unitary responses were stabile with repeated stimulation. To block NMDA receptors 50 $\mu$M AP5 was added to both bath solution and the puffing pipette solution.

### Stimulation and data analysis

Dendritic branches on basal dendritic segments 100–160 $\mu$m from the soma were stimulated by synchronous uncaging of MNI-glutamate at a spatially clustered set (1–32) of visually identified spines

using 0.2 ms uncaging duration with different intervals (in the range of 0.1–5 ms) between synapses. For each dendritic branch either the number of stimuli (control) or the ISI (NMDA block) was varied. To detect non-responsive spines, we fitted the individual responses with a double-exponential function:

$$\phi(t) = \phi_0\left(e^{-(t-t_0)/\tau_1} - e^{-(t-t_0)/\tau_2}\right)$$

(37)

Spines were classified as non-responsive if the rise time of the gluEPSP ($\tau_2$) was slower than 15 ms or its start time ($t_0$) was more than 5 ms (relative to the stimulation time). The original data recorded at 50 kHz were averaged across identical trials, filtered with a Gaussian kernel with $\sigma = 0.2$ ms and subsampled at 2 kHz for analyses.

## Acknowledgements

We thank J-P Pfister for valuable insights and comments on earlier versions of the manuscript, B Gutkin, Sz Káli, and A Gulyás for useful discussions, and J Magee, M Häusser and D Wolpert for support and helpful comments.

## Additional information

### Funding

| Funder | Grant reference number | Author |
|---|---|---|
| Hungarian Scientific Research Fund | OTKA MB08A 84471 | Balazs B Ujfalussy |
| Hungarian Academy of Sciences | LP 2011-012 | Judit K Makara |
| Wellcome Trust | 090915/Z/09/Z | Judit K Makara |
| Medical Research Council | | Tiago Branco |
| Royal Society Fellowship | | Tiago Branco |
| Wellcome Trust | 098400MA | Tiago Branco |
| Wellcome Trust | 095621/Z/11/Z | Máté Lengyel |

The funders had no role in study design, data collection and interpretation, or the decision to submit the work for publication.

### Author contributions

BBU, Concept and design, Acquisition of Data, Analysis and Interpretation of Data, Drafting and Revising the article; JKM, TB, Acquisition of Data, Drafting and Revising the article; ML, Concept and design, Analysis and Interpretation of Data, Drafting and Revising the article

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

# Appendix

## A Derivation of the optimal response

Here we derive the filtering equations for the switching multivariate OU (mOU) process introduced in the main text, such that we generalise the two-state model presented there to an arbitrary number of hidden states.

### A.1 The generative model of presynaptic activities

The generative model of presynaptic activities has two hidden and one visible layers (*Figure 2—figure supplement 1*). The state of the system is described by a global discrete state variable, $z$, and its dynamics is characterised by the state transition matrix, $\boldsymbol{\Omega}$:

$$\mathrm{P}(z_t = z | z_{t-\delta t} = z') \triangleq \delta_{zz'}(1 - \delta t \overline{\Omega}_{z'}) + (1 - \delta_{zz'})\delta t \Omega_{zz'} \tag{A1}$$

where $\Omega_{zz'}$ is the rate of transitioning from state $z'$ to $z$, and

$$\overline{\Omega}_{z'} = \sum_{z \neq z'} \Omega_{zz'}, \ \Omega_{zz'} \geq 0, \ \ \delta t \ll \frac{1}{\max_{z'} \overline{\Omega}_{z'}}$$

and $\delta_{zz'}$ is the Kronecker delta function. This description becomes identical to the state switching dynamics presented in the main text in the case of a binary state variable, $z \in \{-, +\}$.

The dynamics of the membrane potentials $\mathbf{u}$ are modeled as a mOU process where the resting potential, $\overline{\mathbf{u}}^{(z_t)}$, the time constant of the neurons, $\tau^{(z_t)}$, and the covariance of the process noise, $\mathbf{Q}^{(z_t)}$, may depend on the current state, $z_t$:

$$\mathrm{P}(\mathbf{u}_t | \mathbf{u}_{t-\delta t}, z_t) \triangleq \mathcal{N}\left(\mathbf{u}_t; \left(1 - \frac{\delta t}{\tau^{(z_t)}}\right) \mathbf{u}_{t-\delta t} + \frac{\delta t}{\tau^{(z_t)}} \overline{\mathbf{u}}^{(z_t)}, \delta t \, \mathbf{Q}^{(z_t)}\right) \tag{A2}$$

For simplicity, in the main text we assumed that only the resting potential depends on the state variable.

The observations are spike counts, $\mathbf{s}$, which are given by an inhomogeneous Poisson process for each neuron independently from the other neurons given its own membrane potential:

$$\mathrm{P}(\mathbf{s}_t | \mathbf{u}_t) \triangleq \prod_i \mathrm{Poisson}\left(s_{t,i}; \delta t \, g \, e^{\beta \, u_{t,i}}\right) \tag{A3}$$

where $g$ is the baseline firing rate and $\beta$ is a parameter controlling the nonlinearity of the spiking.

### A.2 The inference problem

Our goal is to infer the posterior distribution of the current membrane potentials based on the spiking pattern observed up to time $t$:

$$\mathrm{P}(\mathbf{u}_t | \mathbf{s}_{0:t}) = \sum_{z_t} \mathrm{P}(\mathbf{u}_t | z_t, \mathbf{s}_{0:t}) \, \mathrm{P}(z_t | \mathbf{s}_{0:t}) \tag{A4}$$

$$= \sum_{z_t} \mathrm{P}(\mathbf{u}_t, z_t | \mathbf{s}_{0:t}) \tag{A5}$$

Inference of the value of the hidden variables requires information from two different sources: the likelihood of emitting a particular spiking pattern (*observation*) and the dynamics of the hidden variables combined with the previous estimate (*innovation*).

$$P(\mathbf{u}_t = \mathbf{u}, z_t = z | \mathbf{s}_{0:t}) \propto \overbrace{P(\mathbf{s}_t = \mathbf{s} | \mathbf{u}_t = \mathbf{u})}^{f_s(\delta t, \mathbf{u}): \text{observation}} \cdot \tag{A6}$$

$$\cdot \sum_{z'} P(z_t = z | z_{t-\delta t} = z') \, P(z_{t-\delta t} = z' | \mathbf{s}_{0:t-\delta t}) \cdot$$

$$\cdot \underbrace{\int d\mathbf{u}' \, P(\mathbf{u}_t = \mathbf{u} | \mathbf{u}_{t-\delta t} = \mathbf{u}', z_t = z) \, P(\mathbf{u}_{t-\delta t} = \mathbf{u}' | z_{t-\delta t} = z', \, \mathbf{s}_{0:t-\delta t})}_{h(\delta t, \mathbf{u}, z) = P(\mathbf{u}_t = \mathbf{u}, z_t = z | \mathbf{s}_{0:t-\delta t}): \text{ innovation}}$$

where the likelihood $P(\mathbf{s}_t | \mathbf{u}_t)$ is defined by **Equation A3**, the dynamics of the global state variable $P(z_t | z_{t-\delta t})$ is defined by **Equation A1** and the state-dependent membrane potential dynamics $P(\mathbf{u}_t = \mathbf{u} | \mathbf{u}_{t-\delta t} = \mathbf{u}', z_t = z)$ is given by **Equation A2**. **Equation A6** is the filtering equation, analogous to **Equation 15** in the main text. Note that both the observation $f_s(\delta t, \mathbf{u})$ and the innovation $h(\delta t, \mathbf{u}, z)$ terms depend on $\delta t$, the time step used in the inference.

In general, computing **Equation A6** is intractable so we take an assumed density filtering approach in which we approximate the posterior in each time step with a particular parametric form and derive how the parameters of this approximation change due to **Equation A6.** Specifically, we represent the posterior using the following factorisation (without approximation):

$$P(\mathbf{u}_t = \mathbf{u}, z_t = z | \mathbf{s}_{0:t}) = P(z_t = z | \mathbf{s}_{0:t}) \, P(\mathbf{u}_t = \mathbf{u} | z_t = z, \mathbf{s}_{0:t}) \tag{A7}$$

The first term of **Equation A7** is the posterior over the current value of the discrete hidden state variable, $z_t$, which is a discrete distribution that can be parametrised by a single (vector) parameter, $\zeta_t$ (still without approximation):

$$P(z_t = z | \mathbf{s}_{0:t}) \stackrel{\triangle}{=} \zeta_t^{(z)} \tag{A8}$$

The second term of **Equation A7** is the distribution of the membrane potentials conditioned on the current state, which we approximate by a multivariate Gaussian with parameters $\mu_t^{(z)}$ and $\Sigma_t^{(z)}$ respectively describing the conditional mean and covariance of the presynaptic membrane potentials:

$$P(\mathbf{u}_t = \mathbf{u} | z_t = z, \mathbf{s}_{0:t}) \stackrel{\triangle}{\simeq} \mathcal{N}(\mathbf{u}; \mu_t^{(z)}, \Sigma_t^{(z)}) \tag{A9}$$

In sum, this approximate posterior has three sets of parameters that fully characterise it: $\zeta_t$, $\mu_t^{(z)}$, and $\Sigma_t^{(z)}$. In Sections A.3-A.6 we derive the update dynamics for these parameters when observing arbitrary spiking patterns.

For notational convenience, we also introduce a further quantity, which can be derived from these parameters: the expected firing rate of the cells in each hidden state, either written as a vector $\gamma_t^{(z)}$ or as a diagonal matrix $\Gamma_t^{(z)}$, whose elements are

$$\gamma_{t,i}^{(z)} = \Gamma_{t,ii}^{(z)} = \int P(u_{t,i} = u | z_t = z, \mathbf{s}_{0:t}) \, g e^{\beta u} du = g e^{\beta \mu_{t,i}^{(z)} + \frac{1}{2} \beta^2 \Sigma_{t,ii}^{(z)}} \tag{A10}$$

## A.3 Parameter updates

The advantage of describing the elements of the posterior in a parametric form is that we only need to update the parameters of *Equations A8-A9*. As the parameter updates are derived using *Equation A6*, in which the innovation and the observation terms depended on $\delta t$, the parameter updates will also depend on $\delta t$. Eventually, to derive continuous-time dynamics, we will take the limit of $\delta t \longrightarrow 0$. Incidentally, the Gaussian approximation in *Equation A9* is also more accurate with small $\delta t$.

We rewrite the parameters of the approximate posterior by using auxiliary variables (this also makes it explicit how the update rule depends on $\delta t$):

$$\zeta_t^{(z)}(\delta t) = \frac{a_{\mathbf{s}_t}(\delta t, z)}{b_{\mathbf{s}_t}(\delta t)} \tag{A11}$$

$$\mu_t^{(z)}(\delta t) = \frac{\mathbf{m}_{\mathbf{s}_t}(\delta t, z)}{a_{\mathbf{s}_t}(\delta t, z)} \tag{A12}$$

$$\mathbf{\Xi}_t^{(z)}(\delta t) = \frac{\mathbf{C}_{\mathbf{s}_t}(\delta t, z)}{a_{\mathbf{s}_t}(\delta t, z)} \tag{A13}$$

$$\mathbf{\Sigma}_t^{(z)}(\delta t) = \mathbf{\Xi}_t^{(z)}(\delta t) - \mu_t^{(z)}(\delta t) \left( \mu_t^{(z)}(\delta t) \right)^{\mathrm{T}} \tag{A14}$$

where the auxiliaries are defined as:

$$a_{\mathbf{s}}(\delta t, z) \triangleq \int f_{\mathbf{s}}(\delta t, \mathbf{u}) \; h(\delta t, \mathbf{u}, z) \; \mathrm{d}\mathbf{u} \tag{A15}$$

$$b_{\mathbf{s}}(\delta t) \triangleq \sum_z a_{\mathbf{s}}(\delta t, z) \tag{A16}$$

$$\mathbf{m}_{\mathbf{s}}(\delta t, z) \triangleq \int \mathbf{u} f_{\mathbf{s}}(\delta t, \mathbf{u}) \; h(\delta t, \mathbf{u}, z) \; \mathrm{d}\mathbf{u} \tag{A17}$$

$$\mathbf{C}_{\mathbf{s}}(\delta t, z) \triangleq \int \mathbf{u}\mathbf{u}^{\mathrm{T}} f_{\mathbf{s}}(\delta t, \mathbf{u}) \; h(\delta t, \mathbf{u}, z) \; \mathrm{d}\mathbf{u} \tag{A18}$$

where $f_{\mathbf{s}}(\delta t, \mathbf{u})$ and $h(\delta t, \mathbf{u}, z)$ were defined in *Equation A6*.

## A.4 First-order approximations to parameter updates

Our goal is to derive differential equations for the parameter updates. Therefore we will assume that $\delta t$ is sufficiently small (i.e. $\delta t \longrightarrow 0$) to use a first order approximation of *Equations A11-A13* around $\delta t = 0$:

$$\zeta_t^{(z)}(\delta t) \simeq \zeta_t^{(z)}(0) + \delta t \zeta_t^{'}(0) = \frac{a_{\mathbf{s}_t}(0, z)}{b_{\mathbf{s}_t}(0)} + \delta t \frac{a_{\mathbf{s}_t}^{'}(0, z) \; b_{\mathbf{s}_t}(0) - a_{\mathbf{s}_t}(0, z) \; b_{\mathbf{s}_t}^{'}(0)}{b_{\mathbf{s}_t}^2(0)} \tag{A19}$$

$$\mu_t^{(z)}(\delta t) \simeq \mu_t^{(z)}(0) + \delta t \mu_t^{'(z)}(0) = \frac{\mathbf{m}_{\mathbf{s}_t}(0, z)}{a_{\mathbf{s}_t}(0, z)} + \delta t \frac{\mathbf{m}_{\mathbf{s}_t}^{'}(0, z) \; a_{\mathbf{s}_t}(0, z) - \mathbf{m}_{\mathbf{s}_t}(0, z) \; a_{\mathbf{s}_t}^{'}(0, z)}{a_{\mathbf{s}_t}^2(0, z)} \tag{A20}$$

$$\Xi_t^{(z)}(\delta t) = \Xi_t^{(z)}(0) + \delta t \Xi_t^{'\,(z)}(0) = \frac{\mathbf{C}_{\mathbf{s}_t}(0,z)}{a_{\mathbf{s}_t}(0,z)} + \delta t \frac{\mathbf{C}'_{\mathbf{s}_t}(0,z)\ a_{\mathbf{s}_t}(0,z) - \mathbf{C}_{\mathbf{s}_t}(0,z)\ a'_{\mathbf{s}_t}(0,z)}{a^2_{\mathbf{s}_t}(0,z)} \tag{A21}$$

where

$$a'_s(\delta t, z) = \overbrace{\int f'_s(\delta t, \mathbf{u})\ h(\delta t, \mathbf{u}, z)\ \mathrm{d}\mathbf{u}}^{a_I} + \overbrace{\int f_s(\delta t, \mathbf{u})\ h'(\delta t, \mathbf{u}, z)\ \mathrm{d}\mathbf{u}}^{a_{II}} \tag{A22}$$

$$b'_s(\delta t) = \sum_z a'_s(\delta t, z) \tag{A23}$$

$$m'_s(\delta t, z) = \overbrace{\int \mathbf{u} f'_s(\delta t, \mathbf{u})\ h(\delta t, \mathbf{u}, z)\ \mathrm{d}\mathbf{u}}^{m_I} + \overbrace{\int \mathbf{u} f_s(\delta t, \mathbf{u})\ h'(\delta t, \mathbf{u}, z)\ \mathrm{d}\mathbf{u}}^{m_{II}} \tag{A24}$$

$$C'_s(\delta t, z) = \underbrace{\int \mathbf{u}\mathbf{u}^T f'_s(\delta t, \mathbf{u})\ h(\delta t, \mathbf{u}, z)\ \mathrm{d}\mathbf{u}}_{C_I} + \underbrace{\int \mathbf{u}\mathbf{u}^T f_s(\delta t, \mathbf{u})\ h'(\delta t, \mathbf{u}, z)\ \mathrm{d}\mathbf{u}}_{C_{II}} \tag{A25}$$

In particular, in order to compute **Equations A19-A21** we will need the value of $h(\delta t, \mathbf{u}, z)$ and $h'(\delta t, \mathbf{u}, z)$ with $\delta t = 0$ (see Section A.8.1):

$$h(0, \mathbf{u}, z) = \zeta_{t-\delta t}^{(z)} \cdot \mathcal{N}(\mathbf{u}; \mu_{t-\delta t}^{(z)}, \Sigma_{t-\delta t}^{(z)}) \tag{A26}$$

$$\begin{aligned}
h'(0, \mathbf{u}, z) &= \sum_{z'} \zeta_{t-\delta t}^{(z')} \big[(1 - \delta_{zz'})\Omega_{zz'} - \delta_{zz'}\overline{\Omega}_{z'}\big] \mathcal{N}\left(\mathbf{u}; \mu_{t-\delta t}^{(z')}, \Sigma_{t-\delta t}^{(z')}\right) - \\
&\quad -\frac{1}{2}\zeta_{t-\delta t}^{(z)}\mathcal{N}\left(\mathbf{u}; \mu_{t-\delta t}^{(z)}, \Sigma_{t-\delta t}^{(z)}\right) \cdot \\
&\quad \cdot \Big[\mathrm{Tr}\big[\Sigma_{t-\delta t}^{(z)\,-1} Q^{(z)} + \frac{2}{\tau^{(z)}}\mathbf{I}\big] + \frac{2}{\tau^{(z)}}\left(\overline{\mathbf{u}}^{(z)} - \mu_{t-\delta t}^{(z)}\right)^T \Sigma_{t-\delta t}^{(z)\,-1}\left(\mathbf{u} - \mu_{t-\delta t}^{(z)}\right) - \\
&\quad -\left(\mathbf{u} - \mu_{t-\delta t}^{(z)}\right)^T \Sigma_{t-\delta t}^{(z)\,-1}\left(\mathbf{Q}^{(z)} - \frac{2}{\tau^{(z)}}\Sigma_{t-\delta t}^{(z)}\right)\Sigma_{t-\delta t}^{(z)\,-1}\left(\mathbf{u} - \mu_{t-\delta t}^{(z)}\right)\Big]
\end{aligned} \tag{A27}$$

Having derived the general form of the parameter updates we need to apply it to the possible observations, i.e. derive how observing a particular presynaptic spiking pattern at time $t$ changes the posterior distribution over the state variable, $\mathrm{P}(z_t = z | \mathbf{s}_{0:t})$ and the presynaptic membrane potentials, $\mathrm{P}(\mathbf{u}_t = \mathbf{u} | z_t = z, \mathbf{s}_{0:t})$. In the limit of $\delta t \longrightarrow 0$ we consider only two alternative outcomes of the observation process: the case of observing zero or one spike in a single time bin.

## A.5 Observation: No spikes

We first calculate the likelihood (sec. A.5.1) and the auxiliaries (sec. A.5.2) with $\mathbf{s}_t = 0$ and then derive the corresponding parameter updates in sec. A.5.3.

A.5.1 Likelihood

$$f_0(\delta t, \mathbf{u}) = \mathrm{P}(\mathbf{s}_t = \mathbf{0} | \mathbf{u}_t = \mathbf{u}) \tag{A28}$$

$$= \prod_i \text{Poisson}\left(0;\, \delta t\, g\, e^{\beta\, u_i}\right) \tag{A29}$$

$$= e^{-\delta t g \sum_i e^{\beta\, u_i}} \tag{A30}$$

$$f_0(\delta t = 0, \mathbf{u}) = 1 \tag{A31}$$

$$f_0'(\delta t, \mathbf{u}) = -\left(g \sum_i e^{\beta\, u_i}\right) f_0(\delta t, \mathbf{u}) \tag{A32}$$

$$f_0'(\delta t = 0, \mathbf{u}) = -g \sum_i e^{\beta\, u_i} \tag{A33}$$

### A.5.2 Auxiliaries

Here we describe the values of the auxiliaries required to compute *Equations A19-A21* with the likelihood function given above:

$$a_0(0, z) = \zeta_{t-\delta t}^{(z)} \tag{A34}$$

$$b_0(0) = 1 \tag{A35}$$

$$\mathbf{m}_0(0, z) = \zeta_{t-\delta t}^{(z)} \mu_{t-\delta t}^{(z)} \tag{A36}$$

$$\mathbf{C}_0(0, z) = \zeta_{t-\delta t}^{(z)} \left(\Sigma_{t-\delta t}^{(z)} + \mu_{t-\delta t}^{(z)} (\mu_{t-\delta t}^{(z)})^{\mathrm{T}}\right) \tag{A37}$$

The following quantities have been derived in *Ujfalussy et al., 2011*:

$$a_I(0, z) = -\zeta_{t-\delta t}^{(z)} \overline{\gamma}_{t-\delta t}^{(z)} \tag{A38}$$

$$\mathbf{m}_I(0, z) = -\zeta_{t-\delta t}^{(z)} \left[\overline{\gamma}_{t-\delta t}^{(z)} \mu_{t-\delta t}^{(z)} + \beta \Sigma_{t-\delta t}^{(z)} \gamma_{t-\delta t}^{(z)}\right] \tag{A39}$$

$$\mathbf{C}_I(0, z) = -\zeta_{t-\delta t}^{(z)} \left[\overline{\gamma}_{t-\delta t}^{(z)} (\mu_{t-\delta t}^{(z)} (\mu_{t-\delta t}^{(z)})^{\mathrm{T}} + \Sigma_{t-\delta t}^{(z)}) + 2\beta \mu_{t-\delta t}^{(z)} (\gamma_{t-\delta t}^{(z)})^{\mathrm{T}} \Sigma_{t-\delta t}^{(z)} + \beta^2 \Sigma_{t-\delta t}^{(z)} \Gamma_{t-\delta t}^{(z)} \Sigma_{t-\delta t}^{(z)}\right] \tag{A40}$$

where $\overline{\gamma}_t^{(z)} = \sum_i \gamma_{t,i}^{(z)}$ is the expected population firing rate in state $z$.

The derivation of the following auxiliaries is given in Section A.8.1:

$$a_{II}(0, z) = \sum_{z'} \zeta_{t-\delta t}^{(z')} \left[(1 - \delta_{zz'})\Omega_{zz'} - \delta_{zz'}\overline{\Omega}_{z'}\right] \tag{A41}$$

$$\mathbf{m}_{II}(0, z) = \sum_{z'} \zeta_{t-\delta t}^{(z')} \left[(1 - \delta_{zz'})\Omega_{zz'} - \delta_{zz'}\overline{\Omega}_{z'}\right]\mu_{t-\delta t}^{(z')} + \zeta_{t-\delta t}^{(z')} \frac{\overline{u}^{(z)} - \mu_{t-\delta t}^{(z)}}{\tau^{(z)}} \tag{A42}$$

$$\mathbf{C}_{II}(0,z) = \sum_{z'} \zeta_{t-\delta t}^{(z')} \Big[ (1-\delta_{zz'})\Omega_{zz'} - \delta_{zz'}\overline{\Omega}_{z'} \Big] \Big( \mu_{t-\delta t}^{(z')} (\mu_{t-\delta t}^{(z')})^T + \Sigma_{t-\delta t}^{(z')} \Big) +$$

$$+ \zeta_{t-\delta t}^{(z)} \bigg[ \Big( Q^{(z)} - \frac{2}{\tau^{(z)}} \Sigma_{t-\delta t}^{(z)} \Big) + \frac{1}{\tau^{(z)}} \mu_{t-\delta t}^{(z)} (\overline{\mathbf{u}}^{(z)} - \mu_{t-\delta t}^{(z)})^{\mathrm{T}} +$$

$$+ \frac{1}{\tau^{(z)}} (\overline{\mathbf{u}}^{(z)} - \mu_{t-\delta t}^{(z)})(\mu_{t-\delta t}^{(z)})^{\mathrm{T}} \bigg] \qquad (A43)$$

A.5.3 Parameter updates

Now we are ready to substitute *Equation A34-A43* into *Equation A19-A21* to derive the parameter updates.

**State probabilities**

$$\zeta_t^{(z)}(\delta t) = \frac{a_0(0,z)}{b_0(0)} + \delta t \frac{a_0'(0,z)\ b_0(0) - a_0(0,z)\ b_0'(0)}{b_0^2(0)}$$

$$= \zeta_{t-\delta t}^{(z)} + \delta t \bigg[ -\zeta_{t-\delta t}^{(z)} \overline{\gamma}_{t-\delta t}^{(z)} + \sum_{z'} \zeta_{t-\delta t}^{(z')} \big[ (1-\delta_{zz'})\Omega_{zz'} - \delta_{zz'}\overline{\Omega}_{z'} \big] -$$

$$- \zeta_{t-\delta t}^{(z)} \Big( \sum_{z''} -\zeta_{t-\delta t}^{(z'')} \overline{\gamma}_{t-\delta t}^{(z'')} + \sum_{z',z''} \zeta_{t-\delta t}^{(z')} [(1-\delta_{z''z'})\Omega_{z''z'} - \delta_{z''z'}\overline{\Omega}_{z'}] \Big) \bigg] \qquad (A44)$$

$$= \zeta_{t-\delta t}^{(z)} + \delta t \bigg[ -\zeta_{t-\delta t}^{(z)} \big( \overline{\gamma}_{t-\delta t}^{(z)} - \sum_{z'} \zeta_{t-\delta t}^{(z')} \overline{\gamma}_{t-\delta t}^{(z')} \big) +$$

$$+ \sum_{z'} \zeta_{t-\delta t}^{(z')} [(1-\delta_{zz'})\Omega_{zz'} - \delta_{zz'}\overline{\Omega}_{z'}] - \zeta_{t-\delta t}^{(z)} \sum_{z',z''} \zeta_{t-\delta t}^{(z')} [(1-\delta_{z''z'})\,\Omega_{z''z'} - \delta_{z''z'}\overline{\Omega}_{z'}] \bigg]$$

However, since $\overline{\Omega}_{z'} = \sum_{z''} \Omega_{z''z'}$ the last terms in *Equation A44* cancel each other and we can write that

$$\zeta_t^{(z)}(\delta t) = \zeta_{t-\delta t}^{(z)} + \delta t [ -\zeta_{t-\delta t}^{(z)} (\overline{\gamma}_{t-\delta t}^{(z)} - \langle \overline{\gamma}_{t-\delta t} \rangle) + \sum_{z'} \zeta_{t-\delta t}^{(z')} [(1-\delta_{zz'})\Omega_{zz'} - \delta_{zz'}\overline{\Omega}_{z'}]] \qquad (A45)$$

where $\langle \cdot \rangle$ denotes averaging over $z'$, and so $\langle \overline{\gamma}_t \rangle = \sum_{z'} \zeta_{t-\delta t}^{(z')} \overline{\gamma}_t^{(z')}$ is the average expected population firing rate. In *Equation A45* two different processes are changing the posterior state probabilities. First, changes in $\zeta^{(z)}$ are proportional to the difference between the total firing rate in state $z$, $\overline{\gamma}^{(z)}$, and the average firing rate, $\langle \overline{\gamma} \rangle$. This process causes a decay in the posterior probability of those states in which the expected firing rate is high and thus incompatible with the current observation (no spikes). Second, posterior state probabilities are changing according to the prior state transition dynamics captured by $\Omega$.

**Posterior mean**

$$
\begin{aligned}
\mu_t^{(z)}(\delta t) \;\simeq\; & \frac{\mathbf{m}_0(0,z)}{a_0(0,z)} + \delta t \frac{\mathbf{m}_0'(0,z)\; a_0(0,z) - \mathbf{m}_0(0,z)\; a_0'(0,z)}{a_0^2(0,z)} \\[2mm]
=\; & \frac{\zeta_{t-\delta t}^{(z)}\mu_{t-\delta t}^{(z)}}{\zeta_{t-\delta t}^{(z)}} + \delta t \frac{1}{\zeta_{t-\delta t}^{(z)}{}^2}\Bigg[\Big(-\zeta_{t-\delta t}^{(z)}[\overline{\gamma}_{t-\delta t}^{(z)}\mu_{t-\delta t}^{(z)} + \beta\,\Sigma_{t-\delta t}^{(z)}\gamma_{t-\delta t}^{(z)}] + \\[2mm]
& + \sum_{z'}\zeta_{t-\delta t}^{(z')}[(1-\delta_{zz'})\Omega_{zz'} - \delta_{zz'}\overline{\Omega}_z]\mu_{t-\delta t}^{(z')} + \zeta_{t-\delta t}^{(z)}\,\frac{\overline{\mathbf{u}}^{(z)} - \mu_{t-\delta t}^{(z)}}{\tau^{(z)}}\Big)\,\zeta_{t-\delta t}^{(z)} - \\[2mm]
& - \zeta_{t-\delta t}^{(z)}\mu_{t-\delta t}^{(z)}\Big(-\zeta_{t-\delta t}^{(z)}\overline{\gamma}_{t-\delta t}^{(z)} + \sum_{z'}\zeta_{t-\delta t}^{(z')}[(1-\delta_{zz'})\Omega_{zz'} - \delta_{zz'}\overline{\Omega}_z]\Big)\Bigg] \\[2mm]
=\; & \mu_{t-\delta t}^{(z)} + \delta t\Bigg[\frac{\overline{\mathbf{u}}^{(z)} - \mu_{t-\delta t}^{(z)}}{\tau^{(z)}} - \beta\Sigma_{t-\delta t}^{(z)}\gamma_{t-\delta t}^{(z)} + \sum_{z'}\frac{\zeta_{t-\delta t}^{(z')}}{\zeta_{t-\delta t}^{(z)}}(\mu_{t-\delta t}^{(z')} - \mu_{t-\delta t}^{(z)})[(1-\delta_{zz'})\Omega_{zz'} - \delta_{zz'}\overline{\Omega}_{z'}]\Bigg] \\[2mm]
=\; & \mu_{t-\delta t}^{(z)} + \delta t\Bigg[\frac{\overline{\mathbf{u}}^{(z)} - \mu_{t-\delta t}^{(z)}}{\tau^{(z)}} - \beta\Sigma_{t-\delta t}^{(z)}\gamma_{t-\delta t}^{(z)} + \sum_{z'\neq z}\Omega_{zz'}\frac{\zeta_{t-\delta t}^{(z')}}{\zeta_{t-\delta t}^{(z)}}(\mu_{t-\delta t}^{(z')} - \mu_{t-\delta t}^{(z)})\Bigg]
\end{aligned}
$$

$$(A46)$$

The evolution of the conditional mean is governed by three factors. The first term in the bracket describes the decay of the posterior mean towards the prior mean with time constant $\tau^{(z)}$. The second term captures the effect of the observation (absence of spikes) where the decrease in the posterior mean is proportional to the posterior covariance $\Sigma^{(z)}$ (because when more uncertain *a priori*, the observation should have a larger impact) and the expected firing rate $\gamma^{(z)}$ (because a higher expected firing rate is more incompatible with the absence of spikes and thus requires a larger correction). This term is omitted in the absence of observations, i.e. during the absolute refractory period. Finally, the third term expresses the effect of a possible transition from each of the other states $z'$ to state $z$ which is proportional to the product of the prior transition rate $\Omega_{zz'}$ and the relative probabilities of the two states (i.e. the relative proportion of the probability mass coming from state $z'$ against that already existing in state $z$), and the difference between the respective posterior means (as the more different these are, the larger the bias contributed by the probability mass coming from state $z'$ is).

**Posterior covariance**

$$
\begin{aligned}
\Xi_t^{(z)}(\delta t) \;&=\; \Xi_t^{(z)}(0)+\delta t\,\Xi_t'^{(z)}(0)\\[4pt]
&=\frac{\mathbf{C}_0(0,z)}{a_0(0,z)}+\delta t\,\frac{\mathbf{C}_0'(0,z)\,a_0(0,z)-\mathbf{C}_0(0,z)\,a_0'(0,z)}{a_0^2(0,z)}\\[4pt]
&=\frac{\zeta_{t-\delta t}^{(z)}\big(\,\mathbf{\Sigma}_{t-\delta t}^{(z)}+\mu_{t-\delta t}^{(z)}(\mu_{t-\delta t}^{(z)})^{\mathrm{T}}\big)}{\zeta_{t-\delta t}^{(z)}}+\\[4pt]
&\quad+\delta t\frac{1}{\zeta_{t-\delta t}^{(z)^2}}\bigg[-\zeta_{t-\delta t}^{(z)^2}\Big(\overline{\boldsymbol{\gamma}}_{t-\delta t}^{(z)}[\mu_{t-\delta t}^{(z)}(\mu_{t-\delta t}^{(z)})^{\mathrm{T}}+\mathbf{\Sigma}_{t-\delta t}^{(z)}]+2\beta\mu_{t-\delta t}^{(z)}(\boldsymbol{\gamma}_{t-\delta t}^{(z)})^{\mathrm{T}}\mathbf{\Sigma}_{t-\delta t}^{(z)}\\[4pt]
&\quad+\beta^2\,\mathbf{\Sigma}_{t-\delta t}^{(z)}\Gamma_{t-\delta t}^{(z)}\mathbf{\Sigma}_{t-\delta t}^{(z)}\Big)+\\[4pt]
&\quad+\zeta_{t-\delta t}^{(z)}\sum_{z'}\zeta_{t-\delta t}^{(z')}[(1-\delta_{zz'})\Omega_{zz'}-\delta_{zz'}\overline{\Omega}_{z'}](\mu_{t-\delta t}^{(z')}(\mu_{t-\delta t}^{(z')})^{\mathrm{T}}+\mathbf{\Sigma}_{t-\delta t}^{(z')})+\\[4pt]
&\quad+\zeta_{t-\delta t}^{(z)^2}\bigg[\Big(\mathbf{Q}^{(z)}-\frac{2}{\tau^{(z)}}\,\mathbf{\Sigma}_{t-\delta t}^{(z)}\Big)+\frac{1}{\tau^{(z)}}\mu_{t-\delta t}^{(z)}(\overline{\mathbf{u}}^{(z)}-\mu_{t-\delta t}^{(z)})^{\mathrm{T}}+\\[4pt]
&\quad+\frac{1}{\tau^{(z)}}(\overline{\mathbf{u}}^{(z)}-\mu_{t-\delta t}^{(z)})(\mu_{t-\delta t}^{(z)})^{\mathrm{T}}\bigg]-\zeta_{t-\delta t}^{(z)}(\,\mathbf{\Sigma}_{t-\delta t}^{(z)}+\mu_{t-\delta t}^{(z)}(\mu_{t-\delta t}^{(z)})^{\mathrm{T}})\;(-\zeta_{t-\delta t}^{(z)}\overline{\boldsymbol{\gamma}}_{t-\delta t}^{(z)}\\[4pt]
&\quad+\sum_{z'}\zeta_{t-\delta t}^{(z')}[(1-\delta_{zz'})\Omega_{zz'}-\delta_{zz'}\overline{\Omega}_{z'}])\bigg]\\[4pt]
&=\mathbf{\Sigma}_{t-\delta t}^{(z)}+\mu_{t-\delta t}^{(z)}(\mu_{t-\delta t}^{(z)})^{\mathrm{T}}+\delta t\bigg[-2\beta\mu_{t-\delta t}^{(z)}(\boldsymbol{\gamma}_{t-\delta t}^{(z)})^{\mathrm{T}}\mathbf{\Sigma}_{t-\delta t}^{(z)}-\\[4pt]
&\quad-\beta^2\,\mathbf{\Sigma}_{t-\delta t}^{(z)}\Gamma_{t-\delta t}^{(z)}\mathbf{\Sigma}_{t-\delta t}^{(z)}+\\[4pt]
&\quad+\sum_{z'}\frac{\zeta_{t-\delta t}^{(z')}}{\zeta_{t-\delta t}^{(z)}}[(1-\delta_{zz'})\Omega_{zz'}-\delta_{zz'}\overline{\Omega}_{z'}](\mu_{t-\delta t}^{(z')}(\mu_{t-\delta t}^{(z')})^{\mathrm{T}}+\mathbf{\Sigma}_{t-\delta t}^{(z')})+\\[4pt]
&\quad+\Big(\mathbf{Q}^{(z)}-\frac{2}{\tau^{(z)}}\,\mathbf{\Sigma}_{t-\delta t}^{(z)}\Big)+\frac{1}{\tau^{(z)}}\mu_{t-\delta t}^{(z)}(\overline{\mathbf{u}}^{(z)}-\mu_{t-\delta t}^{(z)})^{\mathrm{T}}+\frac{1}{\tau^{(z)}}(\overline{\mathbf{u}}^{(z)}-\mu_{t-\delta t}^{(z)})(\mu_{t-\delta t}^{(z)})^{\mathrm{T}}-\\[4pt]
&\quad-(\,\mathbf{\Sigma}_{t-\delta t}^{(z)}+\mu_{t-\delta t}^{(z)}(\mu_{t-\delta t}^{(z)})^{\mathrm{T}})\,\Big(\sum_{z'}\frac{\zeta_{t-\delta t}^{(z')}}{\zeta_{t-\delta t}^{(z)}}[(1-\delta_{zz'})\Omega_{zz'}-\delta_{zz'}\overline{\Omega}_{z'}]\Big)\bigg]\\[4pt]
&=\mathbf{\Sigma}_{t-\delta t}^{(z)}+\mu_{t-\delta t}^{(z)}(\mu_{t-\delta t}^{(z)})^{\mathrm{T}}+\delta t\bigg[\mathbf{Q}^{(z)}-\frac{2}{\tau^{(z)}}\,\mathbf{\Sigma}_{t-\delta t}^{(z)}-\beta^2\,\mathbf{\Sigma}_{t-\delta t}^{(z)}\Gamma_{t-\delta t}^{(z)}\mathbf{\Sigma}_{t-\delta t}^{(z)}+\\[4pt]
&\quad+\sum_{z'}\frac{\zeta_{t-\delta t}^{(z')}}{\zeta_{t-\delta t}^{(z)}}[(1-\delta_{zz'})\Omega_{zz'}-\delta_{zz'}\overline{\Omega}_{z'}]\Big(\mu_{t-\delta t}^{(z')}(\mu_{t-\delta t}^{(z')})^{\mathrm{T}}\\[4pt]
&\quad-\mu_{t-\delta t}^{(z)}(\mu_{t-\delta t}^{(z)})^{\mathrm{T}}+\mathbf{\Sigma}_{t-\delta t}^{(z')}-\mathbf{\Sigma}_{t-\delta t}^{(z)}\Big)+\frac{1}{\tau^{(z)}}\mu_{t-\delta t}^{(z)}(\overline{\mathbf{u}}^{(z)}-\mu_{t-\delta t}^{(z)})^{\mathrm{T}}+\\[4pt]
&\quad+\frac{1}{\tau^{(z)}}(\overline{\mathbf{u}}^{(z)}-\mu_{t-\delta t}^{(z)})(\mu_{t-\delta t}^{(z)})^{\mathrm{T}}-2\beta\mu_{t-\delta t}^{(z)}(\boldsymbol{\gamma}_{t-\delta t}^{(z)})^{\mathrm{T}}\mathbf{\Sigma}_{t-\delta t}^{(z)}\bigg]
\end{aligned}
\tag{A47}
$$

And finally, in order to compute the covariance, we will need:

$$
\begin{aligned}
\mu_t^{(z)}(\mu_t^{(z)})^{\mathrm{T}}\;=\;&\mu_{t-\delta t}^{(z)}\,(\mu_{t-\delta t}^{(z)})^{\mathrm{T}}+\delta t\bigg[\frac{\overline{\mathbf{u}}^{(z)}-\mu_{t-\delta t}^{(z)}}{\tau^{(z)}}\Big(\mu_{t-\delta t}^{(z)}\Big)^{\mathrm{T}}+\mu_{t-\delta t}^{(z)}\Big(\frac{\overline{\mathbf{u}}^{(z)}-\mu_{t-\delta t}^{(z)}}{\tau^{(z)}}\Big)^{\mathrm{T}}\\[4pt]
&-2\beta\Sigma_{t-\delta t}^{(z)}\gamma_{t-\delta t}^{(z)}(\mu_{t-\delta t}^{(z)})^{\mathrm{T}}+\\[4pt]
&\sum_{z'}\frac{\zeta_{t-\delta t}^{(z')}}{\zeta_{t-\delta t}^{(z)}}\Big(\mu_{t-\delta t}^{(z')}-\mu_{t-\delta t}^{(z)}\Big)\,(\mu_{t-\delta t}^{(z)})^{\mathrm{T}}[(1-\delta_{zz'})\Omega_{zz'}-\delta_{zz'}\overline{\Omega}_{z'}]+\\[4pt]
&\sum_{z'}\frac{\zeta_{t-\delta t}^{(z')}}{\zeta_{t-\delta t}^{(z)}}\mu_{t-\delta t}^{(z)}\,(\mu_{t-\delta t}^{(z')}-\mu_{t-\delta t}^{(z)})^{\mathrm{T}}[(1-\delta_{zz'})\Omega_{zz'}-\delta_{zz'}\overline{\Omega}_{z'}]\bigg]+\mathcal{O}\big(\delta t^2\big)
\end{aligned}
\tag{A48}
$$

Now we substitute *Equation A47-A48* into *Equation A14* to obtain the linear form for the change of the covariance matrix:

$$
\begin{aligned}
\mathbf{\Sigma}_t^{(z)} &= \mathbf{\Sigma}_{t-\delta t}^{(z)} + \delta t \big[ Q^{(z)} - \frac{2}{\tau^{(z)}} \mathbf{\Sigma}_{t-\delta t}^{(z)} - \beta^2 \mathbf{\Sigma}_{t-\delta t}^{(z)} \Gamma_{t-\delta t}^{(z)} \mathbf{\Sigma}_{t-\delta t}^{(z)} + \\
&\quad + \sum_{z'} \frac{\zeta_{t-\delta t}^{(z')}}{\zeta_{t-\delta t}^{(z)}} \big[ (1-\delta_{zz'})\Omega_{zz'} - \delta_{zz'}\overline{\Omega}_{z'} \big] \big( \mathbf{\Sigma}_{t-\delta t}^{(z')} - \mathbf{\Sigma}_{t-\delta t}^{(z)} \big) \\
&\quad + \sum_{z'} \frac{\zeta_{t-\delta t}^{(z')}}{\zeta_{t-\delta t}^{(z)}} \big[ (1-\delta_{zz'})\Omega_{zz'} - \delta_{zz'}\overline{\Omega}_{z'} \big] \big( \mu_{t-\delta t}^{(z')} - \mu_{t-\delta t}^{(z)} \big)\big( \mu_{t-\delta t}^{(z')} - \mu_{t-\delta t}^{(z)} \big)^{\mathrm{T}} \big] \\
&= \overset{(z)}{\underset{t-\delta t}{\phantom{.}}} + \delta t \big[ \mathbf{Q}^{(z)} - \frac{2}{\tau^{(z)}} \mathbf{\Sigma}_{t-\delta t}^{(z)} - \beta^2 \mathbf{\Sigma}_{t-\delta t}^{(z)} \Gamma_{t-\delta t}^{(z)} \mathbf{\Sigma}_{t-\delta t}^{(z)} + \\
&\quad + \sum_{z' \neq z} \Omega_{zz'} \frac{\zeta_{t-\delta t}^{(z')}}{\zeta_{t-\delta t}^{(z)}} \big( \mathbf{\Sigma}_{t-\delta t}^{(z')} - \mathbf{\Sigma}_{t-\delta t}^{(z)} \big) + \sum_{z' \neq z} \Omega_{zz'} \frac{\zeta_{t-\delta t}^{(z')}}{\zeta_{t-\delta t}^{(z)}} \big( \mu_{t-\delta t}^{(z')} - \mu_{t-\delta t}^{(z)} \big)\big( \mu_{t-\delta t}^{(z')} - \mu_{t-\delta t}^{(z)} \big)^{\mathrm{T}} \big]
\end{aligned}
\tag{A49}
$$

The evolution of the posterior covariance is governed by the same three factors as the evolution of the conditional mean. The first is the decay of the posterior covariance towards the prior covariance $\frac{\tau^{(z)}}{2}\mathbf{Q}^{(z)}$ with time constant $\frac{\tau^{(z)}}{2}$. (Note that this decay is twice faster than that of the mean.) The second term captures the effect of the observation (absence of spikes) such that the decrease in the posterior covariance is proportional to the square of the covariance $\mathbf{\Sigma}^{(z)}$ and the expected firing rate $\Gamma^{(z)}$ (because this observation is more informative, i.e. it causes a larger surprise, when the expected firing rate is higher). Finally, the last term expresses the effect of possible hidden transitions between state $z$ and state $z'$ on the posterior covariance (which is again proportional to the relative proportion of the probability masses and the differences between the two states).

## A.6 Observation: cell $i$ emits a spike

Now we assume that cell $i$ emits a spike while all other neurons remain silent in the current time step. Note that as $\delta t \longrightarrow 0$ the probability of having multiple presynaptic neurons spiking in the same time bin, or the same cell firing twice, converges to zero.

A.6.1 Likelihood

In this case the current spiking pattern is denoted by $\mathbf{s}_t = \hat{\mathbf{s}}^{(i)}$, that is $\hat{\mathbf{s}}_j^{(i)} \overset{\triangle}{=} \delta_{ij}$ and the likelihood is given by

$$
\begin{aligned}
f_{\hat{\mathbf{s}}^{(i)}}(\delta t, \mathbf{u}) &= \mathrm{P}(\mathbf{s}_t = \hat{\mathbf{s}}^{(i)} | \mathbf{u}_t = \mathbf{u}) \\
&= \mathrm{Poisson}(1; \delta t \, g \, e^{\beta \, u_i}) \prod_{j \neq i} \mathrm{Poisson}(0; \delta t \, g \, e^{\beta \, u_j}) \\
&= \delta t \, g \, e^{\beta \, u_i} \cdot e^{-\delta t \, g \sum_j e^{\beta \, u_j}}
\end{aligned}
\tag{A50}
$$

$$
f_{\hat{\mathbf{s}}^{(i)}}(\delta t = 0, \mathbf{u}) = 0
\tag{A51}
$$

The derivative of the likelihood is

$$f'_{\hat{s}^{(i)}}(\delta t, \mathbf{u}) = g\, e^{\beta\, u_i} \cdot e^{-\delta t\, g \sum_j e^{\beta\, u_j}} - \delta t g e^{\beta\, u_i} \cdot g\Big(\sum_j e^{\beta\, u_j}\Big)\, e^{-\delta t\, g \sum_j e^{\beta\, u_j}}$$

$$= g\, e^{\beta\, u_i} \cdot e^{-\delta t\, g \sum_j e^{\beta\, u_j}} \cdot \Big(1 - \delta t\, g \sum_j e^{\beta\, u_j}\Big) \tag{A52}$$

$$f'_{\hat{s}^{(i)}}(\delta t = 0, \mathbf{u}) = g\, e^{\beta\, u_i} \tag{A53}$$

In the following, we follow the same steps as in the previous section, but replacing $f_0(0, \mathbf{u})$ and $f'_0(0, \mathbf{u})$ with $f_{\hat{s}^{(i)}}(0, \mathbf{u})$ and $f'_{\hat{s}^{(i)}}(0, \mathbf{u})$, respectively.

A.6.2 Auxiliaries

Again, the goal is to derive differential equations for the parameter updates. As the number of time bins with spikes does not grow with $\delta t \longrightarrow 0$ we only need to compute terms up to zeroth order (cf. *Equations A19-A21*). However, as the likelihood (*Equation A51*) is 0 when $\delta t = 0$, the values of the auxiliaries in *Equations A54-A56* are also 0 (*Equations A15-A18*), and so we shall use l'Hôpital's rule to deal with the $\delta t \longrightarrow 0$ limit:

$$\zeta_t^{(z)}(\delta t) \simeq \zeta_t^{(z)}(0) = \frac{a_{\mathbf{s}_t}(0, z)}{b_{\mathbf{s}_t}(0)} = \frac{a'_{\mathbf{s}_t}(0, z)}{b'_{\mathbf{s}_t}(0)} \tag{A54}$$

$$\mu_t^{(z)}(\delta t) \simeq \mu_t^{(z)}(0) = \frac{\mathbf{m}_{\mathbf{s}_t}(0, z)}{a_{\mathbf{s}_t}(0, z)} = \frac{\mathbf{m}'_{\mathbf{s}_t}(0, z)}{a'_{\mathbf{s}_t}(0, z)} \tag{A55}$$

$$\Xi_t^{(z)}(\delta t) = \Xi_t^{(z)}(0) = \frac{\mathbf{C}_{\mathbf{s}_t}(0, z)}{a_{\mathbf{s}_t}(0, z)} = \frac{\mathbf{C}'_{\mathbf{s}_t}(0, z)}{a'_{\mathbf{s}_t}(0, z)} \tag{A56}$$

Substituting *Equation A53* into *Equations A22-A25* gives the following values for the corresponding derivatives (for details see *Ujfalussy et al., 2011*):

$$a'_{\hat{s}^{(i)}}(0, z) = \zeta_{t-\delta t}^{(z)} \gamma_{t-\delta t, i}^{(z)} \tag{A57}$$

$$b'_{\hat{s}^{(i)}}(0, z) = \sum_{z'} \zeta_{t-\delta t}^{(z')} \gamma_{t-\delta t, i}^{(z')} \tag{A58}$$

$$\mathbf{m}'_{\hat{s}^{(i)}}(0, z) = \zeta_{t-\delta t}^{(z)} \gamma_{t-\delta t, i}^{(z)} (\mu_{t-\delta t}^{(z)} + \beta\, \Sigma_{t-\delta t}^{(z)} \hat{\mathbf{s}}^{(i)}) \tag{A59}$$

$$\mathbf{C}'_{\hat{s}^{(i)}}(0, z) = \zeta_{t-\delta t}^{(z)} \gamma_{t-\delta t, i}^{(z)} [(\mu_{t-\delta t}^{(z)} + \beta\, \Sigma_{t-\delta t}^{(z)} \hat{\mathbf{s}}^{(i)})(\mu_{t-\delta t}^{(z)} + \beta\, \Sigma_{t-\delta t}^{(z)} \hat{\mathbf{s}}^{(i)})^{\mathrm{T}} + \Sigma_{t-\delta t}^{(z)}] \tag{A60}$$

A.6.3 Parameter updates

Finally, by substituting *Equations A57-A60* back into *Equations A54-A56*, we can derive the parameter updates in the case of observing a spike.

**State probabilities**

$$\zeta_t^{(z)}(\delta t = 0) \;=\; \frac{a_{\hat{\mathbf{s}}^{(i)}}'(0,z)}{\sum_{z'} a_{\hat{\mathbf{s}}^{(i)}}'(0,z')} = \frac{\zeta_{t-\delta t}^{(z)}\gamma_{t-\delta t,i}^{(z)}}{\sum_{z'}\zeta_{t-\delta t}^{(z')}\gamma_{t-\delta t,i}^{(z')}}$$

$$=\; \zeta_{t-\delta t}^{(z)} + \frac{\zeta_{t-\delta t}^{(z)}(\gamma_{t-\delta t,i}^{(z)} - \langle\gamma_{t-\delta t,i}\rangle)}{\langle\gamma_{t-\delta t,i}\rangle} \tag{A61}$$

where $\langle\gamma_{t,i}\rangle = \sum_{z'}\zeta_{t-\delta t}^{(z')}\gamma_{t,i}^{(z')}$ is the average (over states) expected firing rate of cell $i$.
*Equation A61* shows that the posterior state probabilities change instantaneously after observing a spike, and the change is proportional to the normalised difference between the expected firing rate of cell $i$ in state $z$ and its state-averaged expected firing rate. Therefore if state $z$ is more compatible with the observed spiking pattern than state $z'$, i.e. $\gamma_i^{(z)} > \gamma_i^{(z')}$, then the posterior probability of being in state $z$, $\zeta^{(z)}$, will increase while $\zeta^{(z')}$ will decrease.

*Equation A61* can be rewritten into vector form as follows:

$$\frac{\zeta_t^{(z)} - \zeta_{t-\delta t}^{(z)}}{\delta t} = \zeta_{t-\delta t}^{(z)}\frac{\mathbf{s}_t}{\delta t}\langle\Gamma_{t-\delta t}\rangle^{-1}(\gamma_{t-\delta t}^{(z)} - \langle\gamma_{t-\delta t}\rangle)$$

which in turn can be rewritten in differential equation form by taking the $\delta t \longrightarrow 0$ limit:

$$\dot{\zeta}^{(z)} = \zeta^{(z)}\mathbf{s}(t)\,\langle\Gamma\rangle^{-1}(\gamma^{(z)} - \langle\gamma\rangle) \tag{A62}$$

where $\mathbf{s}(t) = \lim_{\delta t \longrightarrow 0} \mathbf{s}_t/\delta t$ is the sum of Dirac-delta functions representing the presynaptic spike trains.

**Posterior mean**

$$\mu_t^{(z)}(\delta t = 0) = \frac{\mathbf{m}_{\hat{\mathbf{s}}^{(i)}}'(0,z)}{a_{\hat{\mathbf{s}}^{(i)}}'(0,z)} = \mu_{t-\delta t}^{(z)} + \beta\,\Sigma_{t-\delta t}^{(z)}\mathbf{s}_t \tag{A63}$$

Thus, the posterior conditional means of the presynaptic membrane potentials also change instantaneously after observing a spike in the presynaptic population. The change is proportional to the posterior variance in the case of the cell that emitted the spike while for other presynaptic neurons the change is proportional to the posterior covariance between the given cell and the neuron that emitted the spike. Note that, similar to the change in the posterior state probabilities, the change of the mean is instantaneous and it does not depend on the time step.

The differential equation form of *Equation A63* reads as

$$\frac{\mu_t^{(z)} - \mu_{t-\delta t}^{(z)}}{\delta t} = \beta\,\Sigma_{t-\delta t}^{(z)}\frac{\mathbf{s}_t}{\delta t}$$
$$\dot{\mu}^{(z)} = \beta\,\Sigma_{t-\delta t}^{(z)}\mathbf{s}(t) \tag{A64}$$

where $\mathbf{s}(t)$ is, again, the sum of Dirac-delta functions representing the presynaptic spike trains.

**Posterior covariance**

$$\Xi_t^{(z)}(0) = \frac{\mathbf{C}_{\hat{\mathbf{s}}^{(i)}}'(0,z)}{a_{\hat{\mathbf{s}}^{(i)}}'(0,z)} = \Sigma_{t-\delta t}^{(z)} + (\mu_{t-\delta t}^{(z)} + \beta\,\Sigma_{t-\delta t}^{(z)}\hat{\mathbf{s}}^{(i)})(\mu_{t-\delta t}^{(z)} + \beta\,\Sigma_{t-\delta t}^{(z)}\hat{\mathbf{s}}^{(i)})^{\mathrm{T}} \tag{A65}$$

$$\Sigma_t^{(z)}(0) = \Sigma_{t-\delta t}^{(z)} \tag{A66}$$

Thus, *Equation A66* indicates that the posterior covariance does not change directly after observing a spike – only indirectly, through the increase in the posterior mean (*Equation A63*) and thus $\Gamma$ (*Equation A10*), which in turn decreases the covariance in the silent period following the spike (*Equation A49*).

## A.7 Differential equations

Here we summarize the results of *Equations A45,A46,A49*, and *Equation A61-A66* as differential equations in continuous time:

$$
\begin{aligned}
\dot{\zeta}^{(z)} &= -\zeta^{(z)}(\overline{\gamma}^{(z)} - \langle\overline{\gamma}\rangle) + \zeta^{(z)}\mathbf{s}(t)^{\mathrm{T}}\langle\Gamma\rangle^{-1}(\gamma^{(z)} - \langle\gamma\rangle) \\
&+ \sum_{z'}\zeta^{(z')}[(1 - \delta_{zz'})\Omega_{zz'} - \delta_{zz'}\overline{\Omega}_{z'}]
\end{aligned}
\tag{A67}
$$

$$
\dot{\mu}^{(z)} = \frac{\overline{\mathbf{u}}^{(z)} - \mu^{(z)}}{\tau^{(z)}} + \beta\,\Sigma^{(z)}\left(\mathbf{s}(t) - \gamma^{(z)}\right) + \sum_{z' \neq z}\frac{\zeta^{(z')}}{\zeta^{(z)}}(\mu^{(z')} - \mu^{(z)})\Omega_{zz'}
\tag{A68}
$$

$$
\begin{aligned}
\dot{\Sigma}^{(z)} &= \mathbf{Q}^{(z)} - \frac{2}{\tau^{(z)}}\,\Sigma^{(z)} - \beta^2\,\Sigma^{(z)}\Gamma^{(z)}\,\Sigma^{(z)} + \\
&+ \sum_{z' \neq z}\frac{\zeta^{(z')}}{\zeta^{(z)}}\Omega_{zz'}[(\Sigma^{(z')} - \Sigma^{(z)}) + (\mu^{(z')} - \mu^{(z)})(\mu^{(z')} - \mu^{(z)})^{\mathrm{T}}]
\end{aligned}
\tag{A69}
$$

Once again note that in the continuous time limit the presynaptic spike trains are represented by the sum of Dirac-delta functions and we denote it with $\mathbf{s}(t)$ instead of $\mathbf{s}_t$.

In the case of two states ($+$ and $-$), we can write *Equation A67* as

$$
\dot{\zeta}^+ = -\zeta^+(1 - \zeta^+)(\overline{\gamma}^+ - \overline{\gamma}^-) + \zeta^+(1 - \zeta^+)\,\mathbf{s}(t)^{\mathrm{T}}\langle\Gamma\rangle^{-1}(\gamma^+ - \gamma^-) + (1 - \zeta^+)\,\Omega_+ - \zeta^+\Omega_-
\tag{A70}
$$

where we emphasise that the observation of either a spike or the absence of a spike causes changes in the state probabilities that are proportional to their uncertainty, $\zeta^+(1 - \zeta^+)$.

## A.8 Derivations of auxiliaries

A.8.1 Derivation of *Equations A26-A27*

After substituting *Equations A1-A2* and *Equation A9* into *Equation A6* we can derive the $\delta t$ dependence of the innovation term:

$$
\begin{aligned}
h(\delta t, \mathbf{u}, z) &= \mathrm{P}(\mathbf{u}_t = \mathbf{u}, z_t = z | \mathbf{s}_{0:t-\delta t}) \\
&= \sum_{z'} \mathrm{P}(z_t = z | z_{t-\delta t} = z') \; \mathrm{P}(z_{t-\delta t} = z' | \mathbf{s}_{0:t-\delta t}) \cdot \\
&\quad \cdot \int d\mathbf{u}' \mathrm{P}(\mathbf{u}_t = \mathbf{u} | \mathbf{u}_{t-\delta t} = \mathbf{u}', z_t = z) \; \mathrm{P}(\mathbf{u}_{t-\delta t} = \mathbf{u}' | z_{t-\delta t} = z', \mathbf{s}_{0:t-\delta t}) \\
&= \sum_{z'} \zeta_{t-\delta t}^{(z')} [\delta_{zz'}(1 - \delta t \overline{\Omega}_{z'}) + (1 - \delta_{zz'})\delta t \Omega_{zz'}] \cdot \\
&\quad \cdot \int d\mathbf{u}' \mathcal{N}\left( \mathbf{u}; \left(1 - \frac{\delta t}{\tau^{(z)}}\right) \mathbf{u}' + \frac{\delta t}{\tau^{(z)}} \overline{\mathbf{u}}^{(z)}, \delta t \mathbf{Q}^{(z)} \right) \mathcal{N}(\mathbf{u}'; \mu_{t-\delta t}^{(z')}, \mathbf{\Sigma}_{t-\delta t}^{(z')}) \\
&= \sum_{z'} \zeta_{t-\delta t}^{(z')} [\delta_{zz'}(1 - \delta t \overline{\Omega}_{z'}) + (1 - \delta_{zz'})\delta t \Omega_{zz'}] \; \frac{1}{1 - \frac{\delta t}{\tau^{(z)}}} \cdot \\
&\quad \cdot \int d\mathbf{u}' \mathcal{N}\left( \mathbf{u}'; \frac{\mathbf{u} - \frac{\delta t}{\tau^{(z)}} \overline{\mathbf{u}}^{(z)}}{1 - \frac{\delta t}{\tau^{(z)}}}, \frac{\delta t}{\left(1 - \frac{\delta t}{\tau^{(z)}}\right)^2} \mathbf{Q}^{(z)} \right) \mathcal{N}(\mathbf{u}'; \mu_{t-\delta t}^{(z')}, \mathbf{\Sigma}_{t-\delta t}^{(z')}) \\
&= \sum_{z'} \zeta_{t-\delta t}^{(z')} [\delta_{zz'}(1 - \delta t \overline{\Omega}_{z'}) + (1 - \delta_{zz'})\delta t \Omega_{zz'}] \; \frac{1}{1 - \frac{\delta t}{\tau^{(z)}}} \cdot \\
&\quad \cdot \mathcal{N}\left( \frac{\mathbf{u} - \frac{\delta t}{\tau^{(z)}} \overline{\mathbf{u}}^{(z)}}{1 - \frac{\delta t}{\tau^{(z)}}}; \mu_{t-\delta t}^{(z')}, \frac{\delta t}{\left(1 - \frac{\delta t}{\tau^{(z)}}\right)^2} \mathbf{Q}^{(z)} + \mathbf{\Sigma}_{t-\delta t}^{(z')} \right) \\
&= \sum_{z'} \zeta_{t-\delta t}^{(z')} [\delta_{zz'}(1 - \delta t \overline{\Omega}_{z'}) + (1 - \delta_{zz'})\delta t \Omega_{zz'}] \cdot \\
&\quad \cdot \mathcal{N}\left( \mathbf{u}; \left(1 - \frac{\delta t}{\tau^{(z)}}\right) \mu_{t-\delta t}^{(z')} + \frac{\delta t}{\tau^{(z)}} \overline{\mathbf{u}}^{(z)}, \delta t \, \mathbf{Q}^{(z)} + \left(1 - \frac{\delta t}{\tau^{(z)}}\right)^2 \mathbf{\Sigma}_{t-\delta t}^{(z')} \right)
\end{aligned}
\tag{A71}
$$

Substituting $\delta t = 0$ into **Equation A71** yields **Equation A26**.

Next, we take the derivative of **Equation A71** with respect to $\delta t$:

$$
\begin{aligned}
h'(\delta t, \mathbf{u}, z) &= \sum_{z'} \zeta_{t-\delta t}^{(z')} [(1 - \delta_{zz'})\Omega_{zz'} - \delta_{zz'} \overline{\Omega}_{z'}] \cdot \\
&\quad \cdot \mathcal{N}\left( \mathbf{u}; \left(1 - \frac{\delta t}{\tau^{(z)}}\right) \mu_{t-\delta t}^{(z')} + \frac{\delta t}{\tau^{(z)}} \overline{\mathbf{u}}^{(z)}, \delta t \mathbf{Q}^{(z)} + \left(1 - \frac{\delta t}{\tau^{(z)}}\right)^2 \mathbf{\Sigma}_{t-\delta t}^{(z')} \right) + \\
&\quad + \zeta_{t-\delta t}^{(z')} [\delta_{zz'}(1 - \delta t \overline{\Omega}_{z'}) + (1 - \delta_{zz'})\delta t \Omega_{zz'}] \cdot \\
&\quad \cdot \left[ \mathcal{N}\left( \mathbf{u}; \mu_{t-\delta t}^{(z')} + \delta t \frac{\overline{\mathbf{u}}^{(z)} - \mu_{t-\delta t}^{(z')}}{\tau^{(z)}}, \mathbf{\Sigma}_{t-\delta t}^{(z')} + \delta t \left( \mathbf{Q}^{(z)} - \frac{2}{\tau^{(z)}} \mathbf{\Sigma}_{t-\delta t}^{(z')} \right) + \delta t^2 \frac{\mathbf{\Sigma}_{t-\delta t}^{(z')}}{\tau^{(z)^2}} \right) \right]'
\end{aligned}
\tag{A72}
$$

where the derivative of the last term is provided below in general form.

$$[\mathcal{N}(\mathbf{u}; \mathbf{a}_0 + \delta t \mathbf{a}_1, \mathbf{B}_0 + \delta t \mathbf{B}_1 + \delta t^2 \mathbf{B}_2)]' =$$

$$= [\frac{1}{\sqrt{(2\pi)^D |\mathbf{B}_0 + \delta t \mathbf{B}_1 + \delta t^2 \mathbf{B}_2|}} e^{-\frac{1}{2}(\mathbf{u} - \mathbf{a}_0 - \delta t \mathbf{a}_1)^T (\mathbf{B}_0 + \delta t \mathbf{B}_1 + \delta t^2 \mathbf{B}_2)^{-1}(\mathbf{u} - \mathbf{a}_0 - \delta t \mathbf{a}_1)}]'$$

$$= [\frac{1}{\sqrt{(2\pi)^D |\mathbf{B}_0 + \delta t \mathbf{B}_1 + \delta t^2 \mathbf{B}_2|}}]' e^{-\frac{1}{2}(\mathbf{u} - \mathbf{a}_0 - \delta t \mathbf{a}_1)^T (\mathbf{B}_0 + \delta t \mathbf{B}_1 + \delta t^2 \mathbf{B}_2)^{-1}(\mathbf{u} - \mathbf{a}_0 - \delta t \mathbf{a}_1)} +$$

$$+ \frac{1}{\sqrt{(2\pi)^D |\mathbf{B}_0 + \delta t \mathbf{B}_1 + \delta t^2 \mathbf{B}_2|}} [e^{-\frac{1}{2}(\mathbf{u} - \mathbf{a}_0 - \delta t \mathbf{a}_1)^T (\mathbf{B}_0 + \delta t \mathbf{B}_1 + \delta t^2 \mathbf{B}_2)^{-1}(\mathbf{u} - \mathbf{a}_0 - \delta t \mathbf{a}_1)}]'$$

$$= -\frac{1}{2} \mathcal{N}(\mathbf{u}; \mathbf{a}_0 + \delta t \mathbf{a}_1, \mathbf{B}_0 + \delta t \mathbf{B}_1 + \delta t^2 \mathbf{B}_2) \cdot$$

$$[\text{In}'(|\mathbf{B}_0 + \delta t \mathbf{B}_1 + \delta t^2 \mathbf{B}_2|)$$

$$+ [(\mathbf{u} - \mathbf{a}_0 - \delta t \mathbf{a}_1)^T (\mathbf{B}_0 + \delta t \mathbf{B}_1 + \delta t^2 \mathbf{B}_2)^{-1}(\mathbf{u} - \mathbf{a}_0 - \delta t \mathbf{a}_1)]']$$

$$= -\frac{1}{2} \mathcal{N}(\mathbf{u}; \mathbf{a}_0 + \delta t \mathbf{a}_1, \mathbf{B}_0 + \delta t \mathbf{B}_1 + \delta t^2 \mathbf{B}_2) \cdot$$

$$\cdot [\text{Tr}\left((\mathbf{B}_0 + \delta t \mathbf{B}_1 + \delta t^2 \mathbf{B}_2)^{-1}(\mathbf{B}_1 + 2\delta t \mathbf{B}_2)\right) +$$

$$- \mathbf{a}_1^T (\mathbf{B}_0 + \delta t \mathbf{B}_1 + \delta t^2 \mathbf{B}_2)^{-1}(\mathbf{u} - \mathbf{a}_0 - \delta t \mathbf{a}_1) +$$

$$- (\mathbf{u} - \mathbf{a}_0 - \delta t \mathbf{a}_1)^T (\mathbf{B}_0 + \delta t \mathbf{B}_1 + \delta t^2 \mathbf{B}_2)^{-1} \mathbf{a}_1 -$$

$$- (\mathbf{u} - \mathbf{a}_0 - \delta t \mathbf{a}_1)^T (\mathbf{B}_0 + \delta t \mathbf{B}_1 + \delta t^2 \mathbf{B}_2)^{-1}$$

$$(\mathbf{B}_1 + 2\delta t \mathbf{B}_2)(\mathbf{B}_0 + \delta t \mathbf{B}_1 + \delta t^2 \mathbf{B}_2)^{-1}(\mathbf{u} - \mathbf{a}_0 - \delta t \mathbf{a}_1)]$$

(A73)

$$[\mathcal{N}(\mathbf{u}; \mathbf{a}_0 + \delta t \mathbf{a}_1, \mathbf{B}_0 + \delta t \mathbf{B}_1 + \delta t^2 \mathbf{B}_2)]'|_{\delta t = 0} =$$

$$= -\frac{1}{2} \mathcal{N}(\mathbf{u}; \mathbf{a}_0, \mathbf{B}_0)[\text{Tr}(\mathbf{B}_0^{-1} \mathbf{B}_1) - 2\mathbf{a}_1^T \mathbf{B}_0^{-1}(\mathbf{u} - \mathbf{a}_0)$$

$$- (\mathbf{u} - \mathbf{a}_0)^T \mathbf{B}_0^{-1} \mathbf{B}_1 \mathbf{B}_0^{-1}(\mathbf{u} - \mathbf{a}_0)]$$

(A74)

Substituting $\delta t = 0$ into **Equation A72** after using **Equation A74** to express its last term yields **Equation A27**.

### A.8.2 Derivation for **Equations A41-A43**

The above result can be used to compute terms appearing in the integrals of **Equations A22-A25** which are involved in computing **Equations A41-A43**

$$\int [\mathcal{N}(\mathbf{u}; \mathbf{a}_0 + \delta t \mathbf{a}_1, \mathbf{B}_0 + \delta t \mathbf{B}_1 + \delta t^2 \mathbf{B}_2)]'|_{\delta t = 0} d\mathbf{u} =$$

$$- \frac{1}{2} \int \mathcal{N}(\mathbf{u}; \mathbf{a}_0, \mathbf{B}_0) [\text{Tr}(\mathbf{B}_0^{-1} \mathbf{B}_1) - 2\mathbf{a}_1^T \mathbf{B}_0^{-1}(\mathbf{u} - \mathbf{a}_0)$$

$$- (\mathbf{u} - \mathbf{a}_0)^T \mathbf{B}_0^{-1} \mathbf{B}_1 \mathbf{B}_0^{-1}(\mathbf{u} - \mathbf{a}_0)] d\mathbf{u} =$$

$$- \frac{1}{2}[\text{Tr}(\mathbf{B}_0^{-1} \mathbf{B}_1) - \text{Tr}(\mathbf{B}_0^{-1} \mathbf{B}_1 \mathbf{B}_0^{-1} \mathbf{B}_0)] = 0$$

(A75)

$$\int \mathbf{u} \ [\mathcal{N}(\mathbf{u}; \mathbf{a}_0 + \delta t \mathbf{a}_1, \mathbf{B}_0 + \delta t \mathbf{B}_1 + \delta t^2 \mathbf{B}_2)]'|_{\delta t = 0} d\mathbf{u} =$$

$$= -\frac{1}{2} \int \mathcal{N}(\mathbf{u}; \mathbf{a}_0, \mathbf{B}_0) \mathbf{u} [\text{Tr}(\mathbf{B}_0^{-1} \mathbf{B}_1) - 2\mathbf{a}_1^T \mathbf{B}_0^{-1}(\mathbf{u} - \mathbf{a}_0)$$

$$- (\mathbf{u} - \mathbf{a}_0)^T \mathbf{B}_0^{-1} \mathbf{B}_1 \mathbf{B}_0^{-1}(\mathbf{u} - \mathbf{a}_0)] d\mathbf{u} =$$

(A76)

$$= -\frac{1}{2}\Big[\mathbf{a}_0\mathrm{Tr}(\mathbf{B}_0^{-1}\mathbf{B}_1) - 2\Big(\int N(u;\,a_0,\mathbf{B}_0) + u\mathbf{a}_1^{\mathrm{T}}\mathbf{B}_0^{-1}\mathbf{u}$$

$$-\mathbf{u}\mathbf{a}_1^{\mathrm{T}}\mathbf{B}_0^{-1}\mathbf{a}_0\,du\Big)$$

$$-\int N\Big(\mathbf{u};\,\mathbf{a}_0,\mathbf{B}_0\Big)(\mathbf{u}\mathbf{u}^{\mathrm{T}}\mathbf{B}_0^{-1}\mathbf{B}_1\mathbf{B}_0^{-1}\mathbf{u} - 2\mathbf{u}\mathbf{a}_0^{\mathrm{T}}\mathbf{B}_0^{-1}\mathbf{B}_1\mathbf{B}_0^{-1}\mathbf{u}$$

$$+\mathbf{u}\mathbf{a}_0^{\mathrm{T}}\mathbf{B}_0^{-1}\mathbf{B}_1\mathbf{B}_0^{-1}\mathbf{a}_0)\,d\mathbf{u}\Big] \tag{A77}$$

$$= -\frac{1}{2}\Big[\mathbf{a}_0\mathrm{Tr}(\mathbf{B}_0^{-1}\mathbf{B}_1) - 2\Big(\mathbf{B}_0^{-1}\mathbf{a}_1(\mathbf{B}_0 + \mathbf{a}_0\mathbf{a}_0^{\mathrm{T}}) - \mathbf{a}_0\mathbf{a}_1^{\mathrm{T}}\mathbf{B}_0^{-1}\mathbf{a}_0\Big) -$$

$$-\mathbf{a}_0\mathrm{Tr}(\mathbf{B}_0^{-1}\mathbf{B}_1) - \mathbf{B}_0\mathbf{B}_0^{-1}\mathbf{B}_1\mathbf{B}_0^{-1}\mathbf{a}_0 - (\mathbf{B}_0 + \mathbf{a}_0\mathbf{a}_0^{\mathrm{T}})\mathbf{B}_0^{-1}\mathbf{B}_1\mathbf{B}_0^{-1}\mathbf{a}_0 +$$

$$+2(\mathbf{B}_0 + \mathbf{a}_0\mathbf{a}_0^{\mathrm{T}})\mathbf{B}_0^{-1}\mathbf{B}_1\mathbf{B}_0^{-1}\mathbf{a}_0 - \mathbf{a}_0\mathbf{a}_0^{\mathrm{T}}\mathbf{B}_0^{-1}\mathbf{B}_1\mathbf{B}_0^{-1}\mathbf{a}_0\Big] \tag{A78}$$

$$= \mathbf{a}_1 \tag{A79}$$

$$\int \mathbf{u}\ \left[\mathcal{N}(\mathbf{u};\mathbf{a}_0 + \delta t\mathbf{a}_1, \mathbf{B}_0 + \delta t\mathbf{B}_1 + \delta t^2\mathbf{B}_2)\right]'\Big|_{\delta t=0}\mathbf{u}^{\mathrm{T}}d\mathbf{u}$$

$$= -\frac{1}{2}\int\ \mathcal{N}(\mathbf{u};\mathbf{a}_0,\mathbf{B}_0)\mathbf{u}[\mathrm{Tr}(\mathbf{B}_0^{-1}\mathbf{B}_1) - 2\mathbf{a}_1^{\mathrm{T}}\mathbf{B}_0^{-1}(\mathbf{u} - \mathbf{a}_0)$$

$$-(\mathbf{u} - \mathbf{a}_0)^{\mathrm{T}}\mathbf{B}_0^{-1}\mathbf{B}_1\mathbf{B}_0^{-1}(\mathbf{u} - \mathbf{a}_0)]\mathbf{u}^{\mathrm{T}}d\mathbf{u} \tag{A80}$$

$$= -\frac{1}{2}\Big[(\mathbf{B}_0 + \mathbf{a}_0\mathbf{a}_0^{\mathrm{T}})\ \mathrm{Tr}(\mathbf{B}_0^{-1}\mathbf{B}_1) -$$

$$-2\int\mathcal{N}(\mathbf{u};\mathbf{a}_0,\mathbf{B}_0)(\mathbf{u}\mathbf{a}_1^{\mathrm{T}}\mathbf{B}_0^{-1}\mathbf{u}\mathbf{u}^{\mathrm{T}} - \mathbf{u}\mathbf{a}_1^{\mathrm{T}}\mathbf{B}_0^{-1}\mathbf{a}_0\mathbf{u}^{\mathrm{T}})d\mathbf{u} -$$

$$-\int\mathcal{N}(\mathbf{u};\mathbf{a}_0,\mathbf{B}_0)(\mathbf{u}\mathbf{u}^{\mathrm{T}}\mathbf{B}_0^{-1}\mathbf{B}_1\mathbf{B}_0^{-1}\mathbf{u}\mathbf{u}^{\mathrm{T}} - 2\mathbf{u}\mathbf{a}_0^{\mathrm{T}}\mathbf{B}_0^{-1}\mathbf{B}_1\mathbf{B}_0^{-1}\mathbf{u}\mathbf{u}^{\mathrm{T}}$$

$$+\mathbf{u}\mathbf{a}_0^{\mathrm{T}}\mathbf{B}_0^{-1}\mathbf{B}_1\mathbf{B}_0^{-1}\mathbf{a}_0\mathbf{u}^{\mathrm{T}})\Big] \tag{A81}$$

$$= -\frac{1}{2}\Big[(\mathbf{B}_0 + \mathbf{a}_0\mathbf{a}_0^{\mathrm{T}})\ \mathrm{Tr}(\mathbf{B}_0^{-1}\mathbf{B}_1) -$$

$$-2\Big(\mathbf{a}_0\mathbf{a}_1^{\mathrm{T}}\mathbf{B}_0^{-1}(\mathbf{B}_0 + \mathbf{a}_0\mathbf{a}_0^{\mathrm{T}}) + (\mathbf{B}_0 + \mathbf{a}_0\mathbf{a}_0^{\mathrm{T}})\mathbf{B}_0^{-1}\mathbf{a}_1\mathbf{a}_0^{\mathrm{T}} + \mathbf{a}_1^{\mathrm{T}}\mathbf{B}_0^{-1}\mathbf{a}_0(\mathbf{B}_0 - \mathbf{a}_0\mathbf{a}_0^{\mathrm{T}})$$

$$-\mathbf{a}_1^{\mathrm{T}}\mathbf{B}_0^{-1}\mathbf{a}_0(\mathbf{B}_0 + \mathbf{a}_0\mathbf{a}_0^{\mathrm{T}})\Big) -$$

$$-\Big(2(\mathbf{B}_0 + \mathbf{a}_0\mathbf{a}_0^{\mathrm{T}})\ \mathbf{B}_0^{-1}\mathbf{B}_1\mathbf{B}_0^{-1}(\mathbf{B}_0 + \mathbf{a}_0\mathbf{a}_0^{\mathrm{T}}) + \mathbf{a}_0^{\mathrm{T}}\mathbf{B}_0^{-1}\mathbf{B}_1\mathbf{B}_0^{-1}\mathbf{a}_0(\mathbf{B}_0 - \mathbf{a}_0\mathbf{a}_0^{\mathrm{T}})$$

$$+(\mathbf{B}_0 + \mathbf{a}_0\mathbf{a}_0^{\mathrm{T}})\ \mathrm{Tr}(\mathbf{B}_0^{-1}\mathbf{B}_1)\Big) +$$

$$+2\Big(\mathbf{a}_0\mathbf{a}_0^{\mathrm{T}}\mathbf{B}_0^{-1}\mathbf{B}_1\mathbf{B}_0^{-1}(\mathbf{B}_0 + \mathbf{a}_0\mathbf{a}_0^{\mathrm{T}}) + (\mathbf{B}_0 + \mathbf{a}_0\mathbf{a}_0^{\mathrm{T}})\mathbf{B}_0^{-1}\mathbf{B}_1\mathbf{B}_0^{-1}\mathbf{a}_0\mathbf{a}_0^{\mathrm{T}}$$

$$+\mathbf{a}_0^{\mathrm{T}}\mathbf{B}_0^{-1}\mathbf{B}_1\mathbf{B}_0^{-1}\mathbf{a}_0(\mathbf{B}_0 - \mathbf{a}_0\mathbf{a}_0^{\mathrm{T}})\Big) -$$

$$-\mathbf{a}_0^{\mathrm{T}}\mathbf{B}_0^{-1}\mathbf{B}_1\mathbf{B}_0^{-1}\mathbf{a}_0(\mathbf{B}_0 + \mathbf{a}_0\mathbf{a}_0^{\mathrm{T}})\Big] \tag{A82}$$

$$= -\frac{1}{2}[-2(\mathbf{a}_0\mathbf{a}_1^{\mathrm{T}} + \mathbf{a}_1\mathbf{a}_0^{\mathrm{T}}) - 2\mathbf{B}_1] = \mathbf{B}_1 + \mathbf{a}_0\mathbf{a}_1^{\mathrm{T}} + \mathbf{a}_1\mathbf{a}_0^{\mathrm{T}} \tag{A83}$$

# B The optimality of the sigmoidal nonlinearity

Neurons need to combine incoming spikes nonlinearly to perform efficient analog computations based on digital spikes. In the main text, and the previous sections of this Appendix, we derived the optimal way of combining incoming spikes (*Equations 20-22*), but the exact implementation of this optimal nonlinear mapping within the dendritic tree is unrealistic. A plausible alternative would be if each dendrite branch acted as a simple linear-nonlinear unit, computing a nonlinear function of the linear combination of its linearly filtered input spike trains (*Poirazi et al., 2003*), but it is unclear whether this could efficiently approximate the optimal response, and if so then what is the specific functional form of the nonlinearity that is able to make this approximation tight. In the main paper, we demonstrated that a sigmoidal nonlinearity approximates the optimal response remarkably well if the dynamics of the presynaptic population is dominated by simultaneous switching between a quiescent and an active state. In this section we provide additional details for this derivation.

## B.1 Simplified presynaptic dynamics

In the following derivations, for convenience, we assume that the weights of the linear computation are roughly uniform and scale inversely with $N$, $w_i \propto^{\sim} 1/N$, although our derivations also remain valid with other scalings of $w_i$ as long as the number of weights significantly greater than $0$ scales with $N$. We start by noting that the optimal response can always be trivially rewritten as requiring the inference of the state, $z(t)$, and conditioned on the state, the average presynaptic membrane potential directly, $m(t) = \sum_i w_i u_i(t) = 1/N \sum_i u_i(t)$ (instead of the individual membrane potentials, as in *Equations 3 and 6*):

$$\tilde{v}(t) = \sum_z \mathrm{P}\Big(z(t) = z | \mathbf{s}(0:t)\Big) \int \mathrm{P}\Big(m(t) = m | z(t) = z, \mathbf{s}(0:t)\Big) \; m \, \mathrm{d}m \tag{A84}$$

$$= \sum_z \zeta^{(z)}(t) \int \mathrm{P}\Big(m(t) = m | z(t) = z, \mathbf{s}(0:t)\Big) \; m \, \mathrm{d}m \tag{A85}$$

where, for simplicity, we took $\tau_{\mathrm{post}} = 0$, although it is not essential for our argument – which instead applies to the total input when $\tau_{\mathrm{post}} \neq 0$. Note that we do not generally use this seemingly simpler form because the likelihood describing the spike generation process (*Equation 8*) becomes more complex, and notably non-factorised, when conditioned on $m(t)$ instead of $\mathbf{u}(t)$. Nevertheless, it is instructive to write down the prior over $m$ (cf. *Equation 7*):

$$\mathrm{P}(m_t | m_{t-\delta t}, z_t = z) = \mathcal{N}\left(m_t; \left(1 - \frac{\delta t}{\tau}\right) \; m_{t-\delta t} + \frac{\delta t}{\tau} \overline{m}^{(z)}, \delta t \overline{q}^2\right) \tag{A86}$$

where $\overline{m}^{(z)} = \frac{1}{N} \sum_i^N \overline{u}_i^{(z)}$ is the mean and $\overline{q}^2 = \frac{1}{N^2} \sum_{ij} Q_{ij}$ is the process noise variance defined in terms of the parameters of the original mOU process. In the following sections, we derive simplified forms of inference, that can be readily related to dendritic processing, by taking different limits to the original model, and in particular *Equation A86*.

## B.2 Sigmoidal nonlinearity with state switching dynamics

We start by assuming that the presynaptic neurons are equally variable and conditionally independent given the population state ($Q_{i,j} = q^2\delta_{ij}$). It is easy to see that in this case $\overline{q}^2 = q^2/N$ and thus the prior variance of $m$ (which is $\overline{q}^2\tau/2$, as can be seen from **Equation A86**) also scales as $1/N$. Thus, for a large presynaptic population, $N \longrightarrow \infty$, the prior variance of $m$ diminishes which means that the posterior inference is also greatly simplified because there is no uncertainty any more about $m$ given the population state, i.e.

$\mathrm{P}\Big(m(t)|z(t) = z, \mathbf{s}(0:t)\Big) = \delta\Big(m(t) - \overline{m}^{(z)}\Big)$, and so the optimal response (**Equations A85**) for two population states simplifies to

$$\tilde{\mathrm{v}}(t) = \zeta(t)\ \overline{m}^+ + (1 - \zeta(t))\overline{m}^-  \tag{A87}$$

Although it seems that the only quantity that needs updating in **Equation A87** is $\zeta(t)$, in general this still requires keeping track of the membrane potentials of individual presynaptic neurons because the dynamics of $\zeta(t)$ (**Equation 20**) depends on the individual expected firing rates (conditioned on each population state). Therefore, to allow further simplification, we assume that all expected firing rates within a state are identical and constant, equal to their prior value, i.e. $\gamma_i^+ \approx \gamma^+ = g e^{\beta\overline{u}^+ + \frac{1}{2}\beta^2 q^2\tau/2}$, and so the population rate is simply $\overline{\gamma}^+ \approx N\gamma^+$ (with corresponding definitions for $\gamma^-$ and $\overline{\gamma}^-$). With these assumptions the dynamics of the posterior estimate of population state reduces to

$$\dot{\zeta} = \underbrace{\Omega_+ - (\Omega_+ + \Omega_-)\ \zeta}_{A} + \underbrace{\frac{\gamma^+ - \gamma^-}{\zeta\gamma^+ + (1-\zeta)\ \gamma^-}}_{B(\zeta)}\zeta(1-\zeta)\ s(t) - \underbrace{(\overline{\gamma}^+ - \overline{\gamma}^-)}_{C}\zeta(1-\zeta)  \tag{A88}$$

As we do not track the membrane potential fluctuations of the individual neurons in this reduced model, we also replaced the $\mathbf{s}(t)$ vector by the total spike train of the population, $s(t) = \sum_i s_i(t)$.

Amongst the parameters of **Equation A88** only $C$ scales with $N$, therefore the last term dominates **Equation A88** during silent periods. Conversely, the second term (involving $B$) will dominate during spikes. Thus, the first term (involving $A$) can be ignored. Finally, we note that

$$B(\zeta) = \frac{1}{\zeta + \frac{\gamma^-}{\gamma^+ - \gamma^-}} = \frac{1}{\zeta + \frac{1}{e^{\beta(\overline{u}^+ - \overline{u}^-)} - 1}}  \tag{A89}$$

and therefore $B(\zeta)$ is nearly constant if the difference between $\gamma^+$ and $\gamma^-$ is small (i.e. $\beta \ll \frac{\ln 2}{\overline{u}^+ - \overline{u}^-}$), in which case

$$B \simeq e^{\beta(\overline{u}^+ - \overline{u}^-)} \ll 1  \tag{A90}$$

These considerations allow us to write the approximate form of **Equation A88** as (cf. **Equation 27**):

$$\dot{\zeta} = \zeta(1-\zeta)\ [Bs(t) - C]  \tag{A91}$$

Solving **Equation A91** for $\zeta(t)$ is possible by noting that it is separable, and so the solution can be written as

$$\int \frac{1}{\zeta(1-\zeta)} d\zeta = \int [Bs(t) - C] \mathrm{d}t  \tag{A92}$$

where solving the integral on the left yields

$$\ln\left(\frac{\zeta}{1-\zeta}\right) + \chi = \int [Bs(t) - C]\mathrm{d}t \tag{A93}$$

In general, the value of $\chi$ would need to be determined by solving the initial value problem, but for simplicity and without loss of generality here we take $\chi = 0$ and assume that $\nu(0)$ has the appropriate value to map to the required $\zeta(0)$.

From **Equation A93**, $\zeta$ can be expressed as

$$\zeta = \frac{1}{1 + e^{-\int [Bs(t) - C]\mathrm{d}t}} \tag{A95}$$

which in turn can be rewritten as

$$\zeta = \frac{1}{1 + e^{-\nu}} \tag{A96}$$

where we introduced the new variable $\nu = \int [Bs(t) - C]\mathrm{d}t$ whose definition can be equally written as a differential equation:

$$\dot{\nu} = Bs(t) - C \tag{A97}$$

The system defined by **Equations A96-A97** has a form that is closely analogous to that of the standard linear-nonlinear model of dendritic processing with a logistic sigmoid nonlinearity, and is presented in the main text (Methods) as **Equations 28-29**.

## B.3 Sublinear integration with second order correlations

We also considered another limiting case when the dynamics of the presynaptic population was fully characterized by purely second-order correlations between the neurons. This means that there is no state switching dynamics, so inference is simplified by taking $\zeta = 1$, but we also cannot assume (as we did in the previous section) that the posterior correlations and thus the posterior variance of $m$ vanishes in the large $N$ limit. Instead, we seek a formalism in which the population activity of $N$ cells, represented by the variables $\{\mathbf{u}, \mathbf{s}\}$, is captured by the lower dimensional dynamics of $\{m, s\}$ (where $s(t) = \sum_i s_i(t)$ is still the 'total' input spike train, as before). Under this simplified model, the prior dynamics of $m$ is as described by **Equation A86**, and the likelihood is analogous to the one we had for individual neurons (**Equation 8**):

$$\mathrm{P}(s_t|m_t) = \mathrm{Poisson}(s_t; \delta t\, r_t), \quad \text{with } r_t = \overline{g}\, e^{\overline{\beta}\, u_t} \tag{A98}$$

As the likelihoods of the two models can not be exactly matched, we chose the parameters $\overline{\beta}$ and $\overline{g}$ to match the mean and the variance of the population firing rate. Thus, there is now a single variable that characterizes the whole population, $m$, whose dynamics (both the prior and likelihood) are analogous to those we had for a single membrane potential in the previous model. Consequently, inference in this collapsed model requires the manipulation of only two scalar variables – instead of the original $\mathcal{O}(N^2)$ –, the posterior mean and variance of $m$, $\overline{\mu}$ and $\overline{\sigma}^2$ respectively, whose dynamics can be derived analogously to those obtained before for the full population (**Equations 21-22**):

$$\dot{\overline{\mu}} = \frac{\overline{m} - \overline{\mu}}{\tau} + \overline{\beta}\overline{\sigma}^2\left(s(t) - \overline{\gamma}\right) \tag{A99}$$

$$\dot{\overline{\sigma}}^2 = \overline{q}^2 - \frac{2}{\tau}\overline{\sigma}^2 - \overline{\beta}^2\overline{\sigma}^4\overline{\gamma} \tag{A100}$$

where, in line with previous notation, $\overline{\gamma} = \overline{g}e^{\overline{\beta}\overline{\mu}+\frac{1}{2}\overline{\beta}^2\overline{\sigma}^2}$ is the posterior mean estimate of the average firing rate of the presynaptic population. (These equations only apply to the special case when $w_i = 1/N$ and $Q_{ij} = \delta_{ij}q + (1 - \delta_{ij})\ \rho q$, but an extension to the more general case is relatively straightforward.) We used this reduced model to infer the mean membrane potential of the presynaptic population – which is relevant in the special case when $w_i = 1/N$ and $\tau_{\mathrm{post}} = 0$ (see above) so that $\tilde{v} = \overline{\mu}$. **Figure 3—figure supplement 1A–C** demonstrates that our reduced model approximates the full model remarkably well provided that there are substantial correlations in the presynaptic population.

Next, we noted that during inference the dynamics of the posterior variance is faster than the dynamics of the posterior mean (**Equations A99-A100**) and therefore the posterior variance can be approximated by its (mean-dependent) steady state value, $\overline{\sigma}^2_\infty(\overline{\mu})$. If we replace the posterior variance in **Equation A100** with its steady state value we obtain the following approximate one-dimensional dynamics:

$$\dot{\overline{\mu}} \simeq \frac{\overline{m} - \overline{\mu}}{\tau} + \overline{\beta}\overline{\sigma}^2_\infty(\overline{\mu})\ \left(s(t) - \overline{g}e^{\overline{\beta}\overline{\mu}+\frac{1}{2}\overline{\beta}^2\overline{\sigma}^2_\infty(\overline{\mu})}\right) \tag{A101}$$

As the optimal response to a single spike is proportional to the posterior variance, the functional form of the steady state is highly informative about the nonlinearity of the integration. **Figure 3—figure supplement 1D** shows that $\overline{\sigma}^2_\infty(\overline{\mu})$ is a decreasing sigmoidal function, thus responses to consecutive spikes will add sublinearly in this case, as each spike increases $\overline{\mu}$, which in turn reduces $\overline{\sigma}^2_\infty$, which results in a smaller response for the next spike.

Expressing **Equation A101** directly in the form of linear-nonlinear dynamics, as in the case of switching dynamics, did not seem feasible, so we focussed instead on establishing a correspondence only at the times of presynaptic spikes, when $s(t) > 0$. This required that there exists an instantaneous mapping $h(v_{\mathrm{lin}})$ such that

$$\overline{\mu} = h(v_{\mathrm{lin}}) \tag{A102}$$

where $v_{\mathrm{lin}}$ is the linearly filtered and integrated inputs, whose dynamics $\dot{v_{lin}}$ is given by **Equation 24**. Differentiating both sides of **Equation A102** wrt. time, and substituting **Equation A101** and **Equation 24**, respectively, to each side yields

$$\overline{\beta}\overline{\sigma}^2_\infty\left(h(v_{\mathrm{lin}})\right) = h^{'}(v_{\mathrm{lin}})\ \mathscr{B} \tag{A103}$$

after only keeping the dominating terms including $s(t)$ on both sides and subsequently simplifying by $s(t)$.

Although there was no closed-form analytical solution for $h(v_{\mathrm{lin}})$ from **Equation A103**, it could be obtained by numerical integration (**Figure 3—figure supplement 1E**, blue line). It is clear that the optimal mapping $h$ is indeed sublinear in this case and it can be reasonably well approximated using the concave half of the logistic sigmoid function derived above, for the switching dynamics, even during silent periods (**Figure 3—figure supplement 1E**, red line versus black dots; **Figure 3**). There is a simple intuition for the sublinearity implied by $h$ (following on from the intuition we gave for the sublinearity of the response after **Equation A101**): according to **Equation A103**, its derivative is proportional to the steady state posterior variance, which in turn is a non-negative monotonically decreasing function (**Figure 3—figure supplement 1D**), and hence the final non-linearity must be monotonically increasing and, importantly, concave – i.e. sublinear.

