## [Decision Letter]

[Editors’ note: a previous version of this study was rejected after peer review, but the authors submitted for reconsideration. The first decision letter after peer review is shown below.]

Thank you for choosing to send your work entitled "Dendritic nonlinearities are tuned for efficient spike-based computations in cortical circuits" for consideration at *eLife*. Your full submission has been evaluated by Eve Marder (Senior editor) and three peer reviewers, one of whom is a member of our Board of Reviewing Editors, and the decision was reached after very extensive discussions between the reviewers. Based on our discussions and the individual reviews below, we have decided not to consider the present manuscript further at this time. However, we would be willing to consider a significantly revised resubmission (below).

While the reviewers all felt that the work was quite interesting, concern was raised in the ensuing discussions as to the unclear presentation of the essence of the work. In fact, the reviewers spent a considerable time discussing what they thought was the structure of the model, and what it means. The fact that the reviewers went back and forth over these issues indicates a lack of clarity in the Results section. Therefore, it was decided that the work needed significant revisions and significant effort would be required to make the work accessible to the general reader, not only in making the take home message clear, but also in explaining how and why detailed mathematical derivations are needed and used.

Overall, two major aspects require consideration, as summarized below with some suggestions. Additional detailed comments from the reviewers are appended below.

1) A clearer and simpler version of the paper is needed.

A clear line through the reasoning that will allow everyone to see the core of what has been done, and why, needs to be presented. Specifically, the paper needs to be transparent and math that obfuscates should be removed or simplified or explained. The authors are encouraged to present their essential results in a much more straightforward fashion to help elucidate their mathematical derivations.

A suggestion is to use a flow chart of the process/protocol (capturing presynaptic firing statistics, fitting, optimizing etc., and lay bare the limitations/assumptions – e.g., the output computation *f*(u) is assumed to be a linear function), describe the essential results in words, and point to the mathematical parts in methods/or a mathematical appendix of how and why. We understand and appreciate that you wish to publish the full mathematical treatment, but most *eLife* readers won't benefit from the math in its present form. The mathematical results could be made complete and readable in an Appendix, while freeing the Results to show the logic of the model and how it works. It looks complicated because of using the most general formulations (multi-state state transitions, and multi-variate normal with arbitrary state-dependent covariance), but the main results are obtained via somewhat simpler formulations. More detailed mathematical derivations could go into a mathematical appendix if necessary.

In this way, people (of diverse backgrounds) would be able to sink their teeth into the contribution, build on it, test it and so on.

2) Take-home message(s) and biological/biophysical intuition needs to be presented.

No biophysical model seems to be included, even simply. Everything seems to be folded into the optimal g(s), so that it is not obvious how best to make a biological/biophysical connection.

The derived 'optimal traces' do, in principle, bring a more stringent test to the theory, but in Figure 4 they have access to the whole pre-synaptic dynamics so the optimal trace can be replaced by the averaged pre-synaptic potential, while in Figure 5 the experimental pattern of stimulation is not sufficiently complex to really probe the detailed temporal structure predicted by the theory. The other figures support qualitative statements. This, for example, could be explicitly stated and rationalized for the reader.

Some aspects came through but because of the first point above, there was not always a clear consensus of what the authors intended as the main message(s) from the work. Could these intents (if appropriate) be better explained?

For example:i) Presynaptic neurons with correlated inputs should target neighboring regions of the dendritic tree so that the inputs can sum nonlinearly while presynaptic neurons with uncorrelated inputs should map onto different portions of the tree and the inputs sum linearly?ii) An optimizing aspect/principle that could ultimately predict the 'location structure' of synapses as a crucial test of their theory.iii) Optimally decoding the information present in the spikes to do the desired computation depends upon the correlationsiv) Addressing the question of why there are nonlinearities in the first place.v) An optimal way to recover the averaged presynaptic potential only from the pre-synaptic spike trains? Can the biophysics approximate this optimal 'decoding' operation?

*Reviewer #1:*

In this most interesting study, the authors propose a principle that dendritic nonlinearity is optimized to integrate synaptic inputs from the statistics of the presynaptic cells. This principle assumes that communication is mainly via spikes whereas computation is not 'punctate', thus identifying the bottleneck of dendritic processing. They go on to use models (clustered connectivity prediction) and experiment to show that this principle is followed.

The authors may have uncovered a fundamental principle, and as they state in the Discussion, "Once patterned dendritic stimulation over a broader and more realistic range of inputs becomes feasible, our theory will provide a principled method for dissecting the roles of presynaptic correlations vs. genuine nonlinear computations in shaping dendritic nonlinearities." Nice I think.

I have a few comments.

1) The authors could state that graded synaptic transmission exists (many examples in invertebrates), so that their principle may not apply there, I assume. Is it definitively known that graded synaptic transmission is not present in cortical circuits?

2) It would be helpful if the authors could expand on the discussion regarding biophysical substrate. That is, to fill in the blanks on how and why "NMDA spikes may provide a general solution". Was some modeling (not shown) done to be able to state this? Please provide some additional discussion/rationale/assumptions for making these statements.

3) Regarding inhibition, the authors discuss potential addition from the perspective of excitatory cells. However, perspectives from inhibitory cells are not expressed. Do the authors expect their principle to hold from the perspective of inhibitory cells (receiving excitatory and/or inhibitory inputs)? The authors should provide some discussion/rationale/assumptions of why this is not included/considered. I do not think it is a given that computation is only from the perspective of excitatory cells? For example, consider the reviews by Klausberger and Somogyi (Science 2008) and Chamberland and Topolnik (Frontiers in Neuroscience 2012).

*Reviewer #2:*

Nonlinear dendritic integration is often seen as a thorn in the foot. Following in vitro work by Larkum, Magee, Stuart, Hausser and others, it is now clear that dendrites introduce various types of nonlinearity to input integration. Further observations showed physiological patterns that appeared to tune specifically the dendritic nonlinearities. Computations that are carried out in the dendrites (as in Taylor et al. 2000) were then discovered. Thus, it has become increasingly difficult to ignore their computational role. The search for a theoretical role (mainly by Mel and coworkers), on the other hand, was mainly focused on rate-based computations. Yet, rate and spiking descriptions are often qualitatively different (as in plasticity or synfire chains). Understanding the key mechanisms for spike-based dendritic computations remains a fundamental problem, but a very difficult one.

In this article, Ujfalussy and coworkers address the hypothesis that nonlinear dendritic integration serves to decode the correlated pre-synaptic spike trains. They describe mainly three observations that back up this hypothesis. First, they show mathematically that the canonical model of nonlinear integration of Poirazi and Mel (P&M) approximates an optimal decoding of the switching between active and quiescent states. Second, they compared how different biophysically detailed models approximated an optimal decoding strategy. They found that synaptic input clustered on nonlinear dendrites performed better at this task than when the same input was lumped in the soma, or synapses randomly assigned to dendrites. Last, they show that in vitro measurements of dendritic integration in a specific cell type matched the optimal integration strategy expected to decode the input statistics of that particular cell in vivo. These are the three main observations, but a particular strength of this article is that it draws a clear top-down rhetoric and provides general estimates of (bayes) optimal spiking communication.

Although I don't subscribe to all the assumptions described here, they are sensible and worth a careful examination. Ujfalussy and coworkers offer a thorough and elegant treatment of the question relying substantially on most of the state-of-the-art experimental recordings relevant to this problem. The conclusions are well tested, often with very stringent criteria. The relevant literature is properly cited. The article is generally well written.

The main contributions will appeal to researchers coming from different approaches. The mathematical neuroscientist will respect the relation between the P&M model and optimal decoding. Those considering decoding spike train from a bayesian perspective will check the derivations in the supplementary material. Others coming from the literature on noise-correlation will appreciate that, with dendrites, some of the reasonings should be revised. Finally, the new theoretical role for dendritic nonlinearities can clearly be tested with present experimental techniques. For these reasons, I believe that even if the supposed role is later shown to be false or incomplete, the article will have had a strong impact on the community.

*Reviewer #2 (Minor Comments):*

Generally, confusion arises in the number of models considered. I think that, when a comparison between models is the main result, these different models should be mentioned at the beginning of the section. For instance in the last paragraph of the section “The form of the optimal nonlinearity depends on the statistics of presynaptic inputs” one suddenly learns of the actual comparisons being made.

On a first reading, I could not understand what was done in the first paragraph of the section” Nonlinear integration in cortical neurons is matched to their input statistics”. After reading the supplementary, it is clear, but I think a few sentences should be added to describe in a little more details what was actually compared. For instance, one can read from the first sentence that to test the theory, we need to measure presynaptic statistics and derive the optimal nonlinearity, then measure the nonlinearity in vitro and compare the nonlinearities. That is not what was done, I know now, but I could not rule that out from the paragraph as is. It is also difficult to understand the end of second paragraph of the same section on the first go.

What is the experimental stimulus protocol used in the uncaging experiments? How broad is the distribution of ISI? Evenly distributed on Figure 5 and g? Are each burst made of evenly spaced pre-synaptic encaging? Can you really say overfitting is small?

Why exactly the multi-state description in Eq. 11 and 19 if it is never used? Personally, it is only when I saw Eq. S67 that I finally reached the proper mathematical intuitions. Other than the need for simplicity, I am concerned that some of the conclusions would not hold in the multi-state scenario suggested by Eq. 11.

Figure 1 is argued to offer a better fit to optimal than 1C. But there are two differences, correlation and scale. Could it be that in the uncorrelated case, the fit to optimal is not as good when the potential fluctuates on greater amplitudes? In fact, I don't understand why the uncorrelated case should not have a saturating dendritic nonlinearity to counteract the exponential spiking nonlinearity."

*Reviewer #3:*

The authors present a detailed mathematical analysis, computations and experimental results aimed at supporting the conclusion that dendritic nonlinearities used to integrate synaptic inputs in a neuron are optimized based upon the statistics of that neuron's presynaptic connections. In particular, the suggested result is that if a neuron's presynaptic connections are uncorrelated, sublinear responses are optimal. If, however, the firing of presynaptic neurons are correlated, supralinear responses are optimal for that neuron. If correct, I would find this conclusion to be interesting.

One of the main difficulties with the manuscript is that the details are not easily approachable. There are many pages of complicated mathematics that one must work through, and in many places there hasn't been much attempt to try to explain things in simpler terms to the general reader – I suspect that one will not be able to follow the details here unless one is already an expert.

While the major focus on presynaptic activity is in terms of spikes (and I will grant that a good explanation for this is given), I think it's a concern is that the focus on postsynaptic activity is in terms of subthreshold responses. Granted, the authors say that they can also perform their calculation if the firing rate is substituted for the subthreshold response. This seems like an inconsistency, however – if two or more neurons share synaptic inputs from overlapping sets of presynaptic neurons, and the correlations in the spiking patterns of the presynaptic neurons matter, why is it that correlations in the output spikes of those two or more neurons don't matter? This would seem to be a basic internal inconsistency in the manuscript.

In addition, while there is a huge amount of detail associated with the mathematical calculations, I found the details associated with the experiments to be a bit too sparse. It would have been useful, for example, to see an image of the neocortical pyramidal cell and the specific synapses which are being stimulated.

[Editors’ note: what now follows is the decision letter after the authors submitted for further consideration.]

Thank you for submitting your work entitled "Dendritic nonlinearities are tuned for efficient spike-based computations in cortical circuits" for peer review at *eLife*. Your submission has been favorably evaluated by Eve Marder (Senior Editor) and three reviewers, one of whom is a member of our Board of Reviewing Editors.

The reviewers have discussed the reviews with one another and the Reviewing editor has drafted this decision to help you prepare a revised submission.

All of the reviewers are strongly positive about this work but they also want to ensure that the work is accessible, understandable and widely appreciated. They would like all aspects be made as clear as possible. Thus, pending the following revisions, we expect to accept the paper.

Please respond directly to the revisions on a point-by-point basis, revising the paper where needed.

1) Double check all equations in the mathematical appendix – there was some concern given potential sign errors in other equations (see below). Additional explanatory guidance through the mathematical steps should be provided for those readers who may be interested in following these details. The math is still very complicated.

2) Check and correct potential inconsistencies related to the description of the function s(*t*). In the subsection “Presynaptic statistics” it is still stated that in the limit in which the time step goes to zero the discrete version becomes binary, i.e., zero or 1. In the subsection “Inference and the optimal response”, however it's stated that s(*t*) is a sum of Dirac-delta functions. The two descriptions can't be consistent, of course, unless there is a division by a time step somewhere. There does not appear to be a careful definition in the manuscript, however.

3) Check the solution of equation (22) given in (23) and (24) for potential sign errors.

4) The additional details about the experiments were appreciated. However, a question arose about the blocking of NMDA channels. It's stated that substantially poorer fits were obtained for this case. Please confirm/clarify whether the postsynaptic filtering parameters were re-fit in this case?

5) It seems like a bit of an inconsistency that while a big point is made that presynaptic neurons must communicate with spikes, no spiking is considered in the postsynaptic neurons. If spiking is the only way that neurons can communicate (the authors' point), why should one care about the subthreshold response of the postsynaptic neuron? Since so much of the manuscript covers the subthreshold response, something explicit (explanation-wise) should be said about this in the manuscript (and we realize that they consider the case of firing rates).

6) Include a 'toy model' explanation (without math, or if possible, with straightforward enough math equations that most can follow) that summarizes the link between pairwise correlation (temporal) and dendritic integration (spatial) before any specific statistical models are discussed. This would help the reader follow the more detailed case, and avoid confusion in consideration of later statements in the paper. A good place for this might be at the beginning of the Results section.

Specifically, the way the text reads is confusing about when *spatial* correlations (for lack of a better term) between different presynaptic neurons are being discussed, and when *temporal* correlations between inputs are being considered. Part of the issue may be that the mathematical formulation in the main text completely hides issues of temporal correlation in the details. Thus one can be left wondering if one of the two types of correlations might be more important and if the theory could shed light on this.

For example, consider these comments from the manuscript:

"However, if presynaptic neurons became correlated the optimal response became nonlinear and the best linear response was unable to accurately follow the fluctuations in the input (Figure 1)."

"Same as (A-C) but for presynaptic neurons exhibiting synchronized switches between a quiescent and an active state (D, bottom), introducing higher order correlations between the neurons." (Figure 2 caption)

"In particular, sublinear integration was optimal when presynaptic activities exhibited simple Gaussian random walks and thus they could not contain statistical dependencies beyond second order correlations (Figure 2)."

The authors here are trying to provide an intuitive explanation as to why input correlations lead to the observed behavior and this, of course, is good and to be encouraged. In each case, however, what type of correlation is being thought of – spatial or temporal? Further elaboration might provide additional insight.

[Editors' note: further revisions were requested prior to acceptance, as described below.]

Thank you for submitting your work entitled "Dendritic nonlinearities are tuned for efficient spike-based computations in cortical circuits" for consideration by *eLife*. Your article has been reviewed by three peer reviewers, one of whom is a member of our Board of Reviewing Editors, and the evaluation has been overseen by Eve Marder as the Senior Editor.

The reviewers have discussed the reviews with one another and the Reviewing editor has drafted this decision to help you prepare a revised submission.

The reviewers appreciated the revisions done by the authors, but it was felt that the intuition as provided was incomplete and the mathematical appendix was still not clearly presented. Specifically, it was noted that is unclear how the authors distinguish autocorrelation from cross-correlation (spatial) in the formalism and the text, and as this is a main point of the paper, it should be up front early on in the Discussion (and not just in caption of Figure 1 and later in Methods). Also, eq A91 has an exact solution (separable). In essence, the math appendix should only include what is needed to explain the methods – anything that is more extensive than needed should be either removed or simplified.

In summary, the authors should carefully go through the entire ms to ensure that these aspects are addressed in general. Please also note specific additional comments from the reviewers to address:

1) The manuscript still has trouble delineating spatial vs temporal correlations when it tries to explain different points. For example:

Introduction section: "We pursued this principle in understanding the integrative properties of individual cortical neurons, for which the relevant statistical input patterns are those characterising the dynamically evolving spike trains of their presynaptic partners." This sounds like the manuscript is concerned with temporal issues.

"The optimal response determined by this statistical model, for essentially any setting of parameters, was inherently nonlinear because the additional effect of a presynaptic spike depended on the pattern of spikes that had been previously received from the presynaptic population and thus the integrated effect of multiple spikes could not be computed as a simple linear sum of their individual effects in isolation." This also sounds to me as if a claim that the temporal pattern of inputs is what's leading to nonlinearity.

2) Regarding the errors in the mathematics, we thought that when the authors repaired the solution of Equation 27 they would also fix other instances of this equation in the manuscript. Equation 27 is recapitulated in the mathematical appendix as Equation A91, but the discussion around Equation A91 is clearly wrong. Somehow a different solution from the solution of Equation 27 is obtained; the answer should be equation A92 with Α = 0,Β = B and C = −A. Here’s how the solution goes: we start withζ˙=ζ(1–ζ)[–Α+Βs(t)].

This equation is separable:dζζ(1−ζ)=[–A+Bs(t)]dt.

The left-hand size can be simplified by partial fractions:1ζ(1–ζ)=1ζ+11–ζ,

so that[1ζ+11−ζ]dζ=[−A+Bs(t)]dt.

Both sides of the equation can now be integrated. Assuming 0 < ζ < 1, we getlnζ–ln(1–ζ)=∫[−A+Bs(t)]dt.

or equivalently,lnζ–ln(1–ζ)=v(t)

wherev˙=–A+Bs(t).

Then solving

ζ1−ζ=ev⇒ζ=ev1+ev+11+e−v.

We don’t know what the authors are trying to do with the rest of Section B.2.2, but it appears to be wrong. We also don’t know to what extent fixing this will affect the rest of the results.

---

## [Author Response]

[Editors’ note: the author responses to the first round of peer review follow.]

We included a thorough answer to all points raised by the reviewers. In particular, we substantially simplified the presentation of the paper by rewriting the introduction and the first part of the results section; adding a flowchart to Figure 1 giving a concise summary of the logic of the manuscript; and rewriting several parts of the Methods section to streamline it. Moreover, we highlighted the take-home messages at the beginning of the Discussion. In addition, we also included experiments using NMDA receptor blockade to provide more mechanistic insights; added a supplementary figure demonstrating that our stimulation protocol is sufficiently complex to probe dendritic integration in the relevant input range and added a supplementary figure where we compared the responses of different models to the averaged presynaptic potential instead of the optimal response.

*[…] Overall, two major aspects require consideration, as summarized below with some suggestions. Additional detailed comments from the reviewers are appended below.*

*1) A clearer and simpler version of the paper is needed. A clear line through the reasoning that will allow everyone to see the core of what has been done, and why, needs to be presented. Specifically, the paper needs to be transparent and math that obfuscates should be removed or simplified or explained.*

We substantially simplified the presentation of the math at the beginning of the paper and removed all math that is not essential to follow the logic of the presentation. We believe that the equations that remained in the main text (4 out of 7 in the previous version) are essential to introduce key concepts of the paper. These equations are now explained and illustrated in the text and presented in a highly simplified way.

*The authors are encouraged to present their essential results in a much more straightforward fashion to help elucidate their mathematical derivations. A suggestion is to use a flow chart of the process/protocol (capturing presynaptic firing statistics, fitting, optimizing etc., and lay bare the limitations/assumptions* – *e.g., the output computation f(u) is assumed to be a linear function),*

As suggested, we added a flowchart presenting the logic of the paper to the first figure. This new panel also serves as a roadmap to the rest of the paper. In the flowchart, we highlighted our assumptions, illustrated the relationships between different parts of the paper and emphasized how specific figures in the paper support our conclusions. Moreover, we added a new paragraph describing the structure of the paper at the end of the Introduction and another paragraph in the Results that uses the flowchart to explain the logic of the Results section after introducing the key concepts at the beginning. We believe that these changes will substantially help the reader to follow the presentation of the paper.

*[…] Describe the essential results in words, and point to the mathematical parts in methods/or a mathematical appendix of how and why.*

We removed a substantial fraction of the equations from the main text (see above), and clarified the presentation in both the Results and Methods sections. Our main goal was to keep only those mathematical formulae in the main text that were absolutely essential to follow the logic of the paper. Those details that are only necessary to reproduce our results are included in the Methods section while further details of the derivations have been moved to the mathematical appendix.

*We understand and appreciate that you wish to publish the full mathematical treatment, but most eLife readers won't benefit from the math in its present form. The mathematical results could be made complete and readable in an Appendix, while freeing the Results to show the logic of the model and how it works. It looks complicated because of using the most general formulations (multi-state state transitions, and multi-variate normal with arbitrary state-dependent covariance), but the main results are obtained via somewhat simpler formulations.*

We changed the formula describing the state transition dynamics in the model as suggested by the Reviewers, and this, indeed, simplified substantially the presentation of both the model for the presynaptic statistics and the optimal response (Eq. 15-17). We also added an intuitive explanation of the equations describing the optimal response. We also greatly simplified the derivation of the dendritic approximation of the optimal response which provided a more intuitive justification for the sigmoidal nonlinearity (see subsection “Dendritic approximation of the optimal response”). In the representation of the optimal response, the multivariate (rather than univariate) normal and the state-dependent posterior covariance matrix are necessary consequences of having more than 1 presynaptic neuron and 1 state, respectively, which are both key to our results, and so we could not simplify further this part of the Methods section.

*More detailed mathematical derivations could go into a mathematical appendix if necessary. In this way, people (of diverse backgrounds) would be able to sink their teeth into the contribution, build on it, test it and so on.*

We have restructured the paper following this suggestion. We believe that the paper is now presented in a form that is suitable for readers with diverse backgrounds: researchers with experimental interest can follow the logic of the paper by reading the main text, while readers interested in the mathematical details can consult the Methods section and the Mathematical Appendix. We believe that the minimal amount of mathematical details that we left in the main text is necessary to clearly define the concepts used in the paper and to avoid misunderstandings due to the novelty of our approach, and that these formulae and mathematical concepts are sufficiently lightweight (with the corresponding explanations in the text) even for the non-specialist reader.

*2) Take-home message(s) and biological/biophysical intuition needs to be presented. No biophysical model seems to be included, even simply. Everything seems to be folded into the optimal g(s), so that it is not obvious how best to make a biological/biophysical connection.*

We note that the first paragraph of the section “Nonlinear dendrites closely approximate the optimal response” in the Results starts by considering potential biophysical substrates of the optimal g(s), and indeed that whole section is devoted to show how the optimal response can be (approximately) implemented by a simplified mechanistic, biophysically motivated model of dendritic integration that has enjoyed prominent success and wide popularity in the field (Poirazi & Mel, 2001). Crucially, this mechanistic model has been extensively linked to biophysical processes by previous studies so we felt that by way of linking our theory to this model we do provide biophysical insights. We also devote the first subsection of the Discussion (now expanded, see below) to discussing biophysical substrates.

Nevertheless, to better manage readers’ expectations, we included “disclaimers” upfront, already in the Introduction, and the first section of the Results, explaining how the inherently top-down nature of our approach does not allow us to make specific biophysical predictions, but how we are still able to provide some biophysical insights. In fact, to provide tighter links with biophysical factors, we also included, as an initial attempt, a novel set of experimental data directly probing the contribution of NMDA receptors to optimal input integration (Figure 5). Specifically, we found that after blocking the NMDA currents dendritic integration became incompatible with the known presynaptic statistics, both in the neocortex and in the hippocampus, supporting the crucial role of NMDA currents in matching dendritic integration to the input statistics.

*The derived 'optimal traces' do, in principle, bring a more stringent test to the theory, but in Figure 4 they have access to the whole pre-synaptic dynamics so the optimal trace can be replaced by the averaged pre-synaptic potential,*

Indeed, in Figure 4 we could show the averaged pre-synaptic potential together with the optimal trace. However, even in this test the computation is spike based and the postsynaptic neuron does not have access to the presynaptic membrane potentials. In this case the optimal response is the theoretically best possible approximation to the presynaptic potentials that is achievable to the postsynaptic cell given the limited information conveyed by the irregular firing of the presynaptic neurons. Therefore, in Figure 4 we compared the models’ responses to the optimal response and included an additional supplementary figure (Figure 4—figure supplement 1) where we compare the responses to the averaged presynaptic potential.

*[…] While in Figure 5 the experimental pattern of stimulation is not sufficiently complex to really probe the detailed temporal structure predicted by the theory.*

We agree with the reviewers that the currently used experimental protocols are relatively simplistic compared to the highly structured spatio-temporal stimuli experienced by the dendrites under in vivo conditions. However, we found that despite their relative simplicity, our in vitro stimuli activated the dendritic branches in a similar range (in terms of the integrated postsynaptic effect of synaptic stimulations) as could be expected during in vivo-like situations. To illustrate this point, we prepared a supplementary figure (Figure 1—figure supplement 2) where we compare the total input to a given branch during an in vitro experiment with the total input when the stimuli is sampled from the model matched to in vivo data. We found an excellent match between these two distributions suggesting that our stimulation protocols probed integration in a single dendritic branch in the physiologically relevant input range.

Crucially, our approach will also allow us in the future to modify the in vitro stimulation protocols to make them similar also in their structural details to the patterns typical in vivo. Designing such novel, richer but experimentally still feasible protocols possibly optimised for gaining information about dendritic nonlinearities is a highly non-trivial endeavour on its own right (where collaboration between theory and experiments can indeed bring significant advances) and was thus beyond the scope of the current paper.

*The other figures support qualitative statements. This, for example, could be explicitly stated and rationalized for the reader.*

We feel that Figure 3 is also quantitative in that it shows a direct numerical comparison of the optimal response and the response of the canonical biophysically-motivated model. Figure 2 is indeed qualitative and we now make this clear upfront in our summary flowchart (Figure 1). More generally, we included a roadmap to the paper at the end of the Introduction which also states explicitly that this first step is qualitative.

*Some aspects came through but because of the first point above, there was not always a clear consensus of what the authors intended as the main message(s) from the work. Could these intents (if appropriate) be better explained? For example:i) Presynaptic neurons with correlated inputs should target neighboring regions of the dendritic tree so that the inputs can sum nonlinearly while presynaptic neurons with uncorrelated inputs should map onto different portions of the tree and the inputs sum linearly?ii) An optimizing aspect/principle that could ultimately predict the 'location structure' of synapses as a crucial test of their theory.iii) Optimally decoding the information present in the spikes to do the desired computation depends upon the correlationsiv) Addressing the question of why there are nonlinearities in the first place.v) An optimal way to recover the averaged presynaptic potential only from the pre-synaptic spike trains? Can the biophysics approximate this optimal 'decoding' operation?*

We rewrote the first paragraph of the Discussion where we first highlight the main messages of the paper and then close the paragraph by a simple take-home message. We hope that these changes, together with the restructured Introduction of the paper will help the reader to identify the major contributions and the main messages of our work.

Reviewer #1:

*1) The authors could state that graded synaptic transmission exists (many examples in invertebrates), so that their principle may not apply there, I assume. Is it definitively known that graded synaptic transmission is not present in cortical circuits?*

Synaptic transmission can be modulated by graded signals, such as the somatic depolarisation of the presynaptic neuron (Shu et al., 2006), and indeed, this is why we wrote in the manuscript that “cortical neurons mainly communicate by action potentials”, but this effect is believed to have a minor importance relative to the coding of information by action potentials. We added a sentence about graded synaptic transmission in the Discussion.

*2) It would be helpful if the authors could expand on the discussion regarding biophysical substrate. That is, to fill in the blanks on how and why "NMDA spikes may provide a general solution". Was some modeling (not shown) done to be able to state this? Please provide some additional discussion/rationale/assumptions for making these statements.*

To thoroughly answer this question of the reviewer we made two distinct changes in the manuscript: First, in order to more directly demonstrate the role of NMDA receptors in tuning dendritic nonlinearities to the statistics of the input we tested how well our theory can predict experimental data recorded under their pharmacological blockade. We found that the prediction of our theory became substantially poorer in both the hippocampus and the neocortex after blocking the NMDA receptors supporting their crucial role in matching dendritic integration to the input statistics. Second, we expanded the Discussion regarding the possible role of NMDA receptors in tuning dendritic nonlinearities to state switching dynamics.

*3) Regarding inhibition, the authors discuss potential addition from the perspective of excitatory cells. However, perspectives from inhibitory cells are not expressed. Do the authors expect their principle to hold from the perspective of inhibitory cells (receiving excitatory and/or inhibitory inputs)? The authors should provide some discussion/rationale/assumptions of why this is not included/considered.*

We added a paragraph to the section “Inhibitory Neurons” discussing the possible application of the theory to inhibitory neurons. In short, our theory applies equally well to dendritic integration in inhibitory cells. We focused on pyramidal neurons because their nonlinearities have been much more extensively characterised.

*I do not think it is a given that computation is only from the perspective of excitatory cells? For example, consider the reviews by Klausberger and Somogyi (Science 2008) and Chamberland and Topolnik (Frontiers in Neuroscience 2012).*

We agree with the reviewer that computation is not necessarily exclusive to excitatory neurons. In fact we discuss that they might have a similar role in single neuron computation as excitatory neurons.

Reviewer #2:

*Nonlinear dendritic integration is often seen as a thorn in the foot. Following in vitro work by Larkum, Magee, Stuart, Hausser and others, it is now clear that dendrites introduce various types of nonlinearity to input integration.*

*[…]*

*For these reasons, I believe that even if the supposed role is later shown to be false or incomplete, the article will have had a strong impact on the community.*

We are grateful for reviewer 2’s constructive comments. The wording of several of his sentences motivated us to rewrite our paper in a clearer way potentially more understandable for people with diverse background.

Reviewer #2 (Minor Comments):

*Generally, confusion arises in the number of models considered. I think that, when a comparison between models is the main result, these different models should be mentioned at the beginning of the section. For instance in the last paragraph of the section “The form of the optimal nonlinearity depends on the statistics of presynaptic inputs” one suddenly learns of the actual comparisons being made.*

We agree with the reviewer that it is crucial to clearly introduce the models used in the manuscript when they are first mentioned. Therefore we carefully checked that not only models but also the important concepts are introduced and defined when first presented in the main text. As for the specific section to which the reviewer is referring, we specified the details of the models the first time they are introduced (sections “Optimal input integration is nonlinear” and” The form of the optimal nonlinearity depends on the statistics of presynaptic inputs”) such that when we refer to them later they do not come as a surprise.

*On a first reading, I could not understand what was done in the first paragraph of the section” Nonlinear integration in cortical neurons is matched to their input statistics”. After reading the supplementary, it is clear, but I think a few sentences should be added to describe in a little more details what was actually compared. For instance, one can read from the first sentence that to test the theory, we need to measure presynaptic statistics and derive the optimal nonlinearity, then measure the nonlinearity in vitro and compare the nonlinearities. That is not what was done, I know now, but I could not rule that out from the paragraph as is. It is also difficult to understand end of the second paragraph of the same section on the first go.*

We added a clause to the referred paragraph and clarified that when testing the theory we compared the actual and the predicted responses instead of the form of the nonlinearities themselves. We also clarified what we mean by the “parameters related to the postsynaptic dendritic filtering”.

*What is the experimental stimulus protocol used in the uncaging experiments? How broad is the distribution of ISI? Evenly distributed on Figure 5 and g? Are each burst made of evenly spaced pre-synaptic encaging?*

We added a separate paragraph about the stimulation protocols we used to the Methods section where we describe that in the uncaging experiments we targeted visually identified dendritic spines (1-7 in the cortex, 1-30 in the hippocampus) with a burst of presynaptic uncaging stimuli evenly spaced in time. We varied either the number of stimuli in the burst (1-7 in the neocortex, 1-30 in the hippocampus) or the delay between the uncaging events within the burst (1-20 ms in the neocortex and 0.1-5 ms in the hippocampus) for each dendrite.

*Can you really say overfitting is small?*

In order to demonstrate that the precise match between the predicted and the actual dendritic nonlinearities is not due to overfitting we performed model selection using Bayesian information criterion that penalises more complex models (see also Methods). We found that the extra flexibility of the neocortical and hippocampal models offered by the one additional parameter compared to the independent model was highly unlikely to account for the superior performance of the more complex models. Moreover, the model with second order correlations had the same number of free parameters as the neocortical and hippocampal models, yet it was unable to fit the data even remotely as well. These results indicate that the good performance of the cortical models is not simply due to the larger number of parameters.

To further demonstrate that overfitting is small, we now also conducted a set of validation simulations. In these simulations we first generated synthetic data matching the data recorded in our hippocampal and neocortical experiments (see below), and fitted this synthetic dataset following the same procedure that we used for fitting real experimental data in the paper, using hippocampal and neocortical input statistics. We expected that if the risk of overfitting were *high*, then we would be able to fit our simulated *hippocampal* dataset with neocortical parameters just as well as with *hippocampal* parameters and vice versa.

Specifically, to generate synthetic data matching the data recorded in our hippocampal experiments, we computed the optimal response for a stimulus (i.e. 30 input spikes with various delays) assuming general presynaptic statistics similar to our hippocampal model (Table 1, HP). We added 20% variability to the presynaptic firing rate and assembly activation rate to simulate a possible mismatch between the true population activity statistics and the statistics used for estimating the optimal response. This added variability within the hippocampal statistics that was small compared to the differences between hippocampal and neocortical population activity statistics. The postsynaptic parameters were sampled randomly from a uniform distribution in the realistic range (Figure 5—figure supplement 1). This protocol provided us with a set of traces very similar to the original experimental data recorded from hippocampal CA3 cells. We followed the same protocol, using neocortical statistics (Table 1, NC), to generate synthetic data similar to our neocortical recordings.

Using these datasets we investigated the accuracy of our fitting procedure with both hippocampal and neocortical presynaptic statistics. We found that when the presynaptic statistics used for fitting matched the presynaptic statistics used to generate the data the fitting error was almost zero (Figure 6) and the estimated postsynaptic parameters were similar to the original postsynaptic parameters used to generate the data (Figure 6). (Note that fitting error was much smaller than with actual experimental data because synthetic data lacked intrinsic variability between responses to the same stimulus.) The non-zero error between the original and the estimated postsynaptic parameters is due to the variability of the presynaptic parameters within a given dataset that could be compensated by altering the postsynaptic parameters. However, the postsynaptic parameters could not compensate for differences in the presynaptic parameters as large as between hippocampal and neocortical statistics, resulting in substantially larger fitting errors (Figure 6) and greater differences between the original and fitted parameters (Figure 6). Taken together, these simulations suggest that the stimulation protocol used in our experiments is sufficiently complex to identify the postsynaptic parameters without the danger of overfitting.

Author response image 1.Results of the validation simulations.(**A**) The fitting error was substantially smaller if we used similar presynaptic parameters to fit the validation data than the true parameters used to generate the data. In this case the fitting error was essentially zero since there is no intrinsic variability in the artificial data. (**B**) The error in the postsynaptic parameters (number of cells targeting the dendritic branch, and the postsynaptic filtering time constant) is substantially smaller when the true presynaptic parameters match those used for fitting the model (blue) than when the parameters are mismatched.**DOI:**
http://dx.doi.org/10.7554/eLife.10056.021

*Why exactly the multi-state description in Eq. 11 and 19 if it is never used? Personally, it is only when I saw Eq. S67 that I finally reached the proper mathematical intuitions. Other than the need for simplicity, I am concerned that some of the conclusions would not hold in the multi-state scenario suggested by Eq. 11.*

We simplified the equations in the main text and only use the two state description in the main text and moved the more general, multi-state equations into the mathematical appendix.

*Figure 1: 1B is argued to offer a better fit to optimal than 1C. But there are two differences, correlation and scale. Could it be that in the uncorrelated case, the fit to optimal is not as good when the potential fluctuates on greater amplitudes?*

In this figure we illustrated a case in which the nonlinearity of the optimal postsynaptic response depends only on the correlations between the presynaptic neurons and not their individual activity dynamics or spiking. Therefore we chose the same parameters for the individual presynaptic neurons in the two panels - in fact, the only difference between C and D (note, that the numbering of the panels has changed in the new version of the manuscript) is that all neurons in D are switching to the active state simultaneously while they are independent in C. To illustrate this point we added the membrane potential of one example presynaptic neuron to each panel, on the same scale.

The large difference between the scale of the postsynaptic response is due to the fact that individual fluctuations average out when neurons in the population are independent, while they do not if they are correlated. To control for the differences in the scale of the fluctuations we normalise the error between the actual and predicted responses by the variance of the true signal, whenever we quantify errors throughout the paper. Moreover, as we show in the Methods section, when presynaptic neurons are independent and the computation is linear (as we assume in panel B), then the optimal response is literally the linear combination of the responses to the spike trains of the individual neurons.

*In fact, I don't understand why the uncorrelated case should not have a saturating dendritic nonlinearity to counteract the exponential spiking nonlinearity."*

We argue here that optimal dendritic integration of spikes from different presynaptic neurons is linear if the presynaptic neurons are independent. We are not saying that the optimal response to the spikes from the same presynaptic neuron would be linear in this case. In fact, as the reviewer suggests, spikes from the same neuron should still be integrated in a nonlinear way even if presynaptic neurons are independent (see Pfister et al., 2010). In the Discussion we suggest that the nonlinearities that are private to the individual inputs should be implemented by nonlinear synapses (short term plasticity) instead of nonlinear dendrites. We have now made this point clear in the caption of Figure 1.

Reviewer #3:

*The authors present a detailed mathematical analysis, computations and experimental results aimed at supporting the conclusion that dendritic nonlinearities used to integrate synaptic inputs in a neuron are optimized based upon the statistics of that neuron's presynaptic connections. In particular, the suggested result is that if a neuron's presynaptic connections are uncorrelated, sublinear responses are optimal. If, however, the firing of presynaptic neurons are correlated, supralinear responses are optimal for that neuron. If correct, I would find this conclusion to be interesting.*

This is probably also what the reviewer meant, but just to be precise: presynaptic correlations imply nonlinear responses, which can be sublinear or supralinear depending on the nature of those correlations, see e.g. Figure 2.

*One of the main difficulties with the manuscript is that the details are not easily approachable. There are many pages of complicated mathematics that one must work through, and in many places there hasn't been much attempt to try to explain things in simpler terms to the general reader – I suspect that one will not be able to follow the details here unless one is already an expert.*

We simplified the math presented in the main text and provided an overview of the main concepts in Figure 1. Our goal was to present the most important concepts of the paper in the main text with only the minimal amount of mathematical details necessary and by illustrating the concepts graphically. Therefore we moved the mathematical details necessary for reproducing the results to the Methods section. Additional details about the mathematical derivations are included in the mathematical appendix.

*While the major focus on presynaptic activity is in terms of spikes (and I will grant that a good explanation for this is given), I think it's a concern is that the focus on postsynaptic activity is in terms of subthreshold responses. Granted, the authors say that they can also perform their calculation if the firing rate is substituted for the subthreshold response. This seems like an inconsistency, however – if two or more neurons share synaptic inputs from overlapping sets of presynaptic neurons, and the correlations in the spiking patterns of the presynaptic neurons matter, why is it that correlations in the output spikes of those two or more neurons don't matter? This would seem to be a basic internal inconsistency in the manuscript.*

We agree with the reviewer that mathematical consistency is an important requirement for any theory. According to our theory, correlations between the activities of any groups of cells presynaptic to a given neuron must be handled by the postsynaptic neuron. In the example brought up by the reviewer, correlations between the neurons in the input neurons determine input integration in the output neurons, whereas correlations in the two output neurons will influence integration in neurons reading them. Thus, correlations in the output spikes of the two neurons DO matter, but only for cells that receive inputs from those particular two neurons. More generally, note that our formalism rests on the very objective of making a self-consistent (purely membrane potential-based, or, equivalently, purely firing rate-based) description of network dynamics work when neurons communicate with spikes.

*In addition, while there is a huge amount of detail associated with the mathematical calculations, I found the details associated with the experiments to be a bit too sparse. It would have been useful, for example, to see an image of the neocortical pyramidal cell and the specific synapses which are being stimulated.*

We added the image of the neurons we recorded from in Figure 5 and provided details about the experimental protocols used in this study in a new paragraph in the Methods section.

[Editors' note: the author responses to the re-review follow.]

*All of the reviewers are strongly positive about this work but they also want to ensure that the work is accessible, understandable and widely appreciated. They would like all aspects be made as clear as possible. Thus, pending the following revisions, we expect to accept the paper. Please respond directly to the revisions on a point-by-point basis, revising the paper where needed.*

*1) Double check all equations in the mathematical appendix* – *there was some concern given potential sign errors in other equations (see below). Additional explanatory guidance through the mathematical steps should be provided for those readers who may be interested in following these details. The math is still very complicated.*

We carefully checked the equations in the mathematical appendix and included additional explanatory text to all important steps.

*2) Check and correct potential inconsistencies related to the description of the function s(t). In subsection “Presynaptic statistics” it is still stated that in the limit in which the time step goes to zero the discrete version becomes binary, i.e., zero or 1. In subsection “Inference and the optimal response”, however it's stated that s(t) is a sum of Dirac-delta functions. The two descriptions can't be consistent, of course, unless there is a division by a time step somewhere. There does not appear to be a careful definition in the manuscript, however.*

We clarified the difference between the discrete and continuous versions of the presynaptic spike trains, *s_t_* and s*(t)* in the Methods section (below Eq. 22.) and provided a formal derivation of *s(t)* from *s_t_* in the mathematical appendix (Eq. A62).

*3) Check the solution of equation (22) given in (23) and (24) for potential sign errors.*

We are grateful to the reviewers for pointing out this sign error. We corrected the sign error in Eq. 22 (now Eq. 27) and in the definition of the constant A in the mathematical appendix (B.1).

*4) The additional details about the experiments were appreciated. However, a question arose about the blocking of NMDA channels. It's stated that substantially poorer fits were obtained for this case. Please confirm/clarify whether the postsynaptic filtering parameters were re-fit in this case?*

Yes, we confirm that the postsynaptic filtering parameters were re-fit in this case. We clarified this in the text.

*5) It seems like a bit of an inconsistency that while a big point is made that presynaptic neurons must communicate with spikes, no spiking is considered in the postsynaptic neurons. If spiking is the only way that neurons can communicate (the authors' point), why should one care about the subthreshold response of the postsynaptic neuron? Since so much of the manuscript covers the subthreshold response, something explicit (explanation-wise) should be said about this in the manuscript (and we realize that they consider the case of firing rates).*

We added a sentence clarifying this issue in the Discussion (section about analog communication) by explaining that our theory is fully self-consistent as we assume that the relevant mapping is between the membrane potentials of the presynaptic and the postsynaptic neurons. This implies that the spikes of the neuron called postsynaptic in the manuscript will influence input integration of its own postsynaptic partners, and the form of the integration depends on the statistical dependences in the new presynaptic population.

*6) Include a 'toy model' explanation (without math, or if possible, with straightforward enough math equations that most can follow) that summarizes the link between pairwise correlation (temporal) and dendritic integration (spatial) before any specific statistical models are discussed. This would help the reader follow the more detailed case, and avoid confusion in consideration of later statements in the paper. A good place for this might be at the beginning of the Results section.*

We were somewhat unsure as to what the Reviewers exactly had in mind here, so we added a simple toy explanation of the main idea of the paper from everyday life at the beginning of the Results section. We are happy to omit this paragraph if the reviewers/editor deem it unnecessary.

Moreover, our original intention with Figure 1 was precisely to show a simple, toy-model example where the requirements for linear versus nonlinear integration become evident for the reader without the need for diving into the mathematical details, as we introduce the specific mathematical model in the next section. Nevertheless, we found it impossible to do this before at least the main mathematical concepts (computation vs. implementation) are introduced at the beginning of the Results section.

We also added a short conclusion to summarise the link between input correlation and dendritic integration at the end of this section.

*Specifically, the way the text reads is confusing about when* spatial *correlations (for lack of a better term) between different presynaptic neurons are being discussed, and when* temporal *correlations between inputs are being considered. Part of the issue may be that the mathematical formulation in the main text completely hides issues of temporal correlation in the details. Thus one can be left wondering if one of the two types of correlations might be more important and if the theory could shed light on this.*

We added a new paragraph in the Methods (presynaptic statistics) explaining the source of spatial and temporal correlations. We also clarified that under *correlations* in the manuscript we always mean (cross-)correlations between the spike trains of different presynaptic neurons (*spatial correlations*). When we refer to the correlations between the firing rates of the same neuron at a particular time delay (*temporal correlations*) we use the term *auto-correlation*. We chose this terminology, rather than explicitly stating in every instance whether its spatial or temporal correlations, so as not to overburden the manuscript with too many technical terms while still keeping its terminology consistent.

*For example, consider these comments from the manuscript: "However, if presynaptic neurons became correlated the optimal response became nonlinear and the best linear response was unable to accurately follow the fluctuations in the input (Figure 1)." "Same as (A-C) but for presynaptic neurons exhibiting synchronized switches between a quiescent and an active state (D, bottom), introducing higher order correlations between the neurons." (Figure 2 caption) "In particular, sublinear integration was optimal when presynaptic activities exhibited simple Gaussian random walks and thus they could not contain statistical dependencies beyond second order correlations (Figure 2)." The authors here are trying to provide an intuitive explanation as to why input correlations lead to the observed behavior and this, of course, is good and to be encouraged. In each case, however, what type of correlation is being thought of – spatial or temporal? Further elaboration might provide additional insight.*

In all of these cases we were referring to statistical dependencies between different presynaptic neurons (spatial correlations in the reviewer’s terminology). This is clarified in the Methods section, where we included the following sentence: “In this paper we informally refer to any statistical dependency (second or higher order) between the spike trains of different neurons simply as ‘correlation’.”

[Editors' note: further revisions were requested prior to acceptance, as described below.]

*The reviewers appreciated the revisions done by the authors, but it was felt that the intuition as provided was incomplete and the mathematical appendix was still not clearly presented. Specifically, it was noted that is unclear how that authors distinguish autocorrelation from cross-correlation (spatial) in the formalism and the text, and as this is a main point of the paper, it should be up front early on in the Discussion (and not just in caption of Figure 1 and later in Methods).*

We now tried to clarify which type of correlations we are referring to wherever it was not absolutely clear from the context. We also added text to define auto- and cross-correlations, and the requirements for nonlinear postsynaptic input integration they respectively imply, at the beginning of the Results. We also explain that only correlations between neurons (i.e. spatial correlations) require nonlinear dendritic integration. We also rewrote parts of the Methods to further clarify this in technical detail. As we write there:

“Also note that temporal and spatial correlations can not be studied in complete isolation […] that we did not consider this case worth pursuing.”

*Also, eq A91 has an exact solution (separable).*

Yes, we do realise it, the results in the Methods were in fact based on this insight. We now provide the detailed derivations for this solution in the Appendix B.2 (Eqs. A92-96).

*In essence, the math appendix should only include what is needed to explain the methods* – *anything that is more extensive than needed should be either removed or simplified.*

We simplified the mathematical appendix, and in particular reduced the size of Appendix B.

*1) The manuscript still has trouble delineating spatial vs temporal correlations when it tries to explain different points. For example:*

*Introduction section: "We pursued this principle in understanding the integrative properties of individual cortical neurons, for which the relevant statistical input patterns are those characterising the dynamically evolving spike trains of their presynaptic partners." This sounds like the manuscript is concerned with temporal issues.*

We changed the text to “the relevant statistical input patterns are those characterising the spatio-temporal dynamics of their presynaptic spike trains” to highlight the importance of the correlations between the presynaptic population.

*"The optimal response determined by this statistical model, for essentially any setting of parameters, was inherently nonlinear because the additional effect of a presynaptic spike depended on the pattern of spikes that had been previously received from the presynaptic population and thus the integrated effect of multiple spikes could not be computed as a simple linear sum of their individual effects in isolation." This also sounds to me as if a claim that the temporal pattern of inputs is what's leading to nonlinearity.*

We changed the text to include the following, more explicit explanation:

“The optimal response determined by this statistical model, for essentially any setting of parameters, was inherently nonlinear because the additional effect of a presynaptic spike depended on the pattern of spikes that had been previously received from the presynaptic population. […] Thus, the integrated effect of multiple spikes could not be computed as a simple linear sum of their individual effects in isolation.”

We hope this clarifies the respective roles of temporal and spatial correlations. We note that, technically, spatial correlations only require the nonlinear integration of non-coincident spikes from different cells (and all spikes are non-coincident with probability 1 in continuous time) if there are also temporal correlations in the presynaptic population. Nevertheless, as we argue above (and in the text), it is meaningless to consider spatial correlations without temporal correlations in the context of neural signals, so we hope the reviewers agree with us that the above statement (that we included in the text) is sufficiently accurate.

*2) Regarding the errors in the mathematics, we thought that when the authors repaired the solution of Equation 27 they would also fix other instances of this equation in the manuscript. Equation 27 is recapitulated in the mathematical appendix as Equation A91, but the discussion around Equation A91 is clearly wrong. Somehow a different solution from the solution of Equation 27 is obtained; the answer should be equation A92 with Α= 0,Β= B and C = −A. Here’s how the solution goes: we start with*

ζ˙=ζ(1–ζ)[–Α+Βs(t)].

*This equation is separable:*

dζζ(1−ζ)=[–A+Bs(t)]dt.

*The left-hand size can be simplified by partial fractions:*

1ζ(1–ζ)=1ζ+11–ζ,

*so that*

[1ζ+11−ζ]dζ=[−A+Bs(t)]dt.

*Both sides of the equation can now be integrated. Assuming 0 <*
ζ
*< 1, we get*lnζ–ln(1–ζ)=∫[−A+Bs(t)]dt.

*or equivalently,*

lnζ–ln(1–ζ)=v(t)

*where*

v˙=–A+Bs(t).

*Then solving*

ζ1−ζ=ev⇒ζ=ev1+ev+11+e−v.

We thank the reviewers for providing a detailed derivation of the solution of our Eq. 27. As we noted above, our own solution in the Methods was in fact based on the same logic. We now included our detailed derivations (which we believe is essentially identical to that of the reviewers) in Appendix B.2 (Eqs. A92-96).

We believe that the reviewers meant that the discussion after A91, i.e. Section B.2.2 in the previous version (and therefore B.3.2, which had essentially the same logic) was wrong. We maintain that while the derivation we presented there might have seemed unnecessarily complicated, it was not wrong – though some ends were undeniably left slightly loose, and so we now enclose with this response a more detailed version to show that indeed it leads to the same results as the derivation favoured by the reviewers. The reason why we originally included this more complicated derivation for supralinear integration (Appendix B.2) was that we built on parts of its logic (the separation of time bins with or without a spike) in the derivation for the sublinear case (Appendix B.3) to which the more straightforward method could not be applied (as the relevant differential equation is not separable). Nevertheless, we also managed to rewrite that part in a simpler way, so that it stands alone without all the machinery of the original derivation, so we could dispose of our original derivation altogether and make the whole Appendix self consistent using only the simpler derivation (reducing the size of the B appendix).

*We don’t know what the authors are trying to do with the rest of Section B.2.2, but it appears to be wrong. We also don’t know to what extent fixing this will affect the rest of the results.*

See above for what we were trying to achieve and why we don’t think it was wrong. The new, simpler derivation has made the algebra and the logic easier to follow (we hope) but has not changed any of our results.